# Integrable Deformations from Twistor Space

Lewis T. Cole[a,1], Ryan A. Cullinan[b,2], Ben Hoare[b,3], Joaquin Liniado[c,4] and Daniel C. Thompson[a,d,5]

[a] *Department of Physics, Swansea University, Swansea SA2 8PP, UK*

[b] *Department of Mathematical Sciences, Durham University, Durham DH1 3LE, UK*

[c] *Instituto de Física La Plata (CONICET and Universidad Nacional de La Plata)*
*CC 67 (1900) La Plata, Argentina*

[d] *Theoretische Natuurkunde, Vrije Universiteit Brussel, and the International Solvay Institutes,*
*Pleinlaan 2, B-1050 Brussels, Belgium*

## Abstract

Integrable field theories in two dimensions are known to originate as defect theories of 4d Chern-Simons and as symmetry reductions of the 4d anti-self-dual Yang-Mills equations. Based on ideas of Costello, it has been proposed in work of Bittleston and Skinner that these two approaches can be unified starting from holomorphic Chern-Simons in 6 dimensions. We provide the first complete description of this diamond of integrable theories for a family of deformed sigma models, going beyond the Dirichlet boundary conditions that have been considered thus far.

Starting from 6d holomorphic Chern-Simons theory on twistor space with a particular meromorphic 3-form $\Omega$, we construct the defect theory to find a novel 4d integrable field theory, whose equations of motion can be recast as the 4d anti-self-dual Yang-Mills equations. Symmetry reducing, we find a multi-parameter 2d integrable model, which specialises to the $\lambda$-deformation at a certain point in parameter space. The same model is recovered by first symmetry reducing, to give 4d Chern-Simons with generalised boundary conditions, and then constructing the defect theory.

---

[1] l.t.cole@pm.me

[2] ryan.a.cullinan@durham.ac.uk

[3] ben.hoare@durham.ac.uk

[4] jliniado@iflp.unlp.edu.ar

[5] d.c.thompson@swansea.ac.uk

# 1 Introduction

Developing a systematic understanding of the landscape of integrable 2-dimensional field theories [IFT$_2$] has been a longstanding challenge. In the case of non-linear sigma-models with target space a group manifold $G$, with associated Lie algebra $\mathfrak{g}$, a significant advance was the introduction of integrable deformations of both the principal chiral model [PCM] and the Wess-Zumino-Witten [WZW] model, known respectively as $\eta$-, or Yang-Baxter [YB] [1], and $\lambda$-models [2]. These discoveries have paved the way to wider classes of integrable deformations with applications to worldsheet string theory and holography (for a recent survey see, e.g., [3]).

At the classical level, the integrability of these models can be captured by a flat $\mathfrak{g}^{\mathbb{C}}$-valued Lax connection $\mathrm{D} = \mathrm{d} + \mathcal{L}$ which depends meromorphically on a spectral parameter $\zeta \in \mathbb{CP}^1$. The Poisson algebra of the spatial component $\mathcal{L}_\sigma$ can be entirely characterised by a single meromorphic function $\varphi(\zeta)$ known as the *twist function*. The form of this algebra implies that an infinite tower of higher-spin conserved charges in involution can be constructed [4].

In a remarkable sequence of works [5, 6, 7, 8, 9] it was shown that, by geometrising the spectral parameter plane and considering it as part of space-time, such 2-dimensional integrable models have a 4-dimensional origin as a Chern-Simons type (CS$_4$) theory.

The action of CS$_4$ for a $\mathfrak{g}^{\mathbb{C}}$-valued gauge field $A$ is defined as

$$S_{\mathrm{CS}_4} = \frac{1}{2\pi\mathrm{i}} \int_{\Sigma \times \mathbb{CP}^1} \omega \wedge \mathrm{Tr}\Big(A \wedge \mathrm{d}A + \frac{2}{3}\, A \wedge A \wedge A\Big)\,, \tag{1.1}$$

in which $\omega$ is a meromorphic differential on $\mathbb{CP}^1$. To recover the integrable 2-dimensional theories above, this meromorphic differential should be specified in terms of the relevant twist function as

$$\omega = \varphi(\zeta)\,\mathrm{d}\zeta\,. \tag{1.2}$$

To fully define the theory, the action should be complemented with a choice of boundary conditions on the gauge field at the location of the poles of $\omega$. With suitable boundary conditions in place this 4-dimensional theory localises to a 2-dimensional integrable theory defined on the worldsheet $\Sigma$. For the case of $\eta$- and $\lambda$-models, the relevant boundary conditions were constructed in [9]. See also [10] for a recent review.

Four-dimensional Chern-Simons theory can in turn be understood [11] as coming from a reduction of 6-dimensional holomorphic Chern-Simons [hCS$_6$] on Euclidean twistor space,

$$S_{\mathrm{hCS}_6} = \frac{1}{2\pi\mathrm{i}} \int_{\mathbb{PT}} \Omega \wedge \mathrm{Tr}\Big(\mathcal{A} \wedge \bar{\partial}\mathcal{A} + \frac{2}{3}\, \mathcal{A} \wedge \mathcal{A} \wedge \mathcal{A}\Big)\,. \tag{1.3}$$

An action of the form (1.3) was first considered in [12] as the cubic open string field theory action for the type B topological string. In the context of type B topological string theory, the target space-time is necessarily Calabi-Yau which ensures it is complemented with a trivial canonical bundle, admitting a globally holomorphic top form $\Omega$. Twistor space however is not Calabi-Yau and as such does not possess a trivial canonical bundle. Therefore, to study (1.3) on $\mathbb{PT}$ we instead require that $\Omega$ is a meromorphic $(3,0)$-form on $\mathbb{PT}$ which, in this work, is assumed to be nowhere vanishing. Schematically, this process can be understood as defining a non-compact Calabi-Yau 3-fold by excising the poles of $\Omega$ from $\mathbb{PT}$, which we can now take to be a consistent target space of our type B topological string [13]. This action has also been studied in [14] with a focus on dimensionally reduced gravity and supergravity.

The connection between hCS$_6$ and CS$_4$ can be immediately anticipated since $\mathbb{PT}$ is diffeomorphic to $\mathbb{E}^4 \times \mathbb{CP}^1$ (see appendix A.1 for twistor space conventions). Identifying the $\mathbb{CP}^1$ with the spectral parameter plane, we specify a reduction ansatz, known as *symmetry reduction*, which identifies the worldsheet $\Sigma \hookrightarrow \mathbb{E}^4$. The details, however, are subtle. While $\omega$ has both zeroes and poles, $\Omega$ only has

poles (the zeroes arise from the data that specifies the symmetry reduction). Moreover, suitable boundary conditions on $\mathcal{A}$ are less well understood and currently only explicitly known for a limited class of theories, primarily Dirichlet-type boundary conditions that give, e.g., the principal chiral model with Wess-Zumino term. The generalisation away from Dirichlet boundary conditions necessary to recover lines of continuous integrable deformations, including the aforementioned $\eta$- and $\lambda$-models, is not known. Indeed, obstacles to the construction of integrable deformations from 6 dimensions were highlighted in [15].

Starting in 6 dimensions, an appealing prospect is to swap the order of symmetry reduction and localisation. Indeed performing the integration over $\mathbb{CP}^1$ (localising to the poles of $\Omega$) results in a 4-dimensional theory. Here the avatar of integrability is that the equations of motions can be recast as an anti-self-dual Yang-Mills [ASDYM] equation. The connection between integrable equations and ASDYM has long been known (see [16]), and ASDYM has been shown to provide a 4-dimensional analogue of 2-dimensional rational CFTs (the WZW model in particular) [17, 18, 19]. We then anticipate a return to the same 2-dimensional integrable model by performing symmetry reduction. This gives rise to a diamond correspondence of theories

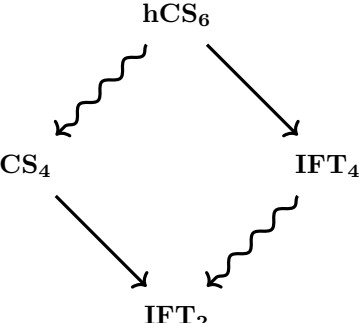

in which the wavy arrows indicate symmetry reduction and straight arrows indicate localisation (integration over $\mathbb{CP}^1$). In the context of integrable deformations of $\text{IFT}_2$, the right-hand side of this diamond is less well understood. As such, a key goal of the present work is to develop this side of the diamond for deformed models, and in particular for $\lambda$-type deformations of the WZW and coupled WZW models.

Briefly, the key results of this work are:

① We establish the consequence of a new class of boundary conditions for $\text{hCS}_6$. These reduce to a wider class of boundary conditions in $\text{CS}_4$ than have previously been considered (relaxing the assumption of an isotropic subalgebra of the defect algebra).

② Integrating over $\mathbb{CP}^1$ results in a novel multi-parametric $\text{IFT}_4$ whose equations of motion can be recast in terms of an anti-self-dual Yang-Mills connection. This new $\text{IFT}_4$ exhibits two semi-local symmetries, which can be understood as the residual symmetries preserving the boundary conditions. For each of these two semi-local symmetries, the Noether currents can be used to construct two inequivalent Lax formulations of the dynamics.

③ Upon symmetry reduction, this $\text{IFT}_4$ descends to the 2-field $\lambda$-type $\text{IFT}_2$ of [20] providing a new multi-parametric sigma-model example of the Ward conjecture [21]. Generically the semi-local symmetries of the $\text{IFT}_4$ reduce to global symmetries of the $\text{IFT}_2$ and the two Lax formulations of the $\text{IFT}_4$ give rise to two Lax connections of $\text{IFT}_2$.

④ When the symmetry reduction constraints are aligned to these semi-local symmetries, the $\text{IFT}_2$ symmetries are enhanced to either affine or fully local (gauge) symmetries. In the latter case, the $\text{IFT}_2$ becomes the standard (1-field) $\lambda$-model.

## 2 Holomorphic 6-Dimensional Chern-Simons Theory

Our primary interest in this work will be the $\text{hCS}_6$ diamond containing the $\lambda$-deformed $\text{IFT}_2$ originally constructed in [2]. By proposing a carefully chosen set of boundary conditions, we will be able to find a diamond of theories that arrives at a multi-parametric class of integrable $\lambda$-deformations between coupled WZW models.

To this end we restrict our study of $\text{hCS}_6$, defined by the action

$$S_{\text{hCS}_6} = \frac{1}{2\pi\text{i}} \int_{\mathbb{PT}} \Omega \wedge \text{Tr}\Big(\mathcal{A} \wedge \bar{\partial}\mathcal{A} + \frac{2}{3}\,\mathcal{A} \wedge \mathcal{A} \wedge \mathcal{A}\Big)\ , \tag{2.1}$$

to the case where the $(3,0)$-form is given by, in the basis of $(1,0)$-forms defined in appendix A.2,

$$\Omega = \frac{1}{2}\,\Phi\, e^0 \wedge e^A \wedge e_A\ , \qquad \Phi = \frac{K}{\langle\pi\alpha\rangle\langle\pi\tilde{\alpha}\rangle\langle\pi\beta\rangle^2}\ . \tag{2.2}$$

Here, we view $\mathbb{PT}$ as diffeomorphic to $\mathbb{E}^4 \times \mathbb{CP}^1$ and adopt the standard coordinates $x^{AA'}$ on $\mathbb{E}^4$ and homogeneous coordinates $\pi_{A'}$ on $\mathbb{CP}^1$. The constant spinors $\alpha$, $\tilde{\alpha}$ and $\beta$ should be understood as part of the definition of the model. See appendix A.1 for further details of twistor notation and conventions. The gauge field is similarly written in the basis of $(0,1)$-forms as

$$\mathcal{A} = \mathcal{A}_0\,\bar{e}^0 + \mathcal{A}_A\,\bar{e}^A\ , \tag{2.3}$$

and the action is invariant under shifts of $\mathcal{A}$ by any $(1,0)$-form, i.e. $\mathcal{A} \mapsto \mathcal{A} + \rho$ where $\rho \in \Omega^{(1,0)}(\mathbb{PT})$.

The first step in studying the 6-dimensional theory is to impose conditions ensuring the vanishing of the 'boundary' term that appears in the variation of the action

$$0 = \int_{\mathbb{PT}} \bar{\partial}\Omega \wedge \text{Tr}\big(\mathcal{A} \wedge \delta\mathcal{A}\big)\ . \tag{2.4}$$

Since $\Omega$ is meromorphic, as opposed to holomorphic, this receives contributions from the poles at $\alpha$, $\tilde{\alpha}$, and $\beta$. We assume that Dirichlet conditions $\mathcal{A}_A|_{\pi=\beta} = 0$ are imposed at the second-order pole. At the first-order poles, we can then evaluate the integral over $\mathbb{CP}^1$ to obtain[6] the condition

$$\frac{1}{\langle\alpha\tilde{\alpha}\rangle\langle\alpha\beta\rangle^2} \int_{\mathbb{E}^4} \text{vol}_4\,\varepsilon^{AB}\text{Tr}\big(\mathcal{A}_A\delta\mathcal{A}_B\big)\big|_{\pi=\alpha} = \frac{1}{\langle\alpha\tilde{\alpha}\rangle\langle\tilde{\alpha}\beta\rangle^2} \int_{\mathbb{E}^4} \text{vol}_4\,\varepsilon^{AB}\text{Tr}\big(\mathcal{A}_A\delta\mathcal{A}_B\big)\big|_{\pi=\tilde{\alpha}}\ . \tag{2.5}$$

For reasons that will shortly become apparent, let us introduce a unit norm spinor $\mu^A$ about which we can expand any spinor $X^A$ as

$$X^A = [X\hat{\mu}]\mu^A - [X\mu]\hat{\mu}^A\ . \tag{2.6}$$

Expanding the gauge field components in terms of the basis $\mu^A$ and $\hat{\mu}^A$, and solving locally pointwise on $\mathbb{E}^4$, this condition may be written as

$$\frac{1}{\langle\alpha\beta\rangle^2}\,\text{Tr}\big([\mathcal{A}\mu][\delta\mathcal{A}\hat{\mu}] - [\mathcal{A}\hat{\mu}][\delta\mathcal{A}\mu]\big)\big|_{\pi=\alpha} = \frac{1}{\langle\tilde{\alpha}\beta\rangle^2}\,\text{Tr}\big([\mathcal{A}\mu][\delta\mathcal{A}\hat{\mu}] - [\mathcal{A}\hat{\mu}][\delta\mathcal{A}\mu]\big)|_{\pi=\tilde{\alpha}}\ . \tag{2.7}$$

The boundary conditions we are led to consider are

$$[\mathcal{A}\mu]|_{\pi=\alpha} = \sigma\,\frac{\langle\alpha\beta\rangle}{\langle\tilde{\alpha}\beta\rangle}\,[\mathcal{A}\mu]|_{\pi=\tilde{\alpha}}\ , \qquad [\mathcal{A}\hat{\mu}]|_{\pi=\alpha} = \sigma^{-1}\,\frac{\langle\alpha\beta\rangle}{\langle\tilde{\alpha}\beta\rangle}\,[\mathcal{A}\hat{\mu}]|_{\pi=\tilde{\alpha}}\ , \tag{2.8}$$

---

[6]To compute the boundary variation of the action, we have used the identities $e^C \wedge e_C \wedge \bar{e}^A \wedge \bar{e}^B = -2\,\text{vol}_4\,\varepsilon^{AB}$ (where $\text{vol}_4 = \text{d}x^0 \wedge \text{d}x^1 \wedge \text{d}x^2 \wedge \text{d}x^3$) and

$$\frac{1}{2\pi\text{i}} \int_{\mathbb{CP}^1} e^0 \wedge \bar{e}^0\,\bar{\partial}_0\Big(\frac{1}{\langle\pi\alpha\rangle}\Big)f(\pi) = f(\alpha)\ .$$

where we have introduced the free parameter $\sigma$, which will play the role of the deformation parameter in the IFT$_4$.

Let us note that these boundary conditions are invariant under the following discrete transformations

$$\alpha \leftrightarrow \tilde{\alpha} \ , \qquad \sigma \mapsto \sigma^{-1} \ , \tag{2.9}$$

$$\mu \mapsto \hat{\mu} \ , \qquad \sigma \mapsto \sigma^{-1} \ . \tag{2.10}$$

These will descend to transformations that leave the IFT$_4$ invariant.

## 2.1 Residual Symmetries and Edge Modes

A general feature of Chern-Simons theory with a boundary is the emergence of propagating edge modes as a consequence of the violation of gauge symmetry by boundary conditions. A similar effect underpins the emergence of the dynamical field content of the lower dimensional theories that descend from hCS$_6$. Generally, group-valued degrees of freedom, here denoted by $h$ and $\tilde{h}$, would be sourced at the locations of the poles of $\Omega$. If however, the boundary conditions (2.8) admit residual symmetries, then these will result in symmetries of the IFT$_4$ potentially mixing the $h$ and $\tilde{h}$ degrees of freedom. These may be global symmetries, gauge symmetries, or semi-local symmetries depending on the constraints imposed by the boundary conditions. It is thus important to understand the nature of any residual symmetry preserved by the boundary conditions (2.8).

Gauge transformations act on the hCS$_6$ gauge field as

$$\hat{g}: \quad \mathcal{A} \mapsto \hat{g}^{-1}\mathcal{A}\hat{g} + \hat{g}^{-1}\bar{\partial}\hat{g} \ . \tag{2.11}$$

In the bulk, i.e. away from the poles of $\Omega$, these are unconstrained, but at the poles they will only leave the action invariant if they preserve the boundary conditions. For later convenience, we will denote the values of the gauge transformation parameters at the poles by

$$\hat{g}|_\alpha = r \ , \quad \hat{g}|_{\tilde{\alpha}} = \tilde{r} \ , \quad \hat{g}|_\beta = \ell^{-1} \ . \tag{2.12}$$

Firstly, the transformation acting at $\beta$ must preserve the constraint $\mathcal{A}_A|_\beta = 0$. Initially, one might suppose that only constant $\ell$ would preserve this boundary condition, but in fact it is sufficient for $\ell$ to be holomorphic with respect to the complex structure defined by $\beta$

$$\beta^{A'}\partial_{AA'}\ell = 0 \quad \Rightarrow \quad \frac{1}{\langle\alpha\beta\rangle}\,\alpha^{A'}\partial_{AA'}\ell = \frac{1}{\langle\tilde{\alpha}\beta\rangle}\,\tilde{\alpha}^{A'}\partial_{AA'}\ell \ . \tag{2.13}$$

These differential constraints arise from the fact that the anti-holomorphic vector fields $\bar{\partial}_A = \pi^{A'}\partial_{AA'}$ are valued in $\mathcal{O}(1)$. In other words, they depend explicitly on the $\mathbb{CP}^1$ coordinate (see appendix A.1 for more details).

Secondly, the transformations acting at $\alpha$ and $\tilde{\alpha}$ must preserve the boundary conditions (2.8), implying the constraints

$$\tilde{r} = r \ ,$$

$$\frac{1}{\langle\alpha\beta\rangle}\,\mu^A\alpha^{A'}\partial_{AA'}r = \frac{\sigma}{\langle\tilde{\alpha}\beta\rangle}\,\mu^A\tilde{\alpha}^{A'}\partial_{AA'}r \ ,$$

$$\frac{1}{\langle\alpha\beta\rangle}\,\hat{\mu}^A\alpha^{A'}\partial_{AA'}r = \frac{\sigma^{-1}}{\langle\tilde{\alpha}\beta\rangle}\,\hat{\mu}^A\tilde{\alpha}^{A'}\partial_{AA'}r \ . \tag{2.14}$$

These residual symmetries are neither constant (i.e. global symmetries) nor fully local (i.e. gauge symmetries). Instead, we expect that our IFT$_4$ should exhibit two semi-local symmetries subject to the above differential constraints, akin to the semi-local symmetries of the 4d WZW model first identified in [17, 18][7].

---

[7]Complementary to this perspective, the WZW$_4$ algebra can also be obtained as a global symmetry of five-dimensional Kähler Chern-Simons on a manifold with boundary [22].

**Symmetry reduction**  As we progress around the diamond, we will perform 'symmetry reduction' (see § 4 and § 5 for details). In essence, this will mean we restrict to fields and gauge parameters that are independent of two directions, i.e. they obey the further differential constraints (where $\gamma^{A'}$ is some constant spinor)

$$\mu^A \gamma^{A'} \partial_{AA'} \hat{g} = 0 \ , \quad \hat{\mu}^A \hat{\gamma}^{A'} \partial_{AA'} \hat{g} = 0 \ . \tag{2.15}$$

We can then predict some special points in the lower dimensional theories by considering how these differential constraints interact with those imposed by the boundary conditions. Generically, these four differential constraints (two from the boundary conditions and two from symmetry reduction) will span a copy of $\mathbb{E}^4$ at each pole, meaning that only constant transformations (i.e. global symmetries) will survive. However, if the symmetry reduction is carefully chosen, the two sets of constraints may partially or entirely coincide. In the case that they entirely coincide, the lower dimensional symmetry parameter will be totally unconstrained, meaning that the $\text{IFT}_2$ will possess a gauge symmetry. Alternatively, if the constraints partially coincide then the lower dimensional theory will have a symmetry with free dependence on half the coordinates, e.g. the chiral symmetries of the 2d WZW model.

# 3   Localisation of $\text{hCS}_6$ to $\text{IFT}_4$

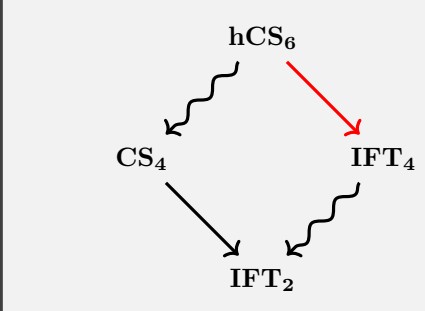

Let us now proceed in navigating the top right-hand side of the diamond. By integrating over $\mathbb{CP}^1$ we will 'localise' $\text{hCS}_6$ on $\mathbb{PT}$ to an effective theory defined on $\mathbb{E}^4$. This resulting theory is 'integrable' in the sense that its equations of motion can be encoded in an anti-self-dual connection.

The localisation analysis is naturally presented in terms of new variables $\mathcal{A}'$ and $\hat{h}$, which are related to the fundamental field by

$$\mathcal{A} = \hat{h}^{-1} \mathcal{A}' \hat{h} + \hat{h}^{-1} \bar{\partial} \hat{h} \ . \tag{3.1}$$

However, there is some redundancy in this new parametrisation. There are internal gauge transformations (leaving $\mathcal{A}$ invariant) given by

$$\hat{g}: \quad \mathcal{A}' \mapsto \hat{g}^{-1} \mathcal{A}' \hat{g} + \hat{g}^{-1} \bar{\partial} \hat{g} \ , \qquad \hat{h} \mapsto \hat{g}^{-1} \hat{h} \ . \tag{3.2}$$

These allow us to impose the constraint $\mathcal{A}'_0 = 0$, i.e. it has no leg in the $\mathbb{CP}^1$-direction. This is done so that $\mathcal{A}'$ may be interpreted as an anti-self-dual Yang-Mills connection on $\mathbb{E}^4$.

There are still internal gauge transformations that are $\mathbb{CP}^1$-independent, and we can use these to fix the value of $\hat{h}$ at one pole. We will therefore impose the additional constraint $\hat{h}|_\beta = \text{id}$ so that we have resolved this internal redundancy. The values of $\hat{h}$ at the remaining poles

$$\hat{h}|_\alpha = h \ , \qquad \hat{h}|_{\tilde{\alpha}} = \tilde{h} \ , \tag{3.3}$$

will be dynamical edge modes as a consequence of the violation of gauge symmetry by boundary conditions. As we will now see, the entire action localises to a theory on $\mathbb{E}^4$ depending only on these edge modes.

The hCS$_6$ action is written in these new variables as

$$S_{\text{hCS}_6} = \frac{1}{2\pi \mathrm{i}} \int_{\mathbb{PT}} \Omega \wedge \text{Tr}\big(\mathcal{A}' \wedge \bar{\partial}\mathcal{A}'\big) + \frac{1}{2\pi \mathrm{i}} \int_{\mathbb{PT}} \bar{\partial}\Omega \wedge \text{Tr}\big(\mathcal{A}' \wedge \bar{\partial}\hat{h}\hat{h}^{-1}\big)$$
$$- \frac{1}{6\pi \mathrm{i}} \int_{\mathbb{PT}} \Omega \wedge \text{Tr}\big(\hat{h}^{-1}\bar{\partial}\hat{h} \wedge \hat{h}^{-1}\bar{\partial}\hat{h} \wedge \hat{h}^{-1}\bar{\partial}\hat{h}\big) \ . \tag{3.4}$$

The cubic term in $\mathcal{A}'$ has dropped out since we have imposed $\mathcal{A}'_0 = 0$. Inspecting the terms in our action involving $\hat{h}$, we see that the second term localises to the poles due to the anti-holomorphic derivative acting on $\Omega$. The third term similarly localises to the poles. For this, we consider a manifold whose boundary is $\mathbb{PT}$.[8] We take the 7-manifold $\mathbb{PT} \times [0,1]$ and extend our field $\hat{h}$ over this interval. We do this by choosing a smooth homotopy to a constant map, such that it's restriction to $\mathbb{PT} \times \{0\}$ coincides with $\hat{h}$. Denoting this extension with the same symbol, we see that the third term in our action may be equivalently written as

$$- \frac{1}{6\pi \mathrm{i}} \int_{\mathbb{PT}\times[0,1]} \mathrm{d}\Big[\Omega \wedge \text{Tr}\big(\hat{h}^{-1}\mathrm{d}\hat{h} \wedge \hat{h}^{-1}\mathrm{d}\hat{h} \wedge \hat{h}^{-1}\mathrm{d}\hat{h}\big)\Big] \ . \tag{3.5}$$

Then, using the closure of the Wess-Zumino 3-form and the fact that all of the holomorphic legs on $\mathbb{PT}$ are saturated by $\Omega$, this is equal to

$$S_{\text{WZ}_4} = - \frac{1}{6\pi \mathrm{i}} \int_{\mathbb{PT}\times[0,1]} \bar{\partial}\Omega \wedge \text{Tr}\big(\hat{h}^{-1}\mathrm{d}\hat{h} \wedge \hat{h}^{-1}\mathrm{d}\hat{h} \wedge \hat{h}^{-1}\mathrm{d}\hat{h}\big) \ . \tag{3.6}$$

Therefore, this contribution also localises, meaning that the only information contained in the field $\hat{h} : \mathbb{PT} \to G$ are its values[9] at the poles of $\Omega$. Explicitly, this contribution is given by[10]

$$S_{\text{WZ}_4} = - \frac{K}{\langle\alpha\tilde{\alpha}\rangle} \int_{\mathbb{E}^4\times[0,1]} \text{vol}_4 \wedge \mathrm{d}\rho\,\varepsilon^{AB}\bigg[ \frac{1}{\langle\alpha\beta\rangle^2}\,\text{Tr}\big(h^{-1}\partial_\rho h \cdot \alpha^{A'}h^{-1}\partial_{AA'}h \cdot \alpha^{B'}h^{-1}\partial_{BB'}h\big)$$
$$- \frac{1}{\langle\tilde{\alpha}\beta\rangle^2}\,\text{Tr}\big(\tilde{h}^{-1}\partial_\rho\tilde{h} \cdot \tilde{\alpha}^{A'}\tilde{h}^{-1}\partial_{AA'}\tilde{h} \cdot \tilde{\alpha}^{B'}\tilde{h}^{-1}\partial_{BB'}\tilde{h}\big)\bigg] \tag{3.7}$$
$$= \frac{K}{6\langle\alpha\tilde{\alpha}\rangle} \int_{\mathbb{E}^4\times[0,1]} \bigg[ \frac{1}{\langle\alpha\beta\rangle^2}\,\mu_\alpha \wedge \text{Tr}\big(h^{-1}\mathrm{d}h\big)^3 - \frac{1}{\langle\tilde{\alpha}\beta\rangle^2}\,\mu_{\tilde{\alpha}} \wedge \text{Tr}\big(\tilde{h}^{-1}\mathrm{d}\tilde{h}\big)^3\bigg] \ ,$$

where

$$\mu_\alpha = \varepsilon_{AB}\alpha_{A'}\alpha_{B'}\,\mathrm{d}x^{AA'} \wedge \mathrm{d}x^{BB'} \ , \qquad \mu_{\tilde{\alpha}} = \varepsilon_{AB}\tilde{\alpha}_{A'}\tilde{\alpha}_{B'}\,\mathrm{d}x^{AA'} \wedge \mathrm{d}x^{BB'} \ , \tag{3.8}$$

are the $(2,0)$-forms with respect to the complex structure on $\mathbb{E}^4$ defined by $\alpha_{A'}$ and $\tilde{\alpha}_{A'}$ respectively.[11]

Knowing that the latter two terms in the action (3.4) localise to the poles, we are one step closer to deriving the IFT$_4$. There are two unresolved problems: the first term is still a genuine bulk term; and the second term contains $\mathcal{A}'$, rather than being written exclusively in terms of the fields $h$ and $\tilde{h}$. Both of these problems will be resolved by invoking the bulk equations of motion for $\mathcal{A}'$. This will completely specify its $\mathbb{CP}^1$-dependence, and, combined with the boundary conditions, we will then be able to solve for $\mathcal{A}'$ in terms of the edge modes $h$ and $\tilde{h}$.

---

[8]More generally, a manifold whose boundary is a disjoint union of copies of $\mathbb{PT}$.

[9]For higher order poles in $\Omega$, the $\mathbb{CP}^1$-derivatives of $\hat{h}$ would also contribute to the action.

[10]In principle there are also contributions from the double pole at $\beta$ both in this term and the second term in the action (3.4). Since this is a double pole, these contributions may depend on $\partial_0\hat{h}|_\beta$, which is unconstrained. However, one can check that they vanish using just the boundary conditions $\mathcal{A}_A|_\beta = 0$ and internal gauge-fixing $\hat{h}|_\beta = \text{id}$. Alternatively, we may use part of the residual external gauge symmetry to fix $\partial_0\hat{h}|_\beta = 0$, which ensures such contributions vanish.

[11]Here we are using the fact that $\mathbb{E}^4$ is endowed with a hyper-Kähler structure such that there is a $\mathbb{CP}^1$ space of complex structures (see appendix A.1).

Varying the first term in the action, which is the only bulk term, we find the equation of motion $\bar{\partial}_0 \mathcal{A}'_A = 0$, which implies that these components are holomorphic. Combined with the knowledge that $\mathcal{A}'_A$ has homogeneous weight 1, we deduce that the $\mathbb{CP}^1$-dependence is given by $\mathcal{A}'_A = \pi^{A'} A_{AA'}$ where $A_{AA'}$ is $\mathbb{CP}^1$-independent.

Turning our attention to the boundary conditions, we first consider the double pole where we have imposed $\mathcal{A}_A|_\beta = 0$. Recalling that $\hat{h}|_\beta = \mathrm{id}$, this simply translates to $\mathcal{A}'_A|_\beta = 0$. This tells us that $\mathcal{A}'_A = \langle \pi\beta \rangle B_A$ for some $B_A$, hence $A_{AA'} = \beta_{A'} B_A$. Therefore, we have that

$$\mathcal{A}_A = \langle \pi\beta \rangle \mathrm{Ad}_{\hat{h}}^{-1} B_A + \pi^{A'} \hat{h}^{-1} \partial_{AA'} \hat{h} \ . \tag{3.9}$$

The solution for $B_A$ found by solving the remaining two boundary conditions (2.8) is written more concisely if we introduce some notation. We will make extensive use of the operators

$$U_\pm = \left(1 - \sigma^{\pm 1}\Lambda\right)^{-1} \ , \qquad \Lambda = \mathrm{Ad}_{\tilde{h}}^{-1} \mathrm{Ad}_h \ , \tag{3.10}$$

which enjoy the useful identities

$$U_+^T + U_- = \mathrm{id} \ , \qquad U_\pm \Lambda = -\sigma^{\mp 1} U_\mp^T \ , \tag{3.11}$$

where transposition is understood to be with respect to the ad-invariant inner product on $\mathfrak{g}$. In terms of the components of $\hat{h}^{-1}\partial_{AA'}\hat{h}$, defined with useful normalisation factors,

$$\begin{aligned} j &= \langle \alpha\beta \rangle^{-1} \mu^A \alpha^{A'} h^{-1} \partial_{AA'} h \ , & \widehat{j} &= \langle \alpha\beta \rangle^{-1} \hat{\mu}^A \alpha^{A'} h^{-1} \partial_{AA'} h \ , \\ \widetilde{j} &= \langle \tilde{\alpha}\beta \rangle^{-1} \mu^A \tilde{\alpha}^{A'} \tilde{h}^{-1} \partial_{AA'} \tilde{h} \ , & \widehat{\widetilde{j}} &= \langle \tilde{\alpha}\beta \rangle^{-1} \hat{\mu}^A \tilde{\alpha}^{A'} \tilde{h}^{-1} \partial_{AA'} \tilde{h} \ , \end{aligned} \tag{3.12}$$

we find that the solutions to the remaining boundary conditions may be written as

$$\mathrm{Ad}_h^{-1} B_A = \widehat{b}\mu_A - b\hat{\mu}_A \ , \qquad b = U_+(j - \sigma\widetilde{j}) \ , \qquad \widehat{b} = U_-(\widehat{j} - \sigma^{-1}\widehat{\widetilde{j}}) \ , \tag{3.13}$$

$$\mathrm{Ad}_{\tilde{h}}^{-1} B_A = \widehat{\widetilde{b}}\mu_A - \widetilde{b}\hat{\mu}_A \ , \qquad \widetilde{b} = U_-^T(\widetilde{j} - \sigma^{-1}j) \ , \qquad \widehat{\widetilde{b}} = U_+^T(\widehat{\widetilde{j}} - \sigma\widehat{j}) \ . \tag{3.14}$$

Note that $b = \mathrm{Ad}_h^{-1}[B\mu]$, $\widehat{b} = \mathrm{Ad}_h^{-1}[B\hat{\mu}]$, etc., and that $b, \widetilde{b}, \widehat{b}$ and $\widehat{\widetilde{b}}$ are related as

$$\widetilde{b} - \widetilde{j} = \sigma^{-1}(b - j) \ , \qquad \widehat{\widetilde{b}} - \widehat{\widetilde{j}} = \sigma(\widehat{b} - \widehat{j}) \ . \tag{3.15}$$

Recovering the IFT$_4$ is then simple. The first term in the action (3.4) vanishes identically on shell, and we can substitute in our solution for $\mathcal{A}'$ in terms of $h$ and $\tilde{h}$ to get a 4d theory only depending on these edge modes. This results in the action

$$\begin{aligned} S_{\mathrm{IFT}_4} &= \frac{1}{2\pi\mathrm{i}} \int_{\mathbb{PT}} \bar{\partial}\Omega \wedge \mathrm{Tr}\left(\mathcal{A}' \wedge \bar{\partial}\hat{h}\hat{h}^{-1}\right) + S_{\mathrm{WZ}_4} \\ &= \frac{K}{\langle \alpha\tilde{\alpha} \rangle} \int_{\mathbb{E}^4} \mathrm{vol}_4 \, \mathrm{Tr}\left(b(\widehat{j} - \Lambda^T\widehat{\widetilde{j}}) - \widehat{b}(j - \Lambda^T\widetilde{j})\right) + S_{\mathrm{WZ}_4} \\ &= \frac{K}{\langle \alpha\tilde{\alpha} \rangle} \int_{\mathbb{E}^4} \mathrm{vol}_4 \, \mathrm{Tr}\left(j(U_+^T - U_-)\widehat{j} + \widetilde{j}(U_+^T - U_-)\widehat{\widetilde{j}} - 2\sigma\widetilde{j}\,U_+^T\,\widehat{j} + 2\sigma^{-1}j\,U_-\,\widehat{\widetilde{j}}\right) + S_{\mathrm{WZ}_4} \ , \end{aligned} \tag{3.16}$$

where

$$S_{\mathrm{WZ}_4} = \frac{K}{\langle \alpha\tilde{\alpha} \rangle} \int_{\mathbb{E}^4 \times [0,1]} \mathrm{vol}_4 \wedge \mathrm{d}\rho \, \mathrm{Tr}\left(h^{-1}\partial_\rho h \cdot [j, \widehat{j}] - \tilde{h}^{-1}\partial_\rho \tilde{h} \cdot [\widetilde{j}, \widehat{\widetilde{j}}]\right) \ . \tag{3.17}$$

Observe that the 4d IFT (3.16) with (3.17) is mapped into itself under the formal transformation

$$h \leftrightarrow \tilde{h} \ , \qquad \alpha \leftrightarrow \tilde{\alpha} \ , \qquad \sigma \mapsto \sigma^{-1} \ , \tag{3.18}$$

interchanging the positions of the two poles. This directly follows from the invariance (2.9) of the $\mathrm{hCS}_6$ boundary conditions. On the other hand, looking at how the transformation (2.10) descends to the $\mathrm{IFT}_4$, we find[12]

$$j \mapsto \widehat{\jmath} \ , \qquad \widehat{\jmath} \mapsto -j \ , \qquad \widetilde{\jmath} \mapsto \widehat{\widetilde{\jmath}} \ , \qquad \widehat{\widetilde{\jmath}} \mapsto -\widetilde{\jmath} \ , \qquad \sigma \mapsto \sigma^{-1} \ . \tag{3.19}$$

It is then straightforward to check that the action (3.16) with (3.17) is invariant under this transformation.

Let us emphasise that, to our knowledge, the $\mathrm{IFT}_4$ described by the action (3.16) with (3.17) has not been considered in the literature before. In the following subsections we will study some properties of this model starting with its symmetries, and moving onto its equations of motion and their relation to the 4d ASDYM equations.

## 3.1 Symmetries of the $\mathrm{IFT}_4$

Having derived the action functional for the $\mathrm{IFT}_4$, we will now examine those symmetries that leave this action invariant. While they may not be obvious from simply looking at the action, in § 2.1 we leveraged the $\mathrm{hCS}_6$ description to predict the symmetries of the $\mathrm{IFT}_4$. These may then be verified by explicit computation.

To this end, we recall that the $\mathrm{hCS}_6$ gauge transformations act as

$$\hat{g} : \quad \mathcal{A} \mapsto \hat{g}^{-1} \mathcal{A} \hat{g} + \hat{g}^{-1} \bar{\partial} \hat{g} \ , \tag{3.20}$$

and we denoted the value of this transformation parameter at the poles by

$$\hat{g}|_\alpha = r \ , \quad \hat{g}|_{\tilde{\alpha}} = \tilde{r} \ , \quad \hat{g}|_\beta = \ell^{-1} \ . \tag{3.21}$$

Tracing through the derivation above, we find that these result in an induced action on the $\mathrm{IFT}_4$ fields,

$$(\ell, r, \tilde{r}) : \quad h \mapsto \ell h r \ , \quad \tilde{h} \mapsto \ell \tilde{h} \tilde{r} \ , \tag{3.22}$$

where $\ell$, $r$ and $\tilde{r}$ obey the constraints (2.13) and (2.14) respectively. One can explicitly verify that the $\mathrm{IFT}_4$ is indeed invariant under these transformations. Key to this is exploiting a Polyakov-Wiegmann identity such the variation of $S_{\mathrm{WZ}_4}$ in eq. (3.17) produces a total derivative. This gives a contribution on $\mathbb{E}^4$ that cancels the variation of the remainder of eq. (3.16). Useful intermediate results to this end are

$$\mathrm{Ad}_h \mapsto \mathrm{Ad}_\ell \mathrm{Ad}_h \mathrm{Ad}_r \ , \quad \mathrm{Ad}_{\tilde{h}} \mapsto \mathrm{Ad}_\ell \mathrm{Ad}_{\tilde{h}} \mathrm{Ad}_r \ , \quad U_\pm \mapsto \mathrm{Ad}_r^{-1} U_\pm \mathrm{Ad}_r \ ,$$
$$b \mapsto \mathrm{Ad}_r^{-1}(b + \langle \alpha\beta \rangle^{-1} \mathrm{Ad}_h^{-1} \mu^A \alpha^{A'} \ell^{-1} \partial_{AA'} \ell) \ , \quad \hat{b} \mapsto \mathrm{Ad}_r^{-1}(\hat{b} + \langle \alpha\beta \rangle^{-1} \mathrm{Ad}_h^{-1} \hat{\mu}^A \alpha^{A'} \ell^{-1} \partial_{AA'} \ell) \ , \tag{3.23}$$

in which the constraints (2.13) and (2.14) have been used.

We can also derive the Noether currents corresponding to these residual semi-local symmetries directly from $\mathrm{hCS}_6$. The variation of the action under an infinitesimal gauge transformation $\delta\mathcal{A} = \bar{\partial}\hat{\epsilon} + [\mathcal{A}, \hat{\epsilon}]$ is

$$\delta S_{\mathrm{6dCS}} = \frac{1}{2\pi \mathrm{i}} \int_{\mathbb{PT}} \bar{\partial}\Omega \wedge \mathrm{Tr}(\mathcal{A} \wedge \bar{\partial}\hat{\epsilon}) \ . \tag{3.24}$$

First let us consider a transformation that descends to the $\ell$-symmetry, i.e. one for which

$$\hat{\epsilon}|_\alpha = \hat{\epsilon}|_{\tilde{\alpha}} = 0 \ , \quad \hat{\epsilon}|_\beta = \epsilon^{(\ell)} \ .$$

The only contribution to the variation localises to $\beta$ and is given by

$$\delta_\ell S_{\mathrm{6dCS}} \propto \int_{\mathbb{E}^4} \mathrm{vol}_4 \, \partial_0 \left( \frac{1}{\langle \pi\alpha \rangle \langle \pi\tilde{\alpha} \rangle} \, \varepsilon^{AB} \mathrm{Tr}(\mathcal{A}_A \bar{\partial}_B \hat{\epsilon}) \right) \Big|_\beta \ . \tag{3.25}$$

---

[12]Note that to derive this we use that $\hat{\mu} = -\mu$ following the "quaternionic conjugation" defined in appendix A.1.

Since $\mathcal{A}_A|_\beta \propto \langle \pi\beta \rangle$ (recall that we fix $\hat{h}|_\beta = \mathrm{id}$) the only way the integrand will be non-vanishing is for the $\partial_0$ operator to act on $\mathcal{A}_A$. Noting that $\partial_0 \langle \pi\beta \rangle|_\beta = 1$ we have that $\partial_0 \mathcal{A}_A|_\beta = B_A$, and hence the variation becomes

$$\delta_\ell S_{6\mathrm{dCS}} \propto \int_{\mathbb{E}^4} \mathrm{vol}_4 \, \mathrm{Tr}\Big( B^B \beta^{B'} \partial_{BB'} \epsilon^{(\ell)} \Big) \, . \tag{3.26}$$

If we think of the $\ell$-symmetry as a global transformation, then this would provide the conservation law associated to the Noether current, i.e.

$$\beta^{B'} \partial_{BB'} B^B = 0 \, , \tag{3.27}$$

and indeed we will see later that this conservation law follows from the equations of motion of the $\mathrm{IFT}_4$. As the parameter $\epsilon^{(\ell)}$ is allowed to be holomorphic with respect to the complex structure defined by $\beta$, the interpretation is more akin to a Kac-Moody current.

For the case corresponding to the $r$-symmetry we have

$$\hat{\epsilon}|_\alpha = \hat{\epsilon}|_{\tilde{\alpha}} = \epsilon^{(r)} \, , \quad \hat{\epsilon}|_\beta = 0 \, .$$

In this case the variation receives two contributions with an opposite sign

$$\delta_r S_{6\mathrm{dCS}} \propto \int_{\mathbb{E}^4} \mathrm{vol}_4 \, \varepsilon^{AB} \, \mathrm{Tr}\bigg( \frac{1}{\langle \alpha\beta \rangle^2} \mathcal{A}_A \bar{\partial}_B \hat{\epsilon}|_\alpha - \frac{1}{\langle \tilde{\alpha}\beta \rangle^2} \mathcal{A}_A \bar{\partial}_B \hat{\epsilon}|_{\tilde{\alpha}} \bigg) \, . \tag{3.28}$$

Integrating by parts gives

$$\delta_r S_{6\mathrm{dCS}} \propto \int_{\mathbb{E}^4} \mathrm{vol}_4 \, \mathrm{Tr}\bigg( \epsilon^{(r)} \bigg( \frac{\alpha^{A'}}{\langle \alpha\beta \rangle^2} \partial_{AA'} \mathcal{A}^A|_\alpha - \frac{\tilde{\alpha}^{A'}}{\langle \tilde{\alpha}\beta \rangle^2} \partial_{AA'} \mathcal{A}^A|_{\tilde{\alpha}} \bigg) \bigg) \, . \tag{3.29}$$

Introducing new currents defined by

$$\langle \alpha\beta \rangle \, C_A = \mathcal{A}_A|_\alpha \, , \quad \langle \tilde{\alpha}\beta \rangle \, \widetilde{C}_A = \mathcal{A}_A|_{\tilde{\alpha}} \, , \tag{3.30}$$

the conservation law associated to the $r$-symmetry is given by

$$\frac{1}{\langle \alpha\beta \rangle} \alpha^{A'} \partial_{AA'} C^A - \frac{1}{\langle \tilde{\alpha}\beta \rangle} \tilde{\alpha}^{A'} \partial_{AA'} \widetilde{C}^A = 0 \, . \tag{3.31}$$

Recalling from eq. (3.9) that $\mathcal{A}_A = \langle \pi\beta \rangle \mathrm{Ad}_{\hat{h}}^{-1} B_A + \pi^{A'} \hat{h}^{-1} \partial_{AA'} \hat{h}$, we can relate the $B$ current to the $C$ and $\widetilde{C}$ currents as follows

$$C_A = \mathrm{Ad}_h^{-1} B_A + \frac{1}{\langle \alpha\beta \rangle} \alpha^{A'} h^{-1} \partial_{AA'} h = (\widehat{b} - \widehat{\jmath}) \mu_A - (b - j) \hat{\mu}_A \, , \tag{3.32}$$

$$\widetilde{C}_A = \mathrm{Ad}_{\tilde{h}}^{-1} B_A + \frac{1}{\langle \tilde{\alpha}\beta \rangle} \tilde{\alpha}^{A'} \tilde{h}^{-1} \partial_{AA'} \tilde{h} = \sigma(\widehat{b} - \widehat{\jmath}) \mu_A - \sigma^{-1}(b - j) \hat{\mu}_A \, , \tag{3.33}$$

where we have used the identities (3.15). The transformation of these currents under the $(\ell, r)$-symmetries is given by

$$
\begin{aligned}
(\ell, r): \quad & B_A \mapsto \mathrm{Ad}_\ell B_A - \langle \alpha\beta \rangle^{-1} \alpha^{A'} \partial_{AA'} \ell \ell^{-1} \, , \\
(\ell, r): \quad & C_A \mapsto \mathrm{Ad}_r^{-1} C_A + \langle \alpha\beta \rangle^{-1} \alpha^{A'} r^{-1} \partial_{AA'} r \, , \\
(\ell, r): \quad & \widetilde{C}_A \mapsto \mathrm{Ad}_r^{-1} \widetilde{C}_A + \langle \tilde{\alpha}\beta \rangle^{-1} \tilde{\alpha}^{A'} r^{-1} \partial_{AA'} r \, .
\end{aligned}
\tag{3.34}
$$

As a consequence notice that the 4d ($\mathbb{CP}^1$-independent) gauge field introduced above, $A_{AA'} = \beta_{A'} B_A$, is invariant under the right action, whereas the left action acts as a conventional gauge transformation

$$(\ell, r): \quad A_{AA'} \mapsto \mathrm{Ad}_\ell A_{AA'} - \partial_{AA'} \ell \ell^{-1} \, , \tag{3.35}$$

albeit semi-local rather than fully local since $\ell$ is constrained as in eq. (2.13). The transformation of these currents and the 4d ASD connection also follows from the $\mathrm{hCS}_6$ description. While the original

gauge transformations act on $\mathcal{A}$, we observe that $r$ and $\tilde{r}$ become right-actions on $\hat{h}$, leaving $\mathcal{A}'$ and $A_{AA'}$ invariant. By comparison, after fixing $\hat{h}|_\beta = \text{id}$, it is only a combination of the 'internal' transformations and the original gauge transformations that preserve this constraint. In particular, $\ell$ has an induced action on $h$, $\tilde{h}$ and $\mathcal{A}'$, thus leading to the above transformations of $B_A$ and $A_{AA'}$.

As we will show momentarily, the equations of motion of the theory correspond to anti-self duality of the field strength of the connection $A_{AA'}$, hence it immediately follows that the equations of motion are preserved by the symmetry transformations (3.35). To close the section we note that the action is concisely given in terms of the currents as

$$S_{\text{IFT}_4} = \frac{K}{\langle\alpha\tilde{\alpha}\rangle} \int_{\mathbb{E}^4} \text{vol}_4 \, \epsilon^{AB} \text{Tr}\big(\text{Ad}_h^{-1} B_A (C_B - \Lambda^T \widetilde{C}_B)\big) + S_{\text{WZ}_4} \, . \tag{3.36}$$

## 3.2   Equations of Motion, 4d ASDYM and Lax Formulation

The equations of motion of the IFT$_4$ can be obtained in a brute force fashion by performing a variation of the action (3.16). This calculation requires the variation of the operators $U_\pm$

$$\delta U_\pm(X) = U_\pm(\delta X) + U_\pm\left([\tilde{h}^{-1}\delta\tilde{h}, U_\mp^T(X)]\right) - U_\mp^T\left([h^{-1}\delta h, U_\pm(X)]\right) \, , \tag{3.37}$$

but is otherwise straightforward. The outcome is that the equations of motion can be written as

$$
\begin{aligned}
-\frac{\mu^A{}_{\alpha'}^{A'}}{\langle\alpha\beta\rangle} \partial_{AA'}\widehat{b} + \frac{\hat{\mu}^A{}_{\alpha'}^{A'}}{\langle\alpha\beta\rangle} \partial_{AA'}b + [\,\widehat{\jmath}, b\,] - [\,j, \widehat{b}\,] - [\,\widehat{b}, b\,] = 0 \, , \\
-\frac{\mu^A{}_{\tilde{\alpha}'}^{A'}}{\langle\tilde{\alpha}\beta\rangle} \partial_{AA'}\widehat{\tilde{b}} + \frac{\hat{\mu}^A{}_{\tilde{\alpha}'}^{A'}}{\langle\tilde{\alpha}\beta\rangle} \partial_{AA'}\tilde{b} + [\,\widehat{\tilde{\jmath}}, \tilde{b}\,] - [\,\tilde{\jmath}, \widehat{\tilde{b}}\,] - [\,\widehat{\tilde{b}}, \tilde{b}\,] = 0 \, ,
\end{aligned}
\tag{3.38}
$$

in which we invoke the definitions of $b$, $\tilde{b}$, $\widehat{b}$ and $\widehat{\tilde{b}}$ above in eqs. (3.13) and (3.14). These equations can be written in terms of the current $B_A$ as

$$
\begin{aligned}
\alpha^{A'}\partial_{AA'}B^A + \frac{1}{2}\langle\alpha\beta\rangle[B_A, B^A] = 0 \, , \\
\tilde{\alpha}^{A'}\partial_{AA'}B^A + \frac{1}{2}\langle\tilde{\alpha}\beta\rangle[B_A, B^A] = 0 \, .
\end{aligned}
\tag{3.39}
$$

Taking a weighted sum and difference equations gives

$$
\begin{aligned}
(\langle\tilde{\alpha}\beta\rangle\alpha^{A'} - \langle\alpha\beta\rangle\tilde{\alpha}^{A'})\partial_{AA'}B^A = -\langle\alpha\tilde{\alpha}\rangle\beta^{A'}\partial_{AA'}B^A = 0 \, , \\
(\langle\tilde{\alpha}\beta\rangle\alpha^{A'} + \langle\alpha\beta\rangle\tilde{\alpha}^{A'})\partial_{AA'}B^A + \langle\tilde{\alpha}\beta\rangle\langle\alpha\beta\rangle[B_A, B^A] = 0 \, .
\end{aligned}
\tag{3.40}
$$

the first of which is the anticipated conservation equation for the $\ell$-symmetry. Making use of the definitions of $C$ and $\widetilde{C}$ in (3.32), the equations of motion are equivalently expressed as

$$
\begin{aligned}
\alpha^{A'}\partial_{AA'}C^A + \frac{1}{2}\langle\alpha\beta\rangle[C_A, C^A] = 0 \, , \\
\tilde{\alpha}^{A'}\partial_{AA'}\widetilde{C}^A + \frac{1}{2}\langle\tilde{\alpha}\beta\rangle[\widetilde{C}_A, \widetilde{C}^A] = 0 \, .
\end{aligned}
\tag{3.41}
$$

Noting that $[C_A, C^A] = [\widetilde{C}_A, \widetilde{C}^A]$ we can take the difference of these equations to obtain

$$\frac{1}{\langle\alpha\beta\rangle} \alpha^{A'}\partial_{AA'}C^A - \frac{1}{\langle\tilde{\alpha}\beta\rangle} \tilde{\alpha}^{A'}\partial_{AA'}\widetilde{C}^A = 0 \, , \tag{3.42}$$

which is the anticipated conservation law for the $r$-symmetry.

**ASDYM.** We will now justify the claim that this theory is integrable by constructing explicit Lax pair formulations of the dynamics in two different fashions. First we will show the equations of motion (3.39) can be recast as the anti-self-dual equation for a Yang-Mills connection. Before demonstrating that this holds for our particular model, let us highlight that this follows from the construction of $hCS_6$ by briefly reviewing the Penrose-Ward correspondence [23]. Recalling that we have resolved one of the $hCS_6$ equations of motion $\mathcal{F}'_{0A} = 0$ to find $\mathcal{A}'_A = \pi^{A'} A_{AA'}$, it follows that the remaining system of equations should be equivalent to the vanishing of the other components of the field strength $\mathcal{F}'_{AB} = 0$. We may express this in terms of the anti-holomorphic covariant derivative $\bar{D}'_A = \bar{\partial}_A + \mathcal{A}'_A$ as $[\bar{D}'_A, \bar{D}'_B] = 0$, which may also be written as

$$\pi^{A'}\pi^{B'}[D_{AA'}, D_{BB'}] = 0 . \tag{3.43}$$

This is equivalent to the vanishing of $\pi^{A'}\pi^{B'}F_{AA'BB'}$ where $F$ is the field strength of the 4d connection $A_{AA'}$. To make contact with the anti-self-dual Yang-Mills equation, note that an arbitrary tensor that is anti-symmetric in Lorentz indices, e.g. $F_{\mu\nu}$, can be expanded in spinor indices as

$$F_{AA'BB'} = \varepsilon_{AB}\,\phi_{A'B'} + \varepsilon_{A'B'}\,\tilde{\phi}_{AB} . \tag{3.44}$$

Here, $\phi$ and $\tilde{\phi}$ are both symmetric and correspond to the self-dual and anti-self-dual components of the field strength respectively. Explicitly computing the contraction (3.43), we find that the remaining equation is simply $\phi = 0$ which is indeed the anti-self-dual Yang-Mills equation. In effect, this argument demonstrates that a holomorphic gauge field on twistor space (which is gauge-trivial in $\mathbb{CP}^1$) is in bijection with a solution to the 4-dimensional anti-self-dual Yang-Mills equation – this statement is the Penrose-Ward correspondence.

Now, returning to the case at hand, recall that our connection is of the form $A_{AA'} = \beta_{A'}B_A$, so the anti-self-dual Yang-Mills equation specialises to

$$\langle\pi\beta\rangle\big(\pi^{A'}\partial_{AA'}B_B - \pi^{B'}\partial_{BB'}B_A + \langle\pi\beta\rangle[B_A, B_B]\big) = 0 . \tag{3.45}$$

This should hold for any $\pi^{A'} \in \mathbb{CP}^1$ and we can extract the key information by expanding $\pi^{A'}$ in the basis formed by $\alpha^{A'}$ and $\tilde{\alpha}^{A'}$, that is

$$\pi^{A'} = \frac{1}{\langle\alpha\tilde{\alpha}\rangle}\left(\langle\pi\tilde{\alpha}\rangle\alpha^{A'} - \langle\pi\alpha\rangle\tilde{\alpha}^{A'}\right) . \tag{3.46}$$

Substituting into (3.45), we find two independent equations with $\mathbb{CP}^1$-dependent coefficients $\langle\pi\beta\rangle\langle\pi\tilde{\alpha}\rangle$ and $\langle\pi\beta\rangle\langle\pi\alpha\rangle$ respectively. These are explicitly given by the two equations in eq. (3.39). Therefore, as expected, the equations of motion for this $IFT_4$ are equivalent to the anti-self-dual Yang-Mills equation for $A_{AA'} = \beta_{A'}B_A$.

Let us comment on the relation to Ward's conjecture which postulates that many[13] integrable models arise as reductions of the ASDYM equations. It is clear that that the equations of motion for the $\lambda$-deformations explored in this paper arise as symmetry reductions of the ASDYM equations for the explicit form of the connection given above. On the other hand, a generic ASDYM connection can be partially gauge-fixed such that the remaining degrees of freedom are completely captured by the so-called Yang's matrix, which is the fundamental field of the $WZW_4$ model. In this case, the equations of motion of the $WZW_4$ model, known as Yang's equations, are the remaining ASDYM equations. It is natural to ask whether a generic ASDYM connection can also be partially gauge-fixed to take the explicit form found in this paper. This would provide a 1-to-1 correspondence between solutions of the ASDYM equations and solutions to our 4d IFT.

---

[13]The original conjecture [21] states that "many (and perhaps all?)" integrable models arise in this manner. However, a notable absentee of this proposal is the Kadomtsev-Petviashvilii (KP) equation, see [24] for a discussion.

**B-Lax.** The anti-self-dual Yang-Mills equation is also 'integrable' in the sense that it admits a Lax formalism. Using the basis of spinors $\mu^A$ and $\hat{\mu}^A$, we define the Lax pair $L$ and $M$ by

$$L^{(B)} = \langle \pi \hat{\gamma} \rangle^{-1} \hat{\mu}^A \pi^{A'} D_{AA'} \,, \qquad M^{(B)} = \langle \pi \gamma \rangle^{-1} \mu^A \pi^{A'} D_{AA'} \,, \tag{3.47}$$

where the normalisations are for later convenience.[14] It is important to emphasise that here $\pi^{A'}$ is not just an *ad hoc* spectral parameter. It is introduced directly as a result of the hCS$_6$ equations of motion and is the coordinate on $\mathbb{CP}^1 \hookrightarrow \mathbb{PT}$. The vanishing of $[L^{(B)}, M^{(B)}] = 0$ for any $\pi^{A'} \in \mathbb{CP}^1$ is equivalent to the anti-self-dual Yang-Mills equation.

**C-Lax.** Let us now turn to the equations of motion cast in terms of the $C$-currents in eq. (3.41). Evidently, looking at the definition of these currents eq. (3.32), we see that when $\sigma = 1$ we have $\widetilde{C} = C$ and the equations of motion have the same form as the $B$-current equations (3.39). Therefore, when $\sigma = 1$, we can package the $C$-equations in terms of a ASDYM connection $A_{AA'}^{(C)} = \beta_{A'} C_A$. Away from this point, when $\widetilde{C} \neq C$ it is not immediately evident if these equations follow from an ASDYM connection. Regardless, we can still give these equations a Lax pair presentation as follows.

Letting $\varrho \in \mathbb{C}$ denote a spectral parameter we define

$$\begin{aligned} L^{(C)} &= \frac{1}{n_L} \hat{\mu}^A \left( \frac{\alpha^{A'}}{\langle \alpha \beta \rangle} (1 + \varrho) + \frac{\sigma^{-1} \tilde{\alpha}^{A'}}{\langle \tilde{\alpha} \beta \rangle} (1 - \varrho) \right) \partial_{AA'} + \frac{1}{n_L} \hat{\mu}^A C_A \,, \\ M^{(C)} &= \frac{1}{n_M} \mu^A \left( \frac{\alpha^{A'}}{\langle \alpha \beta \rangle} (1 + \varrho) + \frac{\sigma \tilde{\alpha}^{A'}}{\langle \tilde{\alpha} \beta \rangle} (1 - \varrho) \right) \partial_{AA'} + \frac{1}{n_M} \mu^A C_A \,. \end{aligned} \tag{3.48}$$

Noting that $\mu^B \widetilde{C}_B = \sigma^{-1} \mu^B C_B$ and $\hat{\mu}^B \widetilde{C}_B = \sigma \hat{\mu}^B C_B$ one immediately sees that the terms inside $[L^{(C)}, M^{(C)}]$ linear in $\varrho$ yield the conservation law eq. (3.42). The contributions independent of $\varrho$, combined with eq. (3.42), give either of eq. (3.41). It will be convenient to fix the overall normalisation of these Lax operators to be

$$n_L = \frac{\langle \alpha \hat{\gamma} \rangle}{\langle \alpha \beta \rangle} (1 + \varrho) + \frac{\langle \tilde{\alpha} \hat{\gamma} \rangle}{\langle \tilde{\alpha} \beta \rangle} \sigma^{-1} (1 - \varrho) \,, \qquad n_M = \frac{\langle \alpha \gamma \rangle}{\langle \alpha \beta \rangle} (1 + \varrho) + \frac{\langle \tilde{\alpha} \gamma \rangle}{\langle \tilde{\alpha} \beta \rangle} \sigma (1 - \varrho) \,. \tag{3.49}$$

Unlike the spectral parameter $\pi_{A'}$ entering the $B$-Lax, there is no clear way to associate the spectral parameter of the $C$-Lax, $\varrho$, with the $\mathbb{CP}^1$ coordinate alone. Indeed, under a natural assumption, we will see that $\varrho$ has origins from both the $\mathbb{CP}^1$ geometry *and* the parameters that enter the boundary conditions.

The existence of a second Lax formulation of the theory, distinct from the ASDYM equations encoded via the $B$-Lax, is an unexpected feature. We will see shortly that, upon symmetry reduction, this twin Lax formulation is inherited by the IFT$_2$.

## 3.3 Reality Conditions and Parameters

To understand how the reality of the action of the IFT$_4$ (3.16) with (3.17), as well as the dependence on the parameters $K$, $\sigma$, $\alpha_{A'}$, $\tilde{\alpha}_{A'}$, $\beta_{A'}$, $\mu^A$ and $\hat{\mu}^A$, let us denote our coordinates

$$\begin{aligned} \mathsf{w} &= \frac{\langle \alpha \beta \rangle}{\langle \alpha \tilde{\alpha} \rangle [\mu \hat{\mu}]} \hat{\mu}_A \tilde{\alpha}_{A'} x^{AA'} \,, & \hat{\mathsf{w}} &= -\frac{\langle \alpha \beta \rangle}{\langle \alpha \tilde{\alpha} \rangle [\mu \hat{\mu}]} \mu_A \tilde{\alpha}_{A'} x^{AA'} \,, \\ \mathsf{z} &= -\frac{\langle \tilde{\alpha} \beta \rangle}{\langle \alpha \tilde{\alpha} \rangle [\mu \hat{\mu}]} \hat{\mu}_A \alpha_{A'} x^{AA'} \,, & \hat{\mathsf{z}} &= \frac{\langle \tilde{\alpha} \beta \rangle}{\langle \alpha \tilde{\alpha} \rangle [\mu \hat{\mu}]} \mu_A \alpha_{A'} x^{AA'} \,, \end{aligned} \tag{3.50}$$

such that

$$j = h^{-1} \partial_{\mathsf{w}} h \,, \qquad \hat{j} = h^{-1} \partial_{\hat{\mathsf{w}}} h \,, \qquad \tilde{j} = \tilde{h}^{-1} \partial_{\mathsf{z}} \tilde{h} \,, \qquad \widehat{\tilde{j}} = \tilde{h}^{-1} \partial_{\hat{\mathsf{z}}} \tilde{h} \,. \tag{3.51}$$

---

[14]The constant spinors $\gamma$ and $\hat{\gamma}$ appear in the symmetry reduction and will be introduced in § 4.

In this subsection we let $\mu^A$ and $\hat{\mu}^A$ be an unconstrained basis of spinors, i.e., not related by Euclidean conjugation or of fixed norm. This means the action (3.16) with (3.17) comes with an extra factor of $[\mu\hat{\mu}]^{-1}$. Writing the volume element $\mathrm{vol}_4 = \frac{1}{12}\varepsilon_{AB}\varepsilon_{CD}\varepsilon_{A'C'}\varepsilon_{B'D'}\,dx^{AA'}\wedge dx^{BB'}\wedge dx^{CC'}\wedge dx^{DD'}$ in terms of the coordinates $\{\mathsf{w},\hat{\mathsf{w}},\mathsf{z},\hat{\mathsf{z}}\}$ we find

$$\mathrm{vol}_4 = \frac{\langle\alpha\tilde{\alpha}\rangle^2[\mu\hat{\mu}]^2}{\langle\alpha\beta\rangle^2\langle\tilde{\alpha}\beta\rangle^2}\,d\mathsf{w}\wedge d\hat{\mathsf{w}}\wedge d\mathsf{z}\wedge d\hat{\mathsf{z}} = \frac{\langle\alpha\tilde{\alpha}\rangle^2[\mu\hat{\mu}]^2}{\langle\alpha\beta\rangle^2\langle\tilde{\alpha}\beta\rangle^2}\,\mathrm{vol}_4' \ . \tag{3.52}$$

Substituting into the action (3.16) with (3.17), we see that the IFT$_4$ now only depends explicitly on two parameters

$$K' = \frac{\langle\alpha\tilde{\alpha}\rangle[\mu\hat{\mu}]}{\langle\alpha\beta\rangle^2\langle\tilde{\alpha}\beta\rangle^2}\,K \quad\text{and}\quad \sigma \ . \tag{3.53}$$

Moreover, the action is invariant under the following $GL(1;\mathbb{C})$ space-time symmetry

$$\mathsf{z}\to e^{\vartheta}\mathsf{z}\ , \qquad \mathsf{w}\to e^{\vartheta}\mathsf{w}\ , \qquad \hat{\mathsf{z}}\to e^{-\vartheta}\hat{\mathsf{z}}\ , \qquad \hat{\mathsf{w}}\to e^{-\vartheta}\hat{\mathsf{w}}\ , \tag{3.54}$$

where $\vartheta\in\mathbb{C}$.

To find a real action we should impose reality conditions on the coordinates $\{\mathsf{w},\hat{\mathsf{w}},\mathsf{z},\hat{\mathsf{z}}\}$, the fields $h$ and $\tilde{h}$, and the parameters $K'$ and $\sigma$. We start by observing four sets of admissible reality conditions simply found by inspection of the 4d IFT. Note that, implicitly, we will not assume Euclidean reality conditions for $x^{AA'}$. Starting from Euclidean reality conditions we complexify and take different split signature real slices. We will then turn to the hCS$_6$ origin of the different reality conditions.

Introducing $\Theta$, the lift of an antilinear involutive automorphism $\theta$ of the Lie algebra $\mathfrak{g}$ to the group $G$, the four sets of reality conditions are as follows:

1. In the first case, the coordinates are all real, $\bar{\mathsf{w}}=\mathsf{w}$, $\bar{\hat{\mathsf{w}}}=\hat{\mathsf{w}}$, $\bar{\mathsf{z}}=\mathsf{z}$, $\bar{\hat{\mathsf{z}}}=\hat{\mathsf{z}}$; $K'$ and $\sigma$ are real; and the group-valued fields are elements of the real form, $\Theta(h)=h$ and $\Theta(\tilde{h})=\tilde{h}$. Under conjugation we have $U_\pm\to U_\pm$.

2. In the second case, the coordinates conjugate as $\bar{\mathsf{w}}=\hat{\mathsf{w}}$, $\bar{\mathsf{z}}=\hat{\mathsf{z}}$; $K'$ is imaginary and $\sigma$ is a phase factor; and the group-valued fields are elements of the real form, $\Theta(h)=h$ and $\Theta(\tilde{h})=\tilde{h}$. Under conjugation we have $U_\pm\to U_\mp$.

3. In the third case, the coordinates conjugate as $\bar{\mathsf{w}}=\hat{\mathsf{z}}$, $\bar{\mathsf{z}}=\hat{\mathsf{w}}$; $K'$ and $\sigma$ are real; and the group-valued fields are related by conjugation $\Theta(h)=\tilde{h}$. Under conjugation we have $U_\pm\to U_\pm^T$.

4. In the final case, the coordinates conjugate as $\bar{\mathsf{w}}=\mathsf{z}$, $\bar{\hat{\mathsf{w}}}=\hat{\mathsf{z}}$; $K'$ is imaginary and $\sigma$ is a phase factor; and the group-valued fields are related by conjugation $\Theta(h)=\tilde{h}$. Under conjugation we have $U_\pm\to U_\mp^T$.

The action (3.16) with (3.17) is real for each of these sets of reality conditions. To determine the signature for each set of reality conditions, we note that[15]

$$\varepsilon_{AB}\varepsilon_{A'B'}dx^{AA'}dx^{BB'} = \frac{2\langle\alpha\tilde{\alpha}\rangle[\mu\hat{\mu}]}{\langle\alpha\beta\rangle\langle\tilde{\alpha}\beta\rangle}\left(d\mathsf{w}d\hat{\mathsf{z}} - d\mathsf{z}d\hat{\mathsf{w}}\right)\ , \tag{3.55}$$

---

[15]Conjugating in Euclidean signature we find the reality conditions

$$\bar{\mathsf{w}} = \frac{\langle\hat{\alpha}\hat{\beta}\rangle}{\langle\hat{\alpha}\hat{\alpha}\rangle}\left(\frac{\langle\alpha\hat{\alpha}\rangle}{\langle\alpha\beta\rangle}\hat{\mathsf{w}} + \frac{\langle\tilde{\alpha}\hat{\alpha}\rangle}{\langle\tilde{\alpha}\beta\rangle}\hat{\mathsf{z}}\right)\ , \qquad \bar{\mathsf{z}} = \frac{\langle\hat{\tilde{\alpha}}\hat{\beta}\rangle}{\langle\hat{\alpha}\hat{\alpha}\rangle}\left(\frac{\langle\tilde{\alpha}\hat{\alpha}\rangle}{\langle\tilde{\alpha}\beta\rangle}\hat{\mathsf{z}} - \frac{\langle\alpha\hat{\alpha}\rangle}{\langle\alpha\beta\rangle}\hat{\mathsf{w}}\right)\ .$$

As an example, let us take $\hat{\alpha}=\tilde{\alpha}$, in which case the reality conditions simplify to $\bar{\mathsf{w}}=\frac{\langle\hat{\alpha}\hat{\beta}\rangle}{\langle\tilde{\alpha}\beta\rangle}\hat{\mathsf{z}}$ and $\bar{\mathsf{z}}=\frac{\langle\alpha\hat{\beta}\rangle}{\langle\alpha\beta\rangle}\hat{\mathsf{w}}$. Substituting into the metric we find $\frac{2\langle\alpha\hat{\alpha}\rangle[\mu\hat{\mu}]}{\langle\alpha\beta\rangle\langle\tilde{\alpha}\beta\rangle}\left(d\mathsf{w}d\bar{\mathsf{w}} + \psi\bar{\psi}d\mathsf{z}d\bar{\mathsf{z}}\right)$ where $\psi = \frac{\langle\alpha\beta\rangle}{\langle\tilde{\alpha}\beta\rangle}$. Since the prefactor is real and positive, this indeed has Euclidean signature. Note that these reality conditions are distinct from case 3 above, and they do not imply reality of the 4d IFT.

It is then straightforward to see that the four sets of reality conditions above all correspond to split signature. Note that for the metric to be real, we require the prefactor to be real in cases 1 and 4 and imaginary in cases 2 and 3. We will see that this is indeed the case when we comment on the $\text{hCS}_6$ origin.

In cases 1 and 4 the reality conditions are preserved by an $SO(1,1)$ space-time symmetry (3.54) with $\vartheta \in \mathbb{R}$. On the other hand, in cases 2 and 3, the reality conditions are preserved by an $SO(2)$ space-time symmetry with $|\vartheta| = 1$. In § 5, we will be interested in symmetry reducing while preserving the space-time symmetry, recovering an action on $\mathbb{R}^2$ or $\mathbb{R}^{1,1}$ that is invariant under the Euclidean or Poincaré groups respectively. We have freedom in how we do this since the action is not invariant under $SO(2)$ rotations acting on $(\mathsf{z}, \mathsf{w})$ and $(\hat{\mathsf{z}}, \hat{\mathsf{w}})$. Therefore, we can choose symmetry reduce along different directions in each of these planes, in principle introducing an additional two parameters. We should note that in the Euclidean case, since the two planes are related by conjugation, we will break the reality properties of the action unless the two symmetry reduction directions are also related by conjugation, reducing the number of parameters by one for a real action. This is not an issue in the Lorentzian case since the coordinates are real, hence we expect to find a four-parameter real Lorentz-invariant $\text{IFT}_2$. We will indeed see that this is the case in § 5.

**Origin of reality conditions from $\text{hCS}_6$.** Let us now briefly describe the origin of the different sets of reality conditions from 6 dimensions. It is shown in [11] that for the $\text{hCS}_6$ action to be real we require that

$$\overline{C(\Phi)} = C(\Phi) \ , \tag{3.56}$$

where $\Phi$ is defined in eq. (2.2) and $C$ is a conjugation that acts on the coordinates $(x, \pi)$, not on the fixed spinors $\alpha$, $\tilde{\alpha}$ and $\beta$.[16] In Euclidean signature this constraint becomes

$$\frac{\bar{K}}{\langle \pi \hat{\alpha} \rangle \langle \pi \hat{\tilde{\alpha}} \rangle \langle \pi \hat{\beta} \rangle^2} = \frac{K}{\langle \pi \alpha \rangle \langle \pi \tilde{\alpha} \rangle \langle \pi \beta \rangle^2} \ . \tag{3.57}$$

We immediately see that this has no solutions since the double pole at $\beta$ is mapped to $\hat{\beta}$ and $\hat{\beta} = \beta$ has no solutions. On the other hand, in split signature we have

$$\frac{\bar{K}}{\langle \pi \bar{\alpha} \rangle \langle \pi \bar{\tilde{\alpha}} \rangle \langle \pi \bar{\beta} \rangle^2} = \frac{K}{\langle \pi \alpha \rangle \langle \pi \tilde{\alpha} \rangle \langle \pi \beta \rangle^2} \ . \tag{3.58}$$

This can be solved by taking $K$ and $\beta$ to be real, and $\alpha$ and $\tilde{\alpha}$ to either both be real or to form a complex conjugate pair.

We also need to ask that the boundary conditions (2.8) are compatible with the reality conditions. Imposing $C^*(\mathcal{A}_A) = \theta(\mathcal{A}_A)$, we can either take $\mu$ and $\hat{\mu}$ to either both be real or to form a complex conjugate pair. The two choices of reality conditions for $(\alpha, \tilde{\alpha})$ and the two for $(\mu, \hat{\mu})$ give a total of four sets of reality conditions, which we anticipate will recover those in the list presented above. With the same ordering, we have the following:

1. In the first case, we take real $(\alpha, \tilde{\alpha})$ and real $(\mu, \hat{\mu})$. Analysing the boundary conditions we find that $\mathcal{A}_A$ is valued in the real form at the poles, implying that $h$ and $\tilde{h}$ are as well, and that $\sigma$ is real. Since both $\langle \alpha \tilde{\alpha} \rangle$ and $[\mu \hat{\mu}]$ are real, real $K$ implies that $K'$ is real using eq. (3.53).

---

[16]Conjugation in Euclidean signature can be defined as $C(\mu_A) = \hat{\mu}_A = \epsilon_A{}^B \bar{\mu}_B$, $C(\gamma'_A) = \hat{\gamma}_{A'} = \varepsilon_{A'}{}^{B'} \bar{\gamma}_{B'}$ and $C(x^{AA'}) = (\epsilon^T)^A{}_B \bar{x}^{BB'} \varepsilon_{B'}{}^{A'}$ with $\varepsilon_1{}^2 = -1$, while in split signature, we take $C(\mu_A) = \bar{\mu}_A$, $C(\gamma_{A'}) = \bar{\gamma}_{A'}$ and $C(x^{AA'}) = \bar{x}^{AA'}$. We will restrict our attention to Euclidean and split signatures since there are no ASD connections in Lorentzian signature [11].

2. In the second case, we take real $(\alpha, \tilde{\alpha})$ and complex conjugate $(\mu, \hat{\mu})$. Analysing the boundary conditions we find that $\mathcal{A}_A$ is valued in the real form at the poles, implying that $h$ and $\tilde{h}$ are as well, and that $\sigma$ is a phase factor. Since $\langle \alpha \tilde{\alpha} \rangle$ is real and $[\mu \hat{\mu}]$ is imaginary, real $K$ implies that $K'$ is imaginary using eq. (3.53).

3. In the third case, we take complex conjugate $(\alpha, \tilde{\alpha})$ and complex conjugate $(\mu, \hat{\mu})$. Analysing the boundary conditions we find that $\mathcal{A}_A$ at $\alpha$ is the conjugate of $\mathcal{A}_A$ at $\tilde{\alpha}$, implying that $h$ and $\tilde{h}$ are also conjugate, and that $\sigma$ is real. Since both $\langle \alpha \tilde{\alpha} \rangle$ and $[\mu \hat{\mu}]$ are imaginary, real $K$ implies that $K'$ is real using eq. (3.53).

4. In the final case, we take complex conjugate $(\alpha, \tilde{\alpha})$ and real $(\mu, \hat{\mu})$. Analysing the boundary conditions we find that $\mathcal{A}_A$ at $\alpha$ is the conjugate of $\mathcal{A}$ at $\tilde{\alpha}$, implying that $h$ and $\tilde{h}$ are also conjugate, and that $\sigma$ is a phase factor. Since $\langle \alpha \tilde{\alpha} \rangle$ is imaginary and $[\mu \hat{\mu}]$ is real, real $K$ implies that $K'$ is imaginary using eq. (3.53).

Finally, one can also check that in split signature, the different reality conditions for $(\alpha, \tilde{\alpha})$ and $(\mu, \hat{\mu})$ imply the different reality conditions for the coordinates $\{\mathsf{w}, \hat{\mathsf{w}}, \mathsf{z}, \hat{\mathsf{z}}\}$ given above.

As implied above, see also [11], a real action in split signature in 4 dimensions is useful for symmetry reducing and constructing real 2d IFTs since both Euclidean and Lorentzian signature in 2 dimensions can be accessed. However, the lack of a real action in Euclidean signature raises questions about the quantisation of the IFT$_4$ itself. We will briefly return to the issue of quantisation in § 7.

## 3.4 Equivalent Forms of the Action and its Limits

In this section we describe alternative, but equivalent ways of writing the action of the 4d IFT (3.16) with (3.17), and consider two interesting limits of the theory. These constructions are motivated by analogous ones that are important in the context of the 2d $\lambda$-deformed WZW model.

First, let us note that the IFT$_4$ (3.16) with (3.17) can be written in the following two equivalent forms

$$
\begin{aligned}
S_{\mathrm{IFT}_4} = K' \int_{\mathbb{E}^4} \mathrm{vol}'_4 \, \mathrm{Tr}\big( (j - \sigma \widetilde{\jmath})(U_+^T - U_-)(\widehat{\jmath} + \sigma^{-1}\widehat{\widetilde{\jmath}}) - \sigma \widetilde{\jmath}\,\widehat{\jmath} + \sigma^{-1} j\,\widehat{\widetilde{\jmath}} \big) + S_{\mathrm{WZ}_4} \\
= K' \int_{\mathbb{E}^4} \mathrm{vol}'_4 \, \mathrm{Tr}\big( (\mathrm{Ad}_h j - \mathrm{Ad}_{\tilde{h}}\widetilde{\jmath})(\widetilde{U}_+^T - \widetilde{U}_-)(\mathrm{Ad}_h \widehat{\jmath} - \mathrm{Ad}_{\tilde{h}}\widehat{\widetilde{\jmath}}) + \mathrm{Ad}_{\tilde{h}}\widetilde{\jmath}\,\mathrm{Ad}_h \widehat{\jmath} - \mathrm{Ad}_h j\,\mathrm{Ad}_{\tilde{h}}\widehat{\widetilde{\jmath}} \big) + S_{\mathrm{WZ}_4} \, ,
\end{aligned}
$$
(3.59)

where

$$
\begin{aligned}
U_\pm &= \big(1 - \sigma^{\pm 1}\Lambda\big)^{-1} \, , & \Lambda &= \mathrm{Ad}_{\tilde{h}}^{-1}\mathrm{Ad}_h \, , \\
\widetilde{U}_\pm &= \big(1 - \sigma^{\pm 1}\widetilde{\Lambda}\big)^{-1} \, , & \widetilde{\Lambda} &= \mathrm{Ad}_h \mathrm{Ad}_{\tilde{h}}^{-1} \, .
\end{aligned}
$$
(3.60)

Written in this way, it is straightforward to see that the symmetries of the 4d IFT are given by transformations of the form (3.22) with

$$
(\partial_{\mathsf{w}} - \sigma\partial_{\mathsf{z}})r = (\partial_{\hat{\mathsf{w}}} - \sigma^{-1}\partial_{\hat{\mathsf{z}}})r = 0 \, , \qquad (\partial_{\mathsf{w}} - \partial_{\mathsf{z}})\ell = (\partial_{\hat{\mathsf{w}}} - \partial_{\hat{\mathsf{z}}})\ell = 0 \, ,
$$
(3.61)

which, as expected, coincide with (2.13) and (2.14) upon using the definitions (3.50).

We can also introduce auxiliary fields $B^A$, $C^A$ and $\widetilde{C}^A$ to rewrite the action as

$$
\begin{aligned}
S_{\mathrm{IFT}_4} = K' \int_{\mathrm{E}_4} \mathrm{vol}'_4 \, \mathrm{Tr}\big( & j\,\widehat{\jmath} - 2j\mathrm{Ad}_h^{-1}[B\hat{\mu}] + 2\widehat{\jmath}[C\mu] - 2[C\mu]\mathrm{Ad}_h^{-1}[B\hat{\mu}] \\
& + \widetilde{\jmath}\,\widehat{\widetilde{\jmath}} - 2\widehat{\widetilde{\jmath}}\mathrm{Ad}_{\tilde{h}}^{-1}[B\mu] + 2\widetilde{\jmath}[\widetilde{C}\hat{\mu}] - 2[\widetilde{C}\hat{\mu}]\mathrm{Ad}_{\tilde{h}}^{-1}[B\mu] \\
& + 2[B\mu][B\hat{\mu}] + 2\sigma^{-1}[C\mu][\widetilde{C}\hat{\mu}] \big) + S_{\mathrm{WZ}_4} \, .
\end{aligned}
$$
(3.62)

Here we take the auxiliary fields $B^A$, $C^A$ and $\widetilde{C}^A$ to be undetermined. Varying the action and solving their equations of motion, we find that on-shell, they are given by the expressions introduced above in eqs. (3.13) and (3.33). Moreover, substituting their on-shell values back into (3.62) we recover the 4d IFT. Using the symmetry (2.9), we note that the action can also be written in a similar equivalent form, in which tilded and untilded quantities are swapped, $\sigma \to \sigma^{-1}$, $K' \to -K'$ and $\widetilde{B} = B$. This can also be seen by making the off-shell replacements $[B\mu] \to [B\mu]$, $[B\hat{\mu}] \to \mathrm{Ad}_h([C\hat{\mu}] + \widehat{\jmath})$, $[C\mu] \to \sigma[\widetilde{C}\mu]$ and $[\widetilde{C}\hat{\mu}] \to \mathrm{Ad}_{\tilde{h}}^{-1}[B\hat{\mu}] - \widehat{\widetilde{\jmath}}$, all of which are compatible with the on-shell values of the auxiliary fields.

The first limit we consider is $\sigma \to 0$, in which the action becomes

$$S_{\mathrm{IFT}_4}|_{\sigma \to 0} = \mathring{S}_{\mathrm{IFT}_4} = K' \int_{\mathbb{E}^4} \mathrm{vol}'_4 \, \mathrm{Tr}\big(j\,\widehat{\jmath} + \widetilde{\jmath}\,\widehat{\widetilde{\jmath}} - 2\mathrm{Ad}_h j \, \mathrm{Ad}_{\tilde{h}}\widehat{\widetilde{\jmath}}\big) + S_{\mathrm{WZ}_4} \ . \tag{3.63}$$

This has the form of a current-current coupling between two building blocks that could be described as 'holomorphic WZW$_4$' of the form

$$S_{\mathrm{hWZW4}}[h,\alpha] = \int_{\mathbb{E}^4} \mathrm{vol}'_4 \, \mathrm{Tr}\big(j\,\widehat{\jmath}\big) - \int_{\mathbb{E}^4 \times [0,1]} \mathrm{vol}'_4 \wedge d\rho \, \mathrm{Tr}\big(h^{-1}\partial_\rho h[j,\widehat{\jmath}]\big) \ . \tag{3.64}$$

This somewhat unusual theory has derivatives only in the holomorphic two-plane singled out by the complex structure on $\mathbb{E}^4$ defined by $\alpha$ (i.e. only $\partial_{\mathsf{w}}$ and $\partial_{\hat{\mathsf{w}}}$ enter), although the field depends on all coordinates of $\mathbb{E}^4$. This structure is quite different (both in the kinetic term and Wess-Zumino term) from the conventional WZW$_4$ [25] for which the action[17] is

$$S_{\mathrm{WZW}_4}[h,\alpha,\beta] = \int_{\mathbb{E}^4} \mathrm{Tr}\big(h^{-1}dh \wedge \star h^{-1}dh\big) + \frac{1}{6}\int_{\mathbb{E}^4 \times [0,1]} \varpi_{\alpha,\beta} \wedge \mathrm{Tr}\big((h^{-1}dh)^3\big) \ ,$$
$$\varpi_{\alpha,\beta} = \epsilon_{AB}\alpha_{A'}\beta_{B'}\mathrm{d}x^{AA'} \wedge \mathrm{d}x^{BB'} \ . \tag{3.65}$$

The Kähler point of the theory is achieved when $\beta = \hat{\alpha}$ such that $\varpi_{\alpha,\beta}$ is the Kähler form associated to the complex structure defined by $\alpha$. In fact, the WZ term of our holomorphic WZW$_4$ is of this general form with $\beta = \alpha$ such that $\varpi_{\alpha,\beta}$ defines a $(2,0)$-form. However even in the $\beta = \alpha$ case, the kinetic term does not match that of the holomorphic WZW$_4$ action.

Returning to the holomorphic WZW$_4$, we can establish that the theory is invariant under a rather large set of symmetries. Since only $\mathsf{w}$ and $\hat{\mathsf{w}}$ derivatives enter, it is immediate that the transformation $h \mapsto l(\hat{\mathsf{z}},\mathsf{z})hr(\hat{\mathsf{z}},\mathsf{z})$ leaves the action eq. (3.64) invariant. These are further enhanced, as in a WZW$_2$, to

$$(\ell, r): \quad h \mapsto \ell(\mathsf{z},\hat{\mathsf{z}},\mathsf{w})\,h\,r(\mathsf{z},\hat{\mathsf{z}},\hat{\mathsf{w}}) \ . \tag{3.66}$$

From this perspective holomorphic WZW$_4$ can be considered the embedding of a WZW$_2$ in 4 dimensions. Similarly, we have a symmetry for the holomorphic WZW$_4$ for $\tilde{h}$

$$(\tilde{\ell}, \tilde{r}): \quad \tilde{h} \mapsto \tilde{\ell}(\hat{\mathsf{z}},\hat{\mathsf{w}},\mathsf{w})\,\tilde{h}\,\tilde{r}(\mathsf{z},\hat{\mathsf{w}},\mathsf{w}) \ . \tag{3.67}$$

The interaction term, $\mathrm{Ad}_h j \, \mathrm{Ad}_{\tilde{h}}\widehat{\widetilde{\jmath}}$, in the action (3.63) preserves the right actions, but breaks the enhanced independent $\ell, \tilde{\ell}$ left actions. Instead a new 'diagonal' left action emerges

$$(\ell, r, \tilde{r}): \quad h \mapsto \ell(\mathsf{z}+\mathsf{w},\hat{\mathsf{z}}+\hat{\mathsf{w}})\,h\,r(\mathsf{z},\hat{\mathsf{z}},\hat{\mathsf{w}}) \ , \quad \tilde{h} \mapsto \ell(\mathsf{z}+\mathsf{w},\hat{\mathsf{z}}+\hat{\mathsf{w}})\,\tilde{h}\,\tilde{r}(\mathsf{z},\mathsf{w},\hat{\mathsf{w}}) \ . \tag{3.68}$$

It is important to emphasise that here the right actions on $h$ and $\tilde{h}$ are independent ($r$ and $\tilde{r}$ are not the same). This stems from the enlargement of the residual symmetries of the 6-dimensional boundary

---

[17]This is also the 4d IFT that was found in [11, 14] from hCS$_6$ with two double poles at $\pi = \alpha$ and $\pi = \beta$, with Dirichlet boundary conditions.

conditions. The constraints of eq. (2.14) are relaxed such that gauge parameters at different poles are unrelated but are chiral.

In this limit the currents associated to the left and right action become

$$
\begin{aligned}
B_A|_{\sigma\to 0} &= \mathring{B}_A = \mathrm{Ad}_{\tilde{h}}\widehat{\tilde{\jmath}}\,\mu_A - \mathrm{Ad}_h j\hat{\mu}_A \ , \\
C_A|_{\sigma\to 0} &= \mathring{C}_A = -\big(\hat{\jmath} - \Lambda^{-1}\widehat{\tilde{\jmath}}\big)\mu_A \ , \\
\widetilde{C}_A|_{\sigma\to 0} &= \mathring{\widetilde{C}}_A = \big(\widetilde{\jmath} - \Lambda j\big)\hat{\mu}_A \ .
\end{aligned}
\tag{3.69}
$$

The conservation laws become

$$
\begin{aligned}
\partial_{\mathsf{w}}(\hat{\jmath} - \Lambda^{-1}\widehat{\tilde{\jmath}}) = 0 \ , &\qquad \partial_{\bar{\mathsf{z}}}(\widetilde{\jmath} - \Lambda j) = 0 \ , \\
\partial_{\hat{\mathsf{w}}}(\mathrm{Ad}_h j) - \partial_{\mathsf{w}}(\mathrm{Ad}_{\tilde{h}}\widehat{\tilde{\jmath}}) + &\partial_{\mathsf{z}}(\mathrm{Ad}_{\tilde{h}}\widehat{\tilde{\jmath}}) - \partial_{\bar{\mathsf{z}}}(\mathrm{Ad}_h j) = 0 \ .
\end{aligned}
\tag{3.70}
$$

To compute the $\mathcal{O}(\sigma)$ correction to the action (3.63) we first note that

$$
B_A = \mathring{B}_A + \sigma\left(\mathrm{Ad}_h\mathring{\widetilde{C}}_A + \mathrm{Ad}_{\tilde{h}}\mathring{C}_A\right) + \mathcal{O}(\sigma^2) \ ,
\tag{3.71}
$$

and that the combination $C_A - \Lambda^T\widetilde{C}_A = \mathring{C}_A - \Lambda^T\mathring{\widetilde{C}}_A$ is independent of $\sigma$. Then from the expression of the IFT$_4$ action in terms of currents (3.36), we see that the leading correction to $\mathring{S}_{\mathrm{IFT}_4}$ is given by

$$
2\sigma K'\int_{\mathbb{E}^4}\mathrm{vol}'_4\,\epsilon^{AB}\mathrm{Tr}\big(\mathring{\widetilde{C}}_A\mathring{C}_B\big) = -2\sigma K'\int_{\mathbb{E}^4}\mathrm{vol}_4\,\mathrm{Tr}\big((\widetilde{\jmath} - \Lambda j)(\hat{\jmath} - \Lambda^{-1}\widehat{\tilde{\jmath}})\big) \ ,
\tag{3.72}
$$

i.e. the perturbing operator is the product of two currents associated to the right-acting symmetries.

The second limit we consider is $\sigma \to 1$. Recall that in this limit, we have that $\widetilde{C} = C$ from eqs. (3.32) and (3.33), and a symmetry emerges interchanging $B$ and $C$, as well as $h$ and $\tilde{h}^{-1}$. This is also evident if we set $\sigma = 1$ in (3.62). An alternative way to take $\sigma \to 1$ is to first set $h = \exp(\frac{\nu}{K'})$ and $\tilde{h} = \exp(\frac{\tilde{\nu}}{K'})$, along with $\sigma = e^{\frac{1}{K'}}$ and take $K' \to \infty$. In this limit the 4d IFT becomes

$$
S_{\mathrm{IFT}_4}|_{K'\to\infty} = -\int_{\mathbb{E}^4}\mathrm{vol}'_4\,\mathrm{Tr}\Big((\partial_{\mathsf{w}}\nu - \partial_{\mathsf{z}}\tilde{\nu})\frac{1}{1 - \mathrm{ad}_\nu + \mathrm{ad}_{\tilde{\nu}}}(\partial_{\hat{\mathsf{w}}}\nu - \partial_{\bar{\mathsf{z}}}\tilde{\nu})\Big) \ ,
\tag{3.73}
$$

which is reminiscent of a 4d version of the non-abelian T-dual of the principal chiral model, albeit with two fields instead of one. If instead we take the limit in the action with auxiliary fields (3.62), also setting $[C\mu] = [B\mu] + \mathcal{O}(K'^{-1})$ and $[\widetilde{C}\hat{\mu}] = [B\hat{\mu}] + \mathcal{O}(K'^{-1})$, we find

$$
\begin{aligned}
S_{\mathrm{IFT}_4}|_{K'\to\infty} = \int_{\mathrm{E}_4}\mathrm{vol}'_4\,\mathrm{Tr}\Big(&2\nu\big(\partial_{\mathsf{w}}[B\hat{\mu}] - \partial_{\hat{\mathsf{w}}}[B\mu] + [[B\hat{\mu}],[B\mu]]\big) \\
&+ 2\tilde{\nu}\big(\partial_{\bar{\mathsf{z}}}[B\mu] - \partial_{\mathsf{z}}[B\hat{\mu}] + [[B\mu],[B\hat{\mu}]]\big) - 2[B\mu][B\hat{\mu}]\Big) \ ,
\end{aligned}
\tag{3.74}
$$

after integrating by parts. Integrating out the auxiliary field $B^A$, we recover the action (3.73). It would be interesting to instead integrate out the fields $\nu$ and $\tilde{\nu}$ to give a 4d analogue of 2d non-abelian T-duality. However, note that, unlike in 2 dimensions, $\nu$ and $\tilde{\nu}$ do not enforce the flatness of a 4d connection, hence there is no straightforward way to parametrise the general solution to their equations.

# 4  Symmetry Reduction of hCS$_6$ to CS$_4$

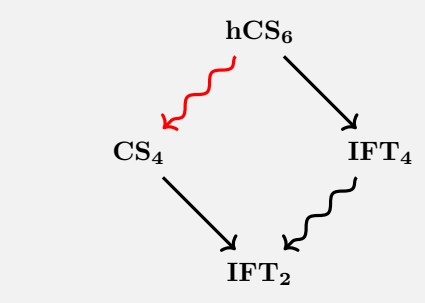

Returning to hCS$_6$, we now descend via the top left-hand side of the diamond by performing a symmetry reduction of the action. Doing so, we find the resulting theory is CS$_4$.

The idea of symmetry reduction is to take a truncation of a $d$-dimensional theory specified by a $d$-form Lagrangian $\mathcal{L}^d$ depending on a set of fields $\{\Phi\}$ to obtain a lower dimensional theory. We assume here that we are reducing along two directions singled out by vector fields $V_i$, $i = 1, 2$. The reduction procedure imposes that all fields are invariant, $L_{V_i}\Phi = 0$, with dynamics now specified by the $d - 2$-form Lagrangian $\mathcal{L}^{d-2} = V_1 \lrcorner V_2 \lrcorner \mathcal{L}^d$. While similar in spirit to a dimensional reduction, there is no requirement that $V_i$ span a compact space, hence there is no scale separation in this truncation.

In order to perform the symmetry reduction, we will introduce a unit norm spinor $\gamma_{A'}$ about which we can expand any spinor $X_{A'}$ as

$$X_{A'} = \langle X\hat{\gamma}\rangle\gamma_{A'} - \langle X\gamma\rangle\hat{\gamma}_{A'} \ . \tag{4.1}$$

The spinor $\gamma_{A'}$ defines another complex structure on $\mathbb{E}^4$ which coincides with the complex structure on $\mathbb{E}^4 \subset \mathbb{PT}$ at the point $\pi_{A'} = \gamma_{A'}$. It coincides with the opposite complex structure – swapping holomorphic and anti-holomorphic – at the antipodal point $\pi_{A'} = \hat{\gamma}_{A'}$. Using the spinor $\mu^A$, we can define a basis of one-forms adapted to this complex structure,

$$
\begin{aligned}
\mathrm{d}z &= \mu_A\gamma_{A'}\mathrm{d}x^{AA'} \ , & \mathrm{d}\bar{z} &= \hat{\mu}_A\hat{\gamma}_{A'}\mathrm{d}x^{AA'} \ , \\
\mathrm{d}w &= \hat{\mu}_A\gamma_{A'}\mathrm{d}x^{AA'} \ , & \mathrm{d}\bar{w} &= -\mu_A\hat{\gamma}_{A'}\mathrm{d}x^{AA'} \ .
\end{aligned}
\tag{4.2}
$$

We will perform symmetry reduction along the vector fields dual to $\mathrm{d}z$ and $\mathrm{d}\bar{z}$,

$$\partial_z = \hat{\mu}^A\hat{\gamma}^{A'}\partial_{AA'} \ , \quad \partial_{\bar{z}} = \mu^A\gamma^{A'}\partial_{AA'} \ . \tag{4.3}$$

The symmetry reduction along the $\partial_z$ and $\partial_{\bar{z}}$ directions takes us from a theory on $\mathbb{PT}$ to a theory on $\Sigma \times \mathbb{CP}^1$ in which $w, \bar{w}$ are coordinates on the worldsheet $\Sigma$.

To perform this reduction, it is expedient [26] to make use of the invariance of the action (2.1) under the addition of any $(1,0)$-form to $\mathcal{A} \mapsto \hat{\mathcal{A}} = \mathcal{A} + \rho_0 e^0 + \rho_A e^A$. By choosing $\rho_A$ appropriately, we can ensure that $\hat{\mathcal{A}}$ has no $\mathrm{d}z$ or $\mathrm{d}\bar{z}$ legs and is given by

$$\hat{\mathcal{A}} = \hat{\mathcal{A}}_w\mathrm{d}w + \hat{\mathcal{A}}_{\bar{w}}\mathrm{d}\bar{w} + \mathcal{A}_0\bar{e}^0 \ , \tag{4.4}$$

where these components are related to those of $\mathcal{A}$ by

$$\hat{\mathcal{A}}_w = -\frac{[\mathcal{A}\mu]}{\langle\pi\gamma\rangle} \ , \qquad \hat{\mathcal{A}}_{\bar{w}} = -\frac{[\mathcal{A}\hat{\mu}]}{\langle\pi\hat{\gamma}\rangle} \ . \tag{4.5}$$

An important feature to note is that $\hat{\mathcal{A}}$ necessarily has singularities at $\gamma$ and $\hat{\gamma}$. While at the 6-dimensional level this is a mere gauge-choice artefact, it plays a crucial role in the construction of the CS$_4$ theory.

In these variables, the boundary variation and boundary condition of hCS$_6$ are restated as

$$r_+\mathrm{Tr}\big(\hat{\mathcal{A}}_w\delta\hat{\mathcal{A}}_{\bar{w}} - \hat{\mathcal{A}}_{\bar{w}}\delta\hat{\mathcal{A}}_w\big)\big|_{\pi=\alpha} = -r_-\mathrm{Tr}\big(\hat{\mathcal{A}}_w\delta\hat{\mathcal{A}}_{\bar{w}} - \hat{\mathcal{A}}_{\bar{w}}\delta\hat{\mathcal{A}}_w\big)\big|_{\pi=\tilde{\alpha}} \ , \tag{4.6}$$

$$\hat{\mathcal{A}}_w|_{\pi=\alpha} = ts\hat{\mathcal{A}}_w|_{\pi=\tilde{\alpha}} \ , \qquad \hat{\mathcal{A}}_{\bar{w}}|_{\pi=\alpha} = t^{-1}s\hat{\mathcal{A}}_{\bar{w}}|_{\pi=\tilde{\alpha}} \ , \tag{4.7}$$

where we have introduced the combinations

$$r_+ = K\,\frac{\langle\alpha\gamma\rangle\langle\alpha\hat{\gamma}\rangle}{\langle\alpha\tilde{\alpha}\rangle\langle\alpha\beta\rangle^2} \ , \qquad\qquad r_- = -K\,\frac{\langle\tilde{\alpha}\gamma\rangle\langle\tilde{\alpha}\hat{\gamma}\rangle}{\langle\alpha\tilde{\alpha}\rangle\langle\tilde{\alpha}\beta\rangle^2} \ ,$$

$$s = \sqrt{-\frac{r_-}{r_+}} = \frac{\langle\alpha\beta\rangle}{\langle\tilde{\alpha}\beta\rangle}\sqrt{\frac{\langle\tilde{\alpha}\gamma\rangle\langle\tilde{\alpha}\hat{\gamma}\rangle}{\langle\alpha\gamma\rangle\langle\alpha\hat{\gamma}\rangle}} \ , \qquad t = \sigma s\,\frac{\langle\tilde{\alpha}\beta\rangle\langle\alpha\hat{\gamma}\rangle}{\langle\alpha\beta\rangle\langle\tilde{\alpha}\hat{\gamma}\rangle} \ . \tag{4.8}$$

Upon symmetry reduction to $\text{CS}_4$, $r_\pm$ will correspond to the residues of the 1-form $\omega$.

Since the shifted gauge field $\hat{\mathcal{A}}$ manifestly has no $\mathrm{d}z$ or $\mathrm{d}\bar{z}$ legs, and we impose $L_z\hat{\mathcal{A}} = L_{\bar{z}}\hat{\mathcal{A}} = 0$, the contraction by $\partial_z$ and $\partial_{\bar{z}}$ only hits $\Omega$. It then follows that the symmetry reduction yields

$$S_{\text{CS}_4} = \frac{1}{2\pi\mathrm{i}}\int_{\Sigma\times\mathbb{CP}^1} \omega \wedge \text{Tr}\left(\hat{\mathcal{A}}\wedge\mathrm{d}\hat{\mathcal{A}} + \frac{2}{3}\hat{\mathcal{A}}\wedge\hat{\mathcal{A}}\wedge\hat{\mathcal{A}}\right) \ , \tag{4.9}$$

in which the meromorphic 1-form on $\mathbb{CP}^1$ is given by

$$\omega = \partial_{\bar{z}}\lrcorner\,\partial_z\lrcorner\,\Omega = \Phi\,\varepsilon_{AB}\left(\partial_{\bar{z}}\lrcorner\,e^A\right)\left(\partial_z\lrcorner\,e^B\right)e^0 = -K\,\frac{\langle\pi\gamma\rangle\langle\pi\hat{\gamma}\rangle}{\langle\pi\alpha\rangle\langle\pi\tilde{\alpha}\rangle\langle\pi\beta\rangle^2}\,e^0 \ . \tag{4.10}$$

To compare with the literature, we will also translate to inhomogeneous coordinates on $\mathbb{CP}^1$. The $\mathbb{CP}^1$ coordinate itself will be given by $\zeta = \pi_{2'}/\pi_{1'}$ on the patch $\pi_{1'} \neq 0$. We also specify representatives for the various other spinors in our theory. Without loss of generality we can choose

$$\alpha_{A'} = (1,\alpha_+) \ , \qquad \tilde{\alpha}_{A'} = (1,\alpha_-) \ , \qquad \beta_{A'} = (0,1) \ , \tag{4.11}$$

such that

$$\langle\tilde{\alpha}\beta\rangle = \langle\alpha\beta\rangle = 1 \ , \qquad \langle\tilde{\alpha}\alpha\rangle = \alpha_+ - \alpha_- = \Delta\alpha \ . \tag{4.12}$$

We also denote the inhomogeneous coordinates for $\gamma_{A'}$ and $\hat{\gamma}_{A'}$ by

$$\gamma_+ = \frac{\gamma_{2'}}{\gamma_{1'}} \ , \qquad \gamma_- = \frac{\hat{\gamma}_{2'}}{\hat{\gamma}_{1'}} = -\frac{\overline{\gamma_{1'}}}{\overline{\gamma_{2'}}} \ , \qquad \gamma_{1'}\overline{\gamma_{2'}} = \frac{1}{\gamma_+ - \gamma_-} = \frac{1}{\Delta\gamma} \ . \tag{4.13}$$

Then, the meromorphic 1-form $\omega$ is written in inhomogeneous coordinates as

$$\omega = \frac{K}{\Delta\gamma}\,\frac{(\zeta-\gamma_+)(\zeta-\gamma_-)}{(\zeta-\alpha_+)(\zeta-\alpha_-)}\,\mathrm{d}\zeta = \varphi(\zeta)\,\mathrm{d}\zeta \ . \tag{4.14}$$

To complete the specification of the theory we simply note that the 6d boundary conditions immediately descend to

$$\hat{\mathcal{A}}_w\,|_{\pi=\tilde{\alpha}} = ts\hat{\mathcal{A}}_w\,|_{\pi=\tilde{\alpha}} \ , \qquad \hat{\mathcal{A}}_{\bar{w}}\,|_{\pi=\alpha} = t^{-1}s\hat{\mathcal{A}}_{\bar{w}}\,|_{\pi=\tilde{\alpha}} \ . \tag{4.15}$$

Before we discuss the residual symmetries of the $\text{CS}_4$ models, let us make two related observations. First, fixing the shift symmetry to ensure $\hat{\mathcal{A}}$ is horizontal with respect to the symmetry reduction introduces poles into our gauge field $\hat{\mathcal{A}}$ at $\gamma$ and $\hat{\gamma}$. Thus, despite starting with potentially smooth field configurations in 6 dimensions we are forced to consider singular ones in 4 dimensions. We can understand the origin of these singularities by recalling the holomorphic coordinates on $\mathbb{E}^4$ with respect to the complex structure on $\mathbb{PT} = \mathbb{CP}^3\backslash\mathbb{CP}^1$. Described in detail in appendix A.1, $\mathbb{PT}$ is only diffeomorphic to $\mathbb{E}^4\times\mathbb{CP}^1$, and the complex structure is more involved in these coordinates. The holomorphic coordinates on $\mathbb{E}^4$ with respect to this complex structure are given by $v^A = \pi_{A'}x^{AA'}$, which align with our coordinates $\{z,w\}$ at $\pi\sim\gamma$ and $\{\bar{z},\bar{w}\}$ at $\pi\sim\hat{\gamma}$. It is precisely at these points that we are forced to introduce poles by the symmetry reduction procedure.

Second, in line with the singular behaviour in the gauge field, we have also introduced zeroes in $\omega$ at $\pi\sim\gamma$ and $\pi\sim\hat{\gamma}$ whereas $\Omega$ in 6 dimensions was nowhere vanishing. Of course, given the pole structure of $\Omega$, the introduction of two zeroes was inevitable given the Riemann-Roch theorem.

## 4.1 Residual Symmetries and the Defect Algebra

Let us take a moment to consider the residual symmetries of these $CS_4$ models. Here the residual symmetry preserving the boundary condition (4.15) is generically constrained to only include constant modes,

$$r = \tilde{r} \ , \qquad (1 - ts)\partial_w r = 0 \ , \qquad (1 - ts^{-1})\partial_{\bar{w}} r = 0 \ . \tag{4.16}$$

At the special 'diagonal' point in parameter space where $t = s = 1$, notice these differential equations are identically solved and we find a local gauge symmetry. This enhancement of residual gauge freedom matches with previous considerations in the context of $CS_4$, where diagonal boundary conditions of the form $A|_\alpha = A|_{\tilde{\alpha}}$ are known to give rise to the $\lambda$-deformed WZW as an $IFT_2$, a theory that depends on a single field $h$. The residual gauge symmetries are those satisfying $\hat{g}|_\alpha = \hat{g}|_{\tilde{\alpha}}$ and can be used to reduce the number of fields appearing in the resulting $IFT_2$ to one (see § 5.4 in [9]).

Another interesting point occurs when we take $t = s$ or $t = s^{-1}$ in which case we retain a chiral residual symmetry. When $t = 0$ the boundary conditions admit an enlarged residual symmetry as there is no requirement that $r = \hat{g}|_\alpha$ and $\tilde{r} = \hat{g}|_{\tilde{\alpha}}$ match. Instead they must be chiral and of opposite chiralities i.e. $\partial_w r = \partial_{\bar{w}} \tilde{r} = 0$. As mentioned earlier, for more generic values of $t$ and $s$ the residual symmetries will be constrained, preventing them from being used to eliminate any degrees of freedom. While these boundary conditions have not been yet considered for $t, s \neq 1$ and $t \neq 0$, we will see that they give rise to the multi-parametric class of $\lambda$-deformations between coupled WZW models introduced in [20].

To make further contact with the literature, it is helpful to rephrase the boundary conditions (4.15) in terms of a defect algebra, which in the case at hand is simply the Lie algebra $\mathfrak{d} = \mathfrak{g} + \mathfrak{g}$ equipped with an ad-invariant pairing

$$\langle\!\langle \mathbb{X}, \mathbb{Y} \rangle\!\rangle = r_+ \text{Tr}(x_1 \, y_1) + r_- \text{Tr}(x_2 \, y_2) \ , \qquad \mathbb{X} = (x_1, x_2) \ , \mathbb{Y} = (y_1, y_2) \ . \tag{4.17}$$

We map our boundary conditions into this algebra by defining $\mathbb{A}_w = (\hat{\mathcal{A}}_w|_{\pi=\alpha}, \hat{\mathcal{A}}_w|_{\pi=\tilde{\alpha}})$ and $\mathbb{A}_{\bar{w}} = (\hat{\mathcal{A}}_{\bar{w}}|_{\pi=\alpha}, \hat{\mathcal{A}}_{\bar{w}}|_{\pi=\tilde{\alpha}})$ such that the requirement that the boundary variation vanishes locally can be recast as

$$0 = \langle\!\langle \mathbb{A}_w, \delta\mathbb{A}_{\bar{w}} \rangle\!\rangle - \langle\!\langle \mathbb{A}_{\bar{w}}, \delta\mathbb{A}_w \rangle\!\rangle \ . \tag{4.18}$$

The boundary conditions (4.15) read

$$\mathbb{A}_w \in \mathfrak{l}_t = \text{span}\{(tsx, x) \mid x \in \mathfrak{g}\} \ , $$
$$\mathbb{A}_{\bar{w}} \in \mathfrak{l}_{t^{-1}} = \text{span}\{(t^{-1}sx, x) \mid x \in \mathfrak{g}\} \ . \tag{4.19}$$

Since $\langle\!\langle \mathfrak{l}_t, \mathfrak{l}_{t^{-1}} \rangle\!\rangle = 0$ the boundary conditions are suitable, however it should be noted that $\mathfrak{l}_t$ is itself neither a subalgebra nor an isotropic subspace of $\mathfrak{d}$. This is more general than boundary conditions previously considered[18] in the context of 4-dimensional Chern Simons theory. In particular, we might expect that generalising [28, 29, 30] to boundary conditions defined by subspaces that are neither a subalgebra nor an isotropic subspace of $\mathfrak{d}$ will lead to novel families of 2-dimensional integrable field theories.

It is worth highlighting that these boundary conditions still define maximal isotropic subspaces, but now inside the space of algebra-valued 1-forms, rather than just the defect algebra. Consider the space of $\mathfrak{g}$-valued 1-forms on $\Sigma \times \mathbb{CP}^1$, equipped with the symplectic structure[19]

$$\mathcal{W}(X, Y) = \int_{\Sigma \times \mathbb{CP}^1} \mathrm{d}\omega \wedge \text{Tr}(X \wedge Y) \ , \quad X, Y \in \Omega^1(\Sigma \times \mathbb{CP}^1) \otimes \mathfrak{g} \ . \tag{4.20}$$

---

[18]Of course in the limit $t, s \to 1$ $\mathfrak{l}_t$ revert to defining the diagonal isotropic subalgebra. In the special case where $t \to 0, \infty$ we recover chiral Dirichlet boundary conditions considered in [8, 27].

[19]As defined, this is not quite a symplectic structure since it is degenerate – for example, it vanishes on the subspace of 1-forms which only have legs along $\mathbb{CP}^1$. A more careful treatment would involve restricting the symplectic form to a subspace where it is non-degenerate, but we will neglect this for the purpose of our brief discussion.

The boundary conditions above define maximal isotropic subspaces with respect to this symplectic structure, that is half-dimensional subspaces $\mathcal{Y} \subset \Omega^1(\Sigma \times \mathbb{CP}^1) \otimes \mathfrak{g}$ such that $\mathcal{W}(X,Y) = 0$ for all $X, Y \in \mathcal{Y}$. Indeed, this is required for them to be 'good' boundary conditions. The isotropic subspaces of the defect algebra described earlier are then special cases of these subspaces.

## 5 Symmetry Reduction of IFT$_4$ to IFT$_2$

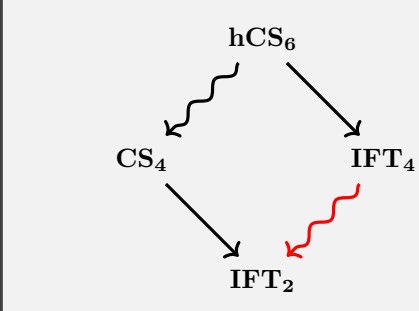

In this section we will apply the same symmetry reduction previously applied to hCS$_6$ to the IFT$_4$. In doing so we derive the IFT$_2$ corresponding to the CS$_4$ model described above.

Recalling that the reduction requires that the fields $h$ and $\tilde{h}$ depend only on $w, \bar{w}$ and not on $z, \bar{z}$, we can simply set $\partial_z = \partial_{\bar{z}} = 0$ in the action eq. (3.16). To compare with the literature, when discussing 2-dimensional theories we will define $\partial_+ \equiv \partial_w$ and $\partial_- \equiv \partial_{\bar{w}}$ (implicitly rotating to 2d Minkowski space where the action is rendered real for real parameters) and denote

$$J_\pm = h^{-1}\partial_\pm h , \qquad \widetilde{J}_\pm = \tilde{h}^{-1}\partial_\pm \tilde{h} . \tag{5.1}$$

To evaluate the symmetry reduction, denoted by $\rightsquigarrow$, of the IFT$_4$ action we first note that

$$j \rightsquigarrow \frac{\langle \alpha\gamma \rangle}{\langle \alpha\beta \rangle} J_+ , \quad \hat{j} \rightsquigarrow \frac{\langle \alpha\hat{\gamma} \rangle}{\langle \alpha\beta \rangle} J_- , \quad \widetilde{j} \rightsquigarrow \frac{\langle \tilde{\alpha}\gamma \rangle}{\langle \tilde{\alpha}\beta \rangle} \widetilde{J}_+ , \quad \widehat{\widetilde{j}} \rightsquigarrow \frac{\langle \tilde{\alpha}\hat{\gamma} \rangle}{\langle \tilde{\alpha}\beta \rangle} \widetilde{J}_- . \tag{5.2}$$

The resulting 2-dimensional action is given by

$$\begin{aligned} S_{\text{IFT}_4} \rightsquigarrow S_{\text{IFT}_2} &= \int_\Sigma \text{vol}_2 \, \text{Tr}\big(r_+ J_+(U_+^T - U_-)J_- - r_- \widetilde{J}_+(U_+^T - U_-)\widetilde{J}_- + r_+ \mathcal{L}_{\text{WZ}}(h) + r_- \mathcal{L}_{\text{WZ}}(\tilde{h}) \\ &\quad - 2\, t\sqrt{-r_+ \cdot r_-}\, \widetilde{J}_+ U_+^T J_- + 2\, t^{-1}\sqrt{-r_+ \cdot r_-}\, J_+ U_- \widetilde{J}_-\big) , \end{aligned} \tag{5.3}$$

where $\text{vol}_2 = \mathrm{d}\bar{w} \wedge \mathrm{d}w = \mathrm{d}\sigma^- \wedge \mathrm{d}\sigma^+$. This theory, depending on two $G$-valued fields, $h$ and $\tilde{h}$, and four independent parameters, $r_\pm$, $t$ and $\sigma$, exactly matches a theory introduced in [20] as a multi-field generalisation of the $\lambda$-deformed WZW model [2]. To make a precise match with [20] we relate their fields $(g_1, g_2)$ to our fields $(h, \tilde{h}^{-1})$. The model in [20] is defined by two WZW levels $k_{1,2}$ and by two deformation matrices which we take to be proportional to the identity with constants of proportionality $\lambda_{1,2}$. The mapping of parameters is then

$$\lambda_1 = \sigma t^{-1} , \qquad \lambda_2 = t , \qquad k_1 = r_+ , \qquad k_2 = -r_- , \qquad \lambda_0 = \sqrt{k_1/k_2} = \sqrt{-r_-/r_+} = s^{-1} . \tag{5.4}$$

In 2d Minkowski space, the Lagrangian (5.3) is real if the parameters $r_\pm$, $s\,t$ and $\sigma$ are all real. Assuming $K$ and $\sigma$ are real, this is the case if $r_+$ and $r_-$ have the opposite sign and the parameters $\alpha_\pm$ and $\gamma_\pm$ lie on the same line in $\mathbb{C}$, which we can take to be the real line without loss of generality. This follows since $r_\pm$, $s$ and $t$ are all expressed as ratios of differences of $\alpha_\pm$ and $\gamma_\pm$, hence are invariant under translations and scalings.[20]

---

[20]Note that this is the subgroup of $SL(2, \mathbb{C})$ transformations that preserves the choice $\beta_{A'} = (0, 1)$.

## 5.1 Limits

The four-parameter model has a number of interesting limits, many of which are discussed in [20]. Here, we briefly summarise some key ones. First, let us take $t \to 0$. In order to have a well-defined limit we keep $\sigma t^{-1} = \lambda$ finite, implying $\sigma \to 0$ as well.[21] The resulting 2d Lagrangian is given by

$$S_{\text{IFT}_2}|_{t,\sigma \to 0} = \int_\Sigma \text{vol}_2 \, \text{Tr}\big(r_+ J_+ J_- - r_- \widetilde{J}_+ \widetilde{J}_- - 2\lambda s r_+ J_+ \Lambda^{-1} \widetilde{J}_- + r_+ \mathcal{L}_{\text{WZ}}(h) + r_- \mathcal{L}_{\text{WZ}}(\tilde{h})\big) \,. \quad (5.5)$$

This current-current deformation preserves half the chiral symmetry of the $G_{r_+} \times G_{-r_-}$ WZW model, which corresponds to the UV fixed point $\lambda = 0$. Indeed, this model can be found by taking chiral Dirichlet boundary conditions in 4d CS [8, 27], corresponding to the special case $t = 0$ in the boundary conditions we find from symmetry reduction (4.19). Assuming $-r_- > r_+$, in the IR we have that $\lambda = s^{-1}$. At this point the Lagrangian can be written as

$$S_{\text{IFT}_2}|_{t,\sigma \to 0, t\sigma^{-1}=s} = \int_\Sigma \text{vol}_2 \, \text{Tr}\big(r_+ (\text{Ad}_h J_+ - \text{Ad}_{\tilde{h}} \widetilde{J}_+)(\text{Ad}_h J_- - \text{Ad}_{\tilde{h}} \widetilde{J}_-) - (r_- + r_+) \widetilde{J}_+ \widetilde{J}_-$$
$$+ r_+ \mathcal{L}_{\text{WZ}}(\tilde{h}^{-1}h) + (r_- + r_+) \mathcal{L}_{\text{WZ}}(\tilde{h})\big) \,. \quad (5.6)$$

Redefining $h \to \tilde{h}h$, we find the $G_{r_+} \times G_{-r_- - r_+}$ WZW model. In the case of equal levels $r_- = -r_+$ this reduces to the $G_{r_+}$ WZW model.

The equal-level, $r_- = -r_+$, version of (5.5), whose classical integrability was first shown in [31], is canonically equivalent [32] and related by a path integral transformation [33] to the $\lambda$-deformed WZW model. Indeed, from the point of view of 4d CS, these two models have the same twist function. To recover (5.5) with equal levels, we take chiral Dirichlet boundary conditions, $t = 0$, $s = 1$ in (4.19), while to recover the $\lambda$-deformed WZW model we take diagonal boundary conditions $t = s = 1$.

It follows that if we take $t = s = 1$ in eq. (5.3), we expect to recover the $\lambda$-deformed WZW model. Indeed, setting $r_- = -r_+$ and $t = 1$, the Lagrangian (5.3) becomes

$$S_{\text{IFT}_2}|_{t=s=1} = \int_\Sigma \text{vol}_2 \, \text{Tr}\big(r_+ (J_+ - \widetilde{J}_+)(U_+^T - U_-)(J_- - \widetilde{J}_-) + r_+ \mathcal{L}_{\text{WZ}}(h\tilde{h}^{-1})\big) \,. \quad (5.7)$$

As explained in subsection 2.1, at this point in parameter space the symmetry reduction directions are aligned such that the constrained symmetry transformations (2.14) become a gauge symmetry of the IFT$_2$. This allows us to fix $\tilde{h} = 1$, recovering the standard form of the $\lambda$-deformed WZW model [2] with $\sigma$ playing the role of $\lambda$. Further taking $\sigma \to 0$, we recover the $G_{r_+}$ WZW model.

Another point in parameter space where we expect a gauge symmetry to emerge is when the symmetry reduction preserves the left-acting symmetry. This corresponds to setting $t = \sigma$ and $s = 1$, i.e. $r_- = -r_+$. Doing so we find

$$S_{\text{IFT}_2}|_{t=\sigma, s=1} = \int_\Sigma \text{vol}_2 \, \text{Tr}\big(r_+ (\text{Ad}_h J_+ - \text{Ad}_{\tilde{h}} \widetilde{J}_+)(\widetilde{U}_+^T - \widetilde{U}_-)(\text{Ad}_h J_- - \text{Ad}_{\tilde{h}} \widetilde{J}_-) + r_+ \mathcal{L}_{\text{WZ}}(\tilde{h}^{-1}h)\big) \,, \quad (5.8)$$

where we recall that $\widetilde{U}_\pm$ are defined in eq. (3.60). This Lagrangian is invariant under a left-acting gauge symmetry as expected, which can be used to fix $\tilde{h} = 1$. We again recover the standard form of the $\lambda$-deformed WZW model with $\sigma$ playing the role of $\lambda$. The CS$_4$ description of this limit is analysed in appendix C.

Before we move onto the integrability of the 2d IFT and its origin from the 4d IFT, let us briefly note the symmetry reduction implications of the formal transformations (3.18) and (3.19), which in turn descended from the discrete invariances of the hCS$_6$ boundary conditions (2.9) and (2.10). The first (3.18) implies that the 2d IFT is invariant under

$$r_+ \leftrightarrow r_- \,, \qquad \sigma \to \sigma^{-1} \,, \qquad t \to t^{-1} \,, \qquad h \leftrightarrow \tilde{h} \,. \quad (5.9)$$

---

[21]An analogous limit is to take $\sigma \to 0$ and keep $t$ finite.

recovering the 'duality' transformation of [20]. Since the second involves interchanging $w$ and $z$, it tells us the parameters are transformed if we symmetry reduce requiring that the fields $h$ and $\tilde{h}$ only depend on $z, \bar{z}$, instead of $w, \bar{w}$. We find that $\sigma \to \sigma^{-1}$ and $t \to t\sigma^{-2}$.

## 5.2 Integrability and Lax Formulation

The analysis of [20] shows that the equations of motion of (5.3) are best cast in terms of auxiliary fields[22] $B_\pm, C_\pm$ which are related to the fundamental fields by

$$
\begin{aligned}
J_- &= \mathrm{Ad}_h^{-1} B_- - \lambda_0^{-1}\lambda_2^{-1} C_- \,, \qquad & \tilde{J}_- &= \lambda_0\lambda_1^{-1}\mathrm{Ad}_{\tilde{h}}^{-1} B_- - C_- \,, \\
J_+ &= \lambda_0^{-1}\lambda_1^{-1}\mathrm{Ad}_h^{-1} B_+ - C_+ \,, \qquad & \tilde{J}_+ &= \mathrm{Ad}_{\tilde{h}}^{-1} B_+ - \lambda_0\lambda_2^{-1} C_+ \,.
\end{aligned}
\tag{5.10}
$$

The equations of motion for $h$ and $\tilde{h}$, together with the Bianchi identities obeyed by their associated Maurer-Cartan forms, can be repackaged into the flatness of two Lax connections with components

$$
\mathcal{L}_\pm^1 = \frac{2\zeta_{\mathrm{GS}}}{\zeta_{\mathrm{GS}} \mp 1} \frac{1 - \lambda_0^{\mp 1}\lambda_1}{1 - \lambda_1^2} B_\pm \,, \qquad \mathcal{L}_\pm^2 = \frac{2\zeta_{\mathrm{GS}}}{\zeta_{\mathrm{GS}} \mp 1} \frac{1 - \lambda_0^{\pm 1}\lambda_2}{1 - \lambda_2^2} C_\pm \,.
\tag{5.11}
$$

Here $\zeta_{\mathrm{GS}}$ is the spectral parameter used in [20]. Taken together, the flatness of this pair of Lax connections implies both the Bianchi identities and the equations of motions. However, if one is prepared to enforce the definition (5.10) of auxiliary fields in terms of fundamental fields (such that the Bianchi equations are automatically satisfied) then either Lax will generically (i.e. away from special points in parameter space such as $\lambda_i = 1$) imply the equations of motion[23] of the theory.

We can relate this discussion to the construction above by symmetry reducing the 4d Lax operators (3.47) and (3.48). First we note that the currents corresponding to the $(\ell, r)$-symmetries reduce to simple combinations of the auxiliary fields introduced in eq. (5.10)

$$
\begin{aligned}
B_A &\rightsquigarrow \frac{\langle \alpha\hat{\gamma} \rangle}{\langle \alpha\beta \rangle} B_- \mu_A - \frac{\langle \tilde{\alpha}\gamma \rangle}{\langle \tilde{\alpha}\beta \rangle} B_+ \hat{\mu}_A \,, \\
C_A &\rightsquigarrow \frac{\langle \tilde{\alpha}\hat{\gamma} \rangle}{\langle \tilde{\alpha}\beta \rangle} \sigma^{-1} C_- \mu_A - \frac{\langle \alpha\gamma \rangle}{\langle \alpha\beta \rangle} C_+ \hat{\mu}_A \,.
\end{aligned}
\tag{5.12}
$$

Notice that all explicit appearances of the operators $U_\pm$ have dropped out such that these currents reduce exactly to the 2-dimensional auxiliary gauge fields.

Using the complex coordinates adapted for symmetry reduction defined in eq. (4.2), and introducing a specialised inhomogeneous coordinate on $\mathbb{CP}^1$ given by $\varsigma = \langle \pi\hat{\gamma} \rangle / \langle \pi\gamma \rangle$, the 4d $B$-Lax pair (3.47) may be written as

$$
L^{(B)} = D_{\bar{w}} - \varsigma^{-1}D_z \,, \qquad M^{(B)} = D_w + \varsigma D_{\bar{z}} \,.
\tag{5.13}
$$

We can symmetry reduce the 4d Lax pairs, $L^{(B/C)}, M^{(B/C)}$ of eqs. (3.47) and (3.48) to obtain

$$
\begin{aligned}
L^{(B)} &\rightsquigarrow \partial_- + (\langle \beta\gamma \rangle - \varsigma^{-1}\langle \beta\hat{\gamma} \rangle)\frac{\langle \alpha\hat{\gamma} \rangle}{\langle \alpha\beta \rangle} B_- \,, \quad & M^{(B)} &\rightsquigarrow \partial_+ + (\varsigma\langle \beta\gamma \rangle - \langle \beta\hat{\gamma} \rangle)\frac{\langle \tilde{\alpha}\gamma \rangle}{\langle \tilde{\alpha}\beta \rangle} B_+ \,, \\
L^{(C)} &\rightsquigarrow \partial_- - \frac{1}{(1 - \varrho) + \lambda_0\lambda_2(1 + \varrho)} C_- \,, \quad & M^{(C)} &\rightsquigarrow \partial_+ - \frac{1}{(1 + \varrho) + \lambda_0^{-1}\lambda_2(1 - \varrho)} C_+ \,.
\end{aligned}
\tag{5.14}
$$

Now using the inhomogeneous coordinates introduced in eqs. (4.11) and (4.13), and the relations between parameters (5.4), the 4d Lax operators immediately descend upon symmetry reduction to the 2d Lax connections (5.11), provided the 4d and 2d spectral parameters are related as

$$
\begin{aligned}
L^{(B)} &\rightsquigarrow \partial_- + \mathcal{L}_-^1 \,, \quad M^{(B)} \rightsquigarrow \partial_+ + \mathcal{L}_+^1 \,, \quad & \zeta_{\mathrm{GS}} &= \frac{\bar{\gamma}_2 + \gamma_1\varsigma}{-\bar{\gamma}_2 + \gamma_1\varsigma} \,, \\
L^{(C)} &\rightsquigarrow \partial_- + \mathcal{L}_-^2 \,, \quad M^{(C)} \rightsquigarrow \partial_+ + \mathcal{L}_+^2 \,, \quad & \zeta_{\mathrm{GS}} &= \frac{1 - \lambda_2^2}{(\lambda_0 - \lambda_0^{-1})\lambda_2 - (1 - \lambda_0\lambda_2)(1 - \lambda_0^{-1}\lambda_2)\varrho} \,.
\end{aligned}
\tag{5.15}
$$

---

[22]To avoid conflict of notation $B_\pm, C_\pm$ here correspond to $A_\pm, B_\pm$ of [20].

[23]It is less evident in contrast if *all* the non-local conserved charges of the theory can be obtained from a single Lax.

The relation between $\zeta_{\text{GS}}$ and $\varsigma$ can be recast in the standard $\mathbb{CP}^1$ homogeneous coordinate $\pi \sim (1, \zeta)$ as

$$\zeta_{\text{GS}} = \frac{\gamma_+ - \gamma_-}{2\varsigma - (\gamma_+ + \gamma_-)} \ , \tag{5.16}$$

such that if we choose to fix $\gamma_\pm = \pm 1$ then $\zeta_{\text{GS}} = \varsigma^{-1}$. If we make the assumption that the $\zeta_{\text{GS}}$ entering in the two different Lax formulations have the same origin then we can map between between $\varrho$ and the $\mathbb{CP}^1$ homogeneous coordinate

$$\varrho = -\frac{1 + \zeta}{2} \frac{1 + ts}{1 - ts} + \frac{1 - \zeta}{2} \frac{1 + ts^{-1}}{1 - ts^{-1}} \ . \tag{5.17}$$

Therefore, under this assumption, we see that $\varrho$ depends on the parameter $t$, which is part of the specification of boundary conditions and not just geometric data of $\mathbb{CP}^1$. Indeed $\varrho$ becomes constant when $t \to 1$, hence there is no spectral parameter dependence left. In contrast, when $t \to 0$, we have $\varrho \to -\zeta$.

# 6   Localisation of CS$_4$ to IFT$_2$

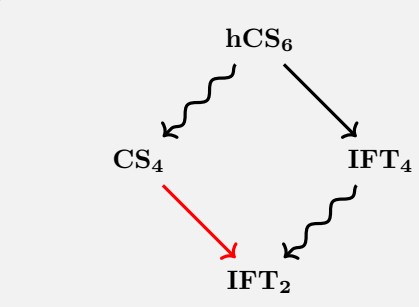

Finally, we localise the CS$_4$ theory obtained by symmetry reduction of hCS$_6$ in § 4. This will result in a 2-dimensional theory on $\Sigma$, which matches the IFT$_2$ derived from symmetry reduction of the IFT$_4$ in § 5.

In the following discussion we will make use of the $\mathcal{E}$-model formulation of CS$_4$ [28, 29, 30]. In this approach we accomplish localisation via algebraic means, constructing from the data of our CS$_4$ a *defect algebra* and *projectors*. The choice of boundary conditions in CS$_4$ corresponds to a choice of two mutually orthogonal subspaces of our defect algebra, from which we can then write down the action and Lax connection for the corresponding 2d IFT. To obtain the IFT$_2$ we could also follow an analogous route to that taken § 3, namely integrating out the $\mathbb{CP}^1$ directions directly. Details of this approach are presented in appendix B.

The gauge field $\hat{\mathcal{A}}$ is related to the 2-dimensional Lax connection by a change of variables that takes the form of a gauge transformation, which importantly does not preserve the boundary conditions,

$$\hat{\mathcal{A}} = \hat{h}^{-1} \mathrm{d}\hat{h} + \hat{h}^{-1} \mathcal{L} \hat{h} \ . \tag{6.1}$$

As before, we fix the redundancy in this parametrisation by demanding that $\mathcal{L}$ has no legs in the $\mathbb{CP}^1$ direction, though may of course depend on it functionally, and that $\hat{h}|_\beta = \mathrm{id}$. The bulk equations of motion, $\omega \wedge F[\hat{\mathcal{A}}] = 0$, ensures that $\mathcal{L}$ is flat and meromorphic in $\zeta$ with analytic structure mirroring $\omega$. The key idea is that the field $\hat{h}$ evaluated at the poles serve as edge modes that become the degrees of freedom of the IFT$_2$, and the boundary conditions will determine the form of the Lax connection $\mathcal{L}$ in terms of these fields. However, the complete construction requires a more careful treatment, especially when $\Omega$ has higher order poles [28].

Let us start by recalling the boundary conditions eq. (4.19) phrased in terms of $\mathbb{A}_w$ and $\mathbb{A}_{\bar{w}}$ valued in the defect algebra $\mathfrak{d} = \mathfrak{g} + \mathfrak{g}$. These are that $\mathbb{A}_w \in \mathfrak{l}_t$ and $\mathbb{A}_{\bar{w}} \in \mathfrak{l}_{t^{-1}}$, where these subspaces are mutually orthogonal $\langle\!\langle \mathfrak{l}_t, \mathfrak{l}_{t^{-1}} \rangle\!\rangle = 0$. Given these boundary conditions for the gauge field at the simple poles, we introduce a group valued field $\mathbb{h} = (h|_\alpha, h|_{\tilde{\alpha}}) \in \mathbb{D} = \exp \mathfrak{d}$ and an algebra element $\mathbb{L} = (\mathcal{L}|_\alpha, \mathcal{L}|_{\tilde{\alpha}}) \in \mathfrak{d}$ such

that

$$\mathbb{A}_w = \mathbb{h}^{-1}\partial_w\mathbb{h} + \mathbb{h}^{-1}\mathbb{L}_w\mathbb{h} \in \mathfrak{l}_t$$

$$\mathbb{A}_{\bar{w}} = \mathbb{h}^{-1}\partial_{\bar{w}}\mathbb{h} + \mathbb{h}^{-1}\mathbb{L}_{\bar{w}}\mathbb{h} \in \mathfrak{l}_{t^{-1}}$$

(6.2)

$\mathbb{L}$ can be understood as the 2d Lax connection lifted to the double by evaluating it at the poles of the spectral parameter. From this, and the known singularity structure of $\hat{\mathcal{A}}$, we will recover the full Lax connection.

It is important to emphasise that most previous treatments have assumed that $\mathbb{A}_w$ and $\mathbb{A}_{\bar{w}}$ lie in the same subspace, and moreover that this space is a maximal isotropic subalgebra $\mathfrak{l} \subset \mathfrak{d}$. The only exception that we know of are the chiral Dirichlet boundary conditions of [8, 27], which are a special case of our boundary conditions. Taking $\mathfrak{l}$ to be an isotropic subalgebra of $\mathfrak{d}$ ensures that the resulting IFT$_2$ has a residual gauge symmetry given by the left action of $\exp(\mathfrak{l})$ on $\mathbb{h}$. This can be fixed by setting $\mathbb{h} \in \mathbb{D}/\exp(\mathfrak{l})$. For example, if we take $r_+ = -r_-$, then $\mathfrak{l} = \mathfrak{g}_{\text{diag}}$, the diagonally embedded $\mathfrak{g}$ in $\mathfrak{d} = \mathfrak{g} + \mathfrak{g}$, is a suitable isotropic subalgebra and denoting $\mathbb{h} = (h, \tilde{h})$ the residual symmetry can be fixed by setting $\tilde{h} = \text{id}$. Here, however, we do not have any such residual gauge symmetry in general and the 2-dimensional theory will depend on the entire field content in $\mathbb{h}$.

As above, we switch notation in 2 dimensions to $\partial_w = \partial_+$ and $\partial_{\bar{w}} = \partial_-$. The IFT$_2$ action is [30, 34][24][25]

$$S_{2\text{d}}(\mathbb{h}) = \int \left( \langle\!\langle \partial_-\mathbb{h}\mathbb{h}^{-1}, W_{\mathbb{h}}^+(\partial_+\mathbb{h}\mathbb{h}^{-1}) \rangle\!\rangle \right.$$
$$\left. - \langle\!\langle \partial_+\mathbb{h}\mathbb{h}^{-1}, W_{\mathbb{h}}^-(\partial_-\mathbb{h}\mathbb{h}^{-1}) \rangle\!\rangle \right) \mathrm{d}\sigma^- \wedge \mathrm{d}\sigma^+ + S_{\text{WZ}}[\mathbb{h}] \ .$$

(6.3)

The projectors $W_{\mathbb{h}}^\pm : \mathfrak{d} \to \mathfrak{d}$ are defined via their kernel and image

$$\text{Ker}\, W_{\mathbb{h}}^\pm = \text{Ad}_{\mathbb{h}}\mathfrak{l}_{t^{\pm 1}} \ , \qquad \text{Im}\, W_{\mathbb{h}}^\pm = \left\{ \left( \frac{\mathsf{y}}{\alpha_+ - \gamma_\pm}, \frac{\mathsf{y}}{\alpha_- - \gamma_\pm} \right) \ \middle| \ \mathsf{y} \in \mathfrak{g} \right\} \ ,$$

(6.4)

where we recall that $\zeta = \gamma_\pm$ are the zeroes of $\omega$. Care is required to correlate the zeroes of omega with the pole structure of $\mathcal{A}$, which has been determined by our symmetry reduction data. In the case at hand for instance, the zeroes at $\pi = \gamma$ and $\pi = \hat{\gamma}$ are associated to poles in $\mathcal{A}_w$ and in $\mathcal{A}_{\bar{w}}$ respectively.

In order to unpack the IFT$_2$ action, let us outline the calculation of $W_{\mathbb{h}}^+(\partial_+\mathbb{h}\mathbb{h}^{-1})$. Defining the useful combinations[26]

$$v_\pm = \alpha_\pm - \gamma_+ \ , \qquad u_\pm = \alpha_\pm - \gamma_- \ ,$$

(6.5)

we can parameterise the kernel and image of $W_{\mathbb{h}}^+$ as

$$\text{Ker}\, W_{\mathbb{h}}^+ = \left\{ \left( \frac{\text{Ad}_h\mathsf{x}}{v_+}, \frac{\sigma^{-1}\text{Ad}_h\Lambda^{-1}\mathsf{x}}{v_-} \right) \ \middle| \ \mathsf{x} \in \mathfrak{g} \right\} \ , \qquad \text{Im}\, W_{\mathbb{h}}^+ = \left\{ \left( \frac{\mathsf{y}}{v_+}, \frac{\mathsf{y}}{v_-} \right) \ \middle| \ \mathsf{y} \in \mathfrak{g} \right\} \ .$$

(6.6)

We decompose $\partial_+\mathbb{h}\mathbb{h}^{-1}$ into the kernel and image by solving

$$\left( \text{Ad}_h J_+, \text{Ad}_h \Lambda^{-1}\widetilde{J}_+ \right) = \left( \frac{\text{Ad}_h\mathsf{x}}{v_+}, \frac{\sigma^{-1}\text{Ad}_h\Lambda^{-1}\mathsf{x}}{v_-} \right) + \left( \frac{\mathsf{y}}{v_+}, \frac{\mathsf{y}}{v_-} \right) \ .$$

(6.7)

This yields

$$\mathsf{y} = \text{Ad}_h U_+ \left( J_+ v_+ - \sigma\widetilde{J}_+ v_- \right) = v_- B_+ \ ,$$

(6.8)

---

[24]The action below is related to the action in [30] with the redefinition $\mathbb{h} \to \mathbb{h}^{-1}$. This is due to the convention on gauge transformations. Indeed, there they consider $A^h = hAh^{-1} - \mathrm{d}hh^{-1}$ in contrast to our choice $A^h = h^{-1}Ah + h^{-1}\mathrm{d}h$.

[25]Note that we have chosen $\frac{1}{2\pi i}$ as an overall coefficient in equation (4.9), in contrast to $\frac{i}{4\pi}$ considered in [30].

[26]In terms of these parameters, the relations (4.8) become

$$r_+ = K\,\frac{u_+ v_+}{\Delta\gamma\Delta\alpha} \ , \qquad r_- = -K\,\frac{u_- v_-}{\Delta\gamma\Delta\alpha} \ , \qquad s = \sqrt{\frac{u_- v_-}{u_+ v_+}} \ , \qquad t = \sigma s\,\frac{u_+}{u_-} \ .$$

in which we see the reappearance of the auxiliary combinations encountered earlier in eq. (5.10). It follows that

$$W_{\mathbb{h}}^+(\partial_+\mathbb{h}\mathbb{h}^{-1}) = \left( \frac{v_-}{v_+}\, B_+, B_+ \right) \ ,$$ (6.9)

from which we can evaluate (trace implicit)

$$\langle\!\langle \partial_-\mathbb{h}\mathbb{h}^{-1}, W_{\mathbb{h}}^+(\partial_+\mathbb{h}\mathbb{h}^{-1}) \rangle\!\rangle = \frac{v_-}{v_+}\, r_+ J_-\mathrm{Ad}_h^{-1}B_+ + r_-\widetilde{J}_-\Lambda\mathrm{Ad}_h^{-1}B_-$$
$$= r_+ J_+ U_+^T J_- + r_-\widetilde{J}_+ U_-\widetilde{J}_- - t\sqrt{-r_-r_+}\,\widetilde{J}_+ U_+^T J_- + t^{-1}\sqrt{-r_-r_+}\, J_+ U_-\widetilde{J}_- \ .$$ (6.10)

In a similar fashion we find that

$$W_{\mathbb{h}}^-(\partial_-\mathbb{h}\mathbb{h}^{-1}) = \left( B_-, \frac{u_+}{u_-}\, B_- \right) \ ,$$ (6.11)

and

$$\langle\!\langle \partial_+\mathbb{h}\mathbb{h}^{-1}, W_{\mathbb{h}}^-(\partial_-\mathbb{h}\mathbb{h}^{-1}) \rangle\!\rangle$$
$$= r_+ J_+ U_- J_- + r_-\widetilde{J}_- U_+^T\widetilde{J}_+ + t\sqrt{-r_-r_+}\,\widetilde{J}_+ U_+^T J_- - t^{-1}\sqrt{-r_-r_+}\, J_+ U_-\widetilde{J}_- \ .$$ (6.12)

Taking the difference of eq. (6.10) and eq. (6.12) we find that the Lagrangian of the 2-dimensional action eq. (6.3) exactly matches the $\mathrm{IFT}_2$ obtained previously in eq. (5.3) by descent on the other side of the diamond. This explicitly verifies our diamond of theories.

Let us note that this $\mathrm{IFT}_2$ has also been constructed from $\mathrm{CS}_4$ in a two-step process in [35]. First, a more general 2-field model based on a twist function with additional poles and zeroes, and the familiar isotropic subalgebra boundary conditions, is constructed. Second, a special decoupling limit is taken, where a subset of these poles and zeroes collide. It remains to understand how to recover our boundary conditions (4.15) from those considered in [35].

To complete the circle of ideas we can also directly obtain a Lax formulation from $\mathrm{CS}_4$. This is essentially achieved by undoing the map into the defect algebra as follows. Given an element $\mathbb{X} = (x, y) \in \mathfrak{d}$, we determine $a, b \in \mathfrak{g}$ such that

$$(x, y) = \left( \frac{a}{u_+} + \frac{b}{v_+}, \frac{a}{u_+} + \frac{b}{v_-} \right) \ .$$ (6.13)

We introduce a map $\wp$ into the space of $\mathfrak{g}$-valued meromorphic functions

$$\wp : \mathbb{X} \mapsto \frac{a}{\zeta - \gamma_-} + \frac{b}{\zeta - \gamma_+} \ ,$$ (6.14)

in terms of which the components of the Lax connection are given by

$$\mathcal{L}_\pm = \wp W_{\mathbb{h}}^\pm(\partial_\pm\mathbb{h}\mathbb{h}^{-1}) = \frac{\alpha_\mp - \gamma_\pm}{\zeta - \gamma_\pm}\, B_\pm \ .$$ (6.15)

In this way we recover the Lax $\mathcal{L}^1$ that we obtained from symmetry reduction of the 4d ASDYM Lax pair with the identification (5.16). There does not appear an algebraic derivation in this spirit of the other Lax $\mathcal{L}^2$. This in contrast to the derivation of this model from $\mathrm{CS}_4$ in [35] where the extra data associated to the additional poles means that both Lax in (5.11) can be directly constructed. More generally, this highlights an interesting question that we leave for the future about the integrability and the counting of conserved charges, beyond the existence of a Lax connection, when we consider boundary conditions not based on isotropic subalgebras.

## 6.1   RG Flow

Let us recall the RG equations given in [20]

$$\dot{\lambda}_i = -\frac{c_G}{2\sqrt{k_1 k_2}}\, \frac{\lambda_i^2(\lambda_i - \lambda_0)(\lambda_i - \lambda_0^{-1})}{(1 - \lambda_i^2)^2} \ , \quad i = 1, 2 \ ,$$ (6.16)

where dot indicates the derivative with respect to RG 'time' $\frac{d}{d\log\mu}$ and $c_G$ is the dual Coxeter number. The levels $k_1$ and $k_2$ and $\lambda_0 = \sqrt{k_1/k_2}$ are RG invariants. In this section we will interpret this flow in terms of the data that is more natural from the perspective of 4d CS, namely the poles and zeroes of the differential

$$\omega = \frac{K}{\Delta\gamma} \frac{(\zeta - \gamma_+)(\zeta - \gamma_-)}{(\zeta - \alpha_+)(\zeta - \alpha_-)} \, d\zeta = \varphi(\zeta) \, d\zeta \ , \tag{6.17}$$

and the boundary conditions of the theory.

Using the map between parameters given in eq. (5.4) we can infer from eq. (6.16) a flow on the parameters $\{t, \alpha_\pm, \gamma_\pm, K\}$. Let us first consider the parameter $t = \lambda_2$. As discussed in [20], there is a flow from $t = 0$ in the UV to $t = \lambda_0$ in the IR (assuming that $\lambda_0 < 1$). Explicitly the flow equation

$$\dot{t} = \frac{c_G}{2k_2\lambda_0} \frac{t^2}{(1-t^2)^2} (t - \lambda_0)(t - \lambda_0^{-1}) \ , \tag{6.18}$$

has the solution

$$f(\lambda_0, t) + f(\lambda_0^{-1}, t) + t + t^{-1} = \frac{c_G}{2\sqrt{k_1 k_2}} \log \mu/\mu_{t_0} \ , \quad f(x, t) = x \log\left(\frac{t^{-1} - x}{t - x}\right) \ . \tag{6.19}$$

The interesting observation is that the boundary conditions

$$\begin{aligned} \mathbb{A}_w \in \mathfrak{l}_t = \text{span}\{(t\lambda_0^{-1} x, x) \mid x \in \mathfrak{g}\} \ , \\ \mathbb{A}_{\bar{w}} \in \mathfrak{l}_{t^{-1}} = \text{span}\{(\lambda_0^{-1} x, tx) \mid x \in \mathfrak{g}\} \ , \end{aligned} \tag{6.20}$$

display algebraic enhancements at the fixed points. In the UV, $t = 0$ limit, these boundary conditions become chiral, $\mathbb{A}_w \in \mathfrak{g}_R \subset \mathfrak{d}$ and $\mathbb{A}_{\bar{w}} \in \mathfrak{g}_L \subset \mathfrak{d}$. While $\mathfrak{g}_{L,R}$ are now subalgebras, neither are isotropic with respect to the inner product (4.17). In non-doubled notation the UV limit becomes

$$\hat{\mathcal{A}}_w|_\alpha = 0 \ , \qquad \hat{\mathcal{A}}_{\bar{w}}|_{\tilde{\alpha}} = 0 \ . \tag{6.21}$$

On the other hand in the IR limit, $t = \lambda_0$, we see that $\mathbb{A}_w \in \mathfrak{g}_{\text{diag}} \subset \mathfrak{d}$, again a subalgebra, but only an isotropic one for $k_1 = k_2$, i.e. $r_+ = -r_-$. In non-doubled notation the IR limit becomes[27]

$$\hat{\mathcal{A}}_w|_\alpha = \hat{\mathcal{A}}_w|_{\tilde{\alpha}} \ , \qquad k_1 \hat{\mathcal{A}}_{\bar{w}}|_\alpha = k_2 \hat{\mathcal{A}}_{\bar{w}}|_{\tilde{\alpha}} \ . \tag{6.22}$$

While in general, there are no residual gauge transformations preserving the boundary conditions, in the UV and IR limits we notice chiral boundary symmetries emerging. For example, in the IR these are those satisfying $g^{-1}\partial_{\bar{w}} g = 0$, which corresponds to $t = s^{-1}$ in eq. (4.16).

Let us now turn to the action of RG on the differential $\omega$. An immediate observation is that the RG invariant WZW levels are given by monodromies about simple poles[28]

$$\pm k_{1,2} = r_\pm = \frac{1}{2\pi i} \oint_{\alpha_\pm} \omega = \text{res}_{\zeta = \alpha_\pm} \varphi(\zeta) \ , \tag{6.23}$$

exactly in line with the conjecture of Costello (reported and supported by Derryberry [36]). While there are more parameters in $\omega$ than there are RG equations, we can form the ratios of poles and zeroes

$$q_\pm = \frac{\alpha_\pm - \gamma_+}{\alpha_\pm - \gamma_-} = \frac{v_\pm}{u_\pm} \ , \tag{6.24}$$

---

[27]The seemingly more democratic boundary condition of $t = 1$,

$$\sqrt{k_1}\,\hat{\mathcal{A}}_w|_\alpha = \sqrt{k_2}\,\hat{\mathcal{A}}_w|_{\tilde{\alpha}} \ , \quad \sqrt{k_1}\,\hat{\mathcal{A}}_{\bar{w}}|_\alpha = \sqrt{k_2}\,\hat{\mathcal{A}}_{\bar{w}}|_{\tilde{\alpha}}$$

which does define an isotropic space of $\mathfrak{d}$ (not a subalgebra however) is *not* attained along this flow.

[28]The monodromy about the double pole at infinity is trivially RG invariant since the sum of all the residues vanishes.

in terms of which the RG system of [20] translates to

$$\dot{q}_\pm = -\frac{c_G}{2K}\frac{(1+q_\mp)}{(-1+q_\mp)}\,q_\pm\,, \qquad \dot{K} = -\frac{c_G}{2}\frac{q_- + q_+}{(1-q_-)(1-q_+)}\ . \tag{6.25}$$

The RG invariants are given by

$$k_1 k_2 = \frac{K^2 q_- q_+}{(q_+ - q_-)^2}\,, \qquad \frac{k_1}{k_2} = \lambda_0^2 = \frac{q_+}{(1-q_+)^2}\frac{(1-q_-)^2}{q_-}\,, \tag{6.26}$$

which allows us to retain either of $q_\pm$ as independent variables. We can directly solve these equations

$$\sqrt{k_1 k_2}\,\frac{q_+ - q_-}{\sqrt{q_+ q_-}} + k_1 \log q_+ - k_2 \log q_- = \frac{c_G}{2}\log \mu/\mu_{q_0}\,, \tag{6.27}$$

and a remarkable feature, also conjectured by Costello, is that this quantity is precisely the contour integral between zeroes

$$\frac{\mathrm{d}}{\mathrm{d}\log\mu}\int_{\gamma_-}^{\gamma_+}\omega = \frac{c_G}{2}\ . \tag{6.28}$$

To best understand the action of the RG flow on the locations of the poles directly, we replace $K$ with the RG invariant $k_2$ (or $k_1$), and fix the zeroes to be located at $\gamma_\pm = \pm 1$. This yields the RG invariant relation

$$1 - \alpha_+^2 - \lambda_0^2(1 - \alpha_-^2) = 0\,, \tag{6.29}$$

and a flow equation

$$\dot{\alpha}_- = \frac{c_G}{8k_2}\frac{\alpha_+(1-\alpha_-^2)^2}{\alpha_- - \alpha_+}\,, \tag{6.30}$$

the solution of which is

$$\frac{\alpha_+ - \alpha_-}{1-\alpha_+^2} + \frac{1}{2}\log\frac{\alpha_+ + 1}{\alpha_+ - 1} - \frac{1}{2\lambda_0^2}\log\frac{\alpha_- + 1}{\alpha_- - 1} = \frac{c_G}{4k_1}\log \mu/\mu_{\alpha_0}\,. \tag{6.31}$$

As illustrated in fig. 6.1, this system displays a finite RG trajectory linking fixed points. In the UV limit the poles accumulate to different zeroes, and in the IR the poles accumulate to the same zero. Let us consider the upper red trajectory of fig. 6.1 in which we choose $\lambda_0 < 1$ and pick the positive branch of the solution $\alpha_+ = +\sqrt{1 - \lambda_0^2(1-\alpha_-^2)}$. With this choice we see that there are finite fixed points[29] such that the right hand side of eq. (6.30) vanishes at

$$\text{UV}:\quad (\alpha_-, \alpha_+) = (-1, 1)\,, \quad \lambda_1 = 0\,, \qquad \text{IR}:\quad (\alpha_-, \alpha_+) = (1, 1)\,, \quad \lambda_1 = \lambda_0\,, \tag{6.32}$$

in which we recall the map

$$\lambda_1 = \left(\frac{(1+\alpha_-)(-1+\alpha_+)}{(-1+\alpha_-)(1+\alpha_+)}\right)^{\frac{1}{2}}\ . \tag{6.33}$$

One of the appealing features of the IFT$_2$ (5.3) is that it provides a classical Lagrangian interpolation that includes its own UV and IR limits [20]. That is to say these CFTs can be obtained directly from the Lagrangian eq. (5.3) by tuning the parameters of the theory to their values at the end points of the RG flow. Given the interpretation of these RG flows as describing poles colliding with zeroes it is natural to expect that a similar interpolation can be obtained directly in 4d by taking limits of the differential $\omega$ in eq. (6.17).

Here we will explore how this works for the IFT$_2$ (5.3) in the IR. The limit we will consider is to collide the poles at $\alpha_\pm$ with the zero at $\gamma_+$, following the upper red trajectory in fig. 6.1. This corresponds to taking $q_\pm \to 0$. In order to be consistent with the RG invariants (6.26), we take this limit as

$$q_+ = k_1\epsilon + \mathcal{O}(\epsilon^2)\,, \qquad q_- = k_2\epsilon + \mathcal{O}(\epsilon^2)\,, \qquad K = k_1 - k_2 + \mathcal{O}(\epsilon)\,, \qquad \epsilon \to 0\,. \tag{6.34}$$

---

[29]There are also fixed points to the RG flow at $\alpha_+ = 0$ with $\alpha_-^2 = 1 - \frac{k_2}{k_1}$ however by assumption $k_2 > k_1$, and so these do not correspond to real values of $\alpha_-$ and consequently $\lambda_1$ is imaginary. We do not consider such complex limits here.

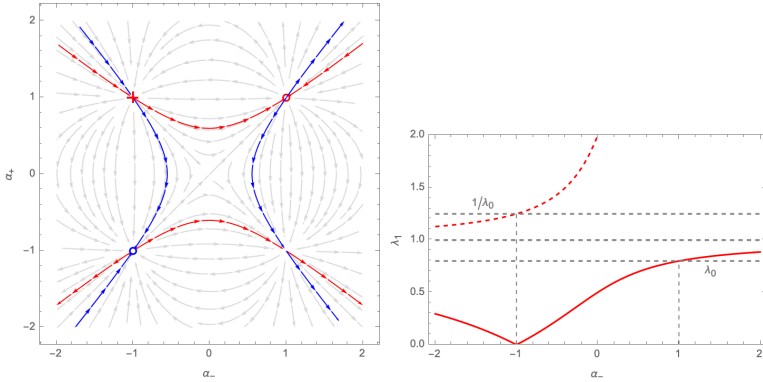

**Figure 6.1:** Left: RG flow across the $\alpha_+$, $\alpha_-$ plane with arrows directed to the IR. The highlighted parabola are the solutions that lie on the locus of the RG invariant quantity $\lambda_0^2 = k_1/k_2$, plotted here for $\lambda_0 = 0.8$ (red) and $\lambda_0 = 1.2$ (blue). Right: The value of $\lambda_1$ plotted along the red loci of the left panel (upper branch solid and lower branch dotted). In both cases $\lambda_1 \to 1$ asymptotically as $\alpha_- \to \pm\infty$. Of note is the flow displayed by the upper red branch between the UV fixed point $(\alpha_-, \alpha_+) = (-1, 1)$ with $\lambda_1 = 0$ and the IR fixed point $(\alpha_-, \alpha_+) = (1, 1)$ with $\lambda_1 = \lambda_0$.

Taking this limit in (6.17), and redefining the spectral parameter such that the remaining pole and zero are fixed to 1 and $-1$ respectively, yields

$$\omega \to \frac{k_1 - k_2}{2} \frac{\zeta + 1}{\zeta - 1} \, d\zeta \; . \tag{6.35}$$

Let us consider the implication of this limit from the $CS_4$ perspective. Given that the pole structure of $\omega$ is modified in this limit, so will the double $\mathfrak{d}$, and thus we should be careful in our interpretation of the boundary conditions. If we take $\omega$ to be given by (6.35) and consider the boundary conditions (6.20) with $t = 0$, the condition $\mathbb{A}_w \in \mathfrak{l}_t$ becomes $\hat{\mathcal{A}}_w|_\alpha = 0$. From eq. (4.5) we know that $\hat{\mathcal{A}}_w$ has a pole at $\gamma_+$. In other words, we can write

$$\hat{\mathcal{A}}_w = \frac{(\zeta - \alpha_+)}{(\zeta - \gamma_+)} \Xi(\zeta) \; , \tag{6.36}$$

with $\Xi(\zeta)$ regular as $\alpha_+ \to \gamma_+$. Hence, in the IR there is no boundary condition for $\hat{\mathcal{A}}_w$ at $\zeta = 1$. On the other hand, the boundary condition $\mathbb{A}_{\bar{w}} \in \mathfrak{l}_{t^{-1}}$ for $t = 0$ is $\hat{\mathcal{A}}_{\bar{w}}|_{\bar{\alpha}} = 0$ which in the limit $\alpha_+, \gamma_+ \to 1$ becomes a chiral boundary condition for the $\bar{w}$ component

$$\hat{\mathcal{A}}_{\bar{w}}|_{\zeta=1} = 0 \; . \tag{6.37}$$

For this choice of boundary condition one can localise the $CS_4$ action following the procedure described in either § 4 or appendix B, and the resulting two dimensional IFT is the WZW model at level $k_2 - k_1$.

In contrast, from the 2-dimensional perspective it is known that the full result at this IR fixed point is actually a product WZW model on $G_{k_2} \times G_{k_1 - k_2}$ [20]. This indicates that there is some delicacy in taking the IR limit directly as a Lagrangian interpolation in 4d even when it is possible in 2d. One reason for this is there is also the freedom to perform redefinitions of the spectral parameter, which can, in general, produce non-equivalent limits of $\omega$. Such limits are known as decoupling limits [37, 35] in the literature, and have been investigated for the UV fixed point of the bi-Yang-Baxter model in [38].

## 7 Discussion and Outlook

In this work we have constructed a diamond of integrable models related by localisation and symmetry reduction. Starting from holomorphic Chern-Simons theory with the meromorphic (3,0)-form (2.2), we

have found a new choice of admissible boundary conditions, which leads to a well-defined 6-dimensional theory. This generalises the analysis carried out in [11, 14] to a new class of boundary conditions not of Dirichlet type.

By first viewing twistor space, $\mathbb{PT}$, as a $\mathbb{CP}^1$ bundle over $\mathbb{E}^4$, we solved the equations of motion along the $\mathbb{CP}^1$ fibres. In doing so we fully specified the dependence of the integrand on the $\mathbb{CP}^1$ fibre and thus could perform a fibrewise integration along those directions. Consequently our 6-dimensional theory then localised to the poles of $\Omega$, leading to a new 4-dimensional theory on $\mathbb{E}^4$ given by the action (3.16). Indeed, this 4-dimensional theory is 'integrable' in the sense that its equations of motion can be encoded in an anti-self-dual connection, as expected from the Penrose-Ward correspondence. Moreover, this new $\mathrm{IFT}_4$ exhibits two semi-local symmetries, which can be understood as the residual symmetries preserving the boundary conditions of $\mathrm{hCS}_6$. For each of these semi-local symmetries, the Noether currents can be used to construct inequivalent Lax formulations of the dynamics.

On the other hand, symmetry reducing $\mathrm{hCS}_6$ along two directions of the $\mathbb{E}^4$ in $\mathbb{PT} \cong \mathbb{CP}^1 \times \mathbb{E}^4$, leads to an effective $\mathrm{CS}_4$ theory on $\mathbb{CP}^1 \times \mathbb{E}^2$. Under this procedure, the meromorphic $(3,0)$-form reduces to the meromorphic $(1,0)$-form used in [9] to construct the $\lambda$-model, whereas the 6-dimensional boundary conditions reduce to a class of boundary conditions in $\mathrm{CS}_4$ that have not been previously considered. Specifically, they relax the assumption of an isotropic subalgebra of the defect algebra. By performing the standard localisation procedure of $\mathrm{CS}_4$ we obtain the 2-field $\lambda$-type $\mathrm{IFT}_2$ introduced in [20].

Notably, this same multi-parametric class of integrable $\lambda$-deformations between coupled WZW models can be obtained by symmetry reduction (along the same directions) of the novel $\mathrm{IFT}_4$ mentioned above. Furthermore, the semi-local symmetries of the $\mathrm{IFT}_4$ reduce to global symmetries of the $\mathrm{IFT}_2$ and the two Lax formulations of the $\mathrm{IFT}_4$ give rise to two Lax connections for the $\mathrm{IFT}_2$. When the directions of the symmetry reduction are aligned to these semi-local symmetries, the $\mathrm{IFT}_2$ symmetries are enhanced to either affine or fully local (gauge) symmetries. In the latter case, the $\mathrm{IFT}_2$ becomes the standard (1-field) $\lambda$-model.

This work opens up a range of interesting further directions. There are a selection of direct generalisations that can be made to incorporate the wide variety of integrable deformations known in the literature. Perhaps the most interesting outcome of this would be the construction of swathes of new four-dimensional integrable field theories. Our work focused on the case where $\Omega$ was nowhere vanishing; it would be interesting to explore the relaxation of this condition together with its possible boundary conditions, and how the ASDYM equations are modified. Moreover, one might hope that the study of boundary conditions in $\mathrm{hCS}_6$ could lead to a full classification of the landscape of integrable sigma-models in 2d, and perhaps result in theories not yet encountered in the literature.

From the perspective of the $\mathrm{IFT}_2$, there is a close relationship between the notions of Poisson-Lie symmetry, duality and integrability [39, 40, 41, 42]. This poses an interesting question as to the implications of such dualities for both the $\mathrm{IFT}_4$ and $\mathrm{hCS}_6$. For the model considered here, we might seek to understand the semi-local symmetries of $\mathrm{IFT}_4$ in the context of the $q$-deformed symmetries expected to underpin the $\mathrm{IFT}_2$.

In this work, the integrable models we have studied can be viewed as descending from the open string sector of a type B topological string. An interesting direction for future work is to consider the closed string sector [43] and its possible integrable descendants. A tantalising prospect is to understand the closed string counterparts of the integrable deformations we have considered in the context of the non-linear graviton construction for self-dual space-times [44, 45, 46].

By coupling the open and closed string sectors [47, 48, 49, 50] one can find an anomaly free quantization to all loop orders in perturbation theory of the coupled $\mathrm{hCS}_6$-BCOV action. This mechanism has already

proven a powerful tool in the context of the top-down approach to celestial holography [13, 51]. This could provide an angle of attack to address the important questions of when the IFT$_4$ can be quantised, if the IFT$_4$ is *quantum* integrable, and if there is a higher dimensional origin of IFT$_2$ as quantum field theories.

# Acknowledgements

The authors would like to thank Roland Bittleston, Mathew Bullimore, Kevin Costello, Simon Heuveline, Sylvain Lacroix, Sean Seet, Konstantinos Sfetsos and Benoit Vicedo for interesting discussions. L.T.C. would especially like to thank Roland Bittleston and Sean Seet for their generosity in answering numerous questions on the subtleties of twistor space, and Kevin Costello for discussion on residual symmetries in holomorphic Chern-Simons theory. J.L. would also like to thank Horacio Falomir and Pablo Pisani for their endless support as PhD advisors. The work of B.H. was supported by a UKRI Future Leaders Fellowship (grant number MR/T018909/1). The work of J.L. is supported by CONICET. D.C.T. is supported by The Royal Society through a University Research Fellowship Generalised Dualities in String Theory and Holography URF 150185 and in part by STFC grant ST/X000648/1.

# A Conventions

## A.1 Twistor Space and Homogeneous Coordinates

We begin by reviewing the construction of twistor space originally introduced by Penrose [52, 53, 54]. For further references see [55, 56]. The projective spin bundle $\mathbb{CP}^1 \hookrightarrow \mathbb{PS}_\mathbb{C} \twoheadrightarrow \mathbb{C}^4$ can be trivialised with coordinates $x^{AA'} \in \mathbb{C}^4$ and $\pi_{A'} \in \mathbb{CP}^1$ where this is defined relative to the equivalence relation $\pi_{A'} \sim r\pi_{A'}$ for $r \in \mathbb{C}\backslash\{0\}$ and in which indices run over $A \in \{1, 2\}$ and $A' \in \{1', 2'\}$.

Twistor space $\mathbb{PT}_\mathbb{C}$ is defined to be the image of $\mathbb{PS}_\mathbb{C}$ under the projection

$$p : \mathbb{PS}_\mathbb{C} \to \mathbb{CP}^3 , \quad \mathbb{PT}_\mathbb{C} := p(\mathbb{PS}_\mathbb{C}) \subset \mathbb{CP}^3 ,$$
$$p : (x^{AA'}, \pi_{A'}) \mapsto (\omega^A, \pi_{A'}) = (x^{AA'}\pi_{A'}, \pi_{A'}) . \tag{A.1}$$

Here, we are using homogeneous coordinates $Z^\alpha = (\omega^A, \pi_{A'}) \in \mathbb{CP}^3$ defined up to the equivalence relation $Z^\alpha \sim rZ^\alpha$ for $r \in \mathbb{C}\backslash\{0\}$. Notice that the image of $\mathbb{PS}_\mathbb{C}$ covers all of $\mathbb{CP}^3$ save for a $\mathbb{CP}^1$ worth of points defined by $\pi_{A'} = (0, 0)$. Hence

$$\mathbb{PT}_\mathbb{C} = \mathbb{CP}^3 \setminus \mathbb{CP}^1 = \left\{ Z^\alpha = (\omega^A, \pi_{A'}) \mid Z^\alpha \sim rZ^\alpha , \ \pi_{A'} \neq (0, 0) \right\} . \tag{A.2}$$

In this work we shall be predominantly interested in the restriction to Euclidean twistor space $\mathbb{PT}_\mathbb{E}$. This follows similarly to above but now starting with the Euclidean projective spin bundle obtained by taking a real slice of the base $\mathbb{C}^4$. On this slice, we can be more explicit about the base manifold coordinates $x^{AA'} \in \mathbb{E}^4$, which can be written in terms of four real variables $x^0, x^2, x^2, x^3 \in \mathbb{R}$ as

$$\mathbb{E}^4 \ : \quad x^{AA'} = \frac{1}{\sqrt{2}} \begin{pmatrix} x^0 + \mathrm{i}x^1 & \mathrm{i}x^3 - x^2 \\ \mathrm{i}x^3 + x^2 & x^0 - \mathrm{i}x^1 \end{pmatrix} . \tag{A.3}$$

Our choice of orientation is such that $\mathrm{vol}_4 = \mathrm{d}x^0 \wedge \mathrm{d}x^1 \wedge \mathrm{d}x^2 \wedge \mathrm{d}x^3$. For this special case of Euclidean signature,[30] the image of the projection map, hence the twistor space is unchanged,

$$\mathbb{PT}_\mathbb{E} = \mathbb{CP}^3 \setminus \mathbb{CP}^1 = \left\{ Z^\alpha = (\omega^A, \pi_{A'}) \mid Z^\alpha \sim rZ^\alpha , \ \pi_{A'} \neq (0, 0) \right\} . \tag{A.4}$$

---

[30]For brevity, we suppress the subscript $\mathbb{E}$ in the main document where we will always work in Euclidean signature, unless specified otherwise.

Furthermore, for Euclidean space the projection map is invertible, hence $\mathbb{PT}_\mathbb{E}$ and $\mathbb{PS}_\mathbb{E}$ are diffeomorphic as real manifolds. Importantly, the complex structure on $\mathbb{PT}_\mathbb{E}$ inherited from being a subset of $\mathbb{CP}^3$ is not trivially expressed in terms of the coordinates on $\mathbb{PS}_\mathbb{E}$. Nonetheless, we are free to use two choices of coordinates on $\mathbb{PT}_\mathbb{E}$, viewing it either as a subset of $\mathbb{CP}^3$, or as diffeomorphic to $\mathbb{PS}_\mathbb{E} = \mathbb{E}^4 \times \mathbb{CP}^1$.

We will now give this diffeomorphism concretely by introducing additional notation for the $\mathbb{CP}^3$ coordinates on $\mathbb{PT}_\mathbb{E}$. We define the quaternionic conjugation operation (denoted by $\hat{\cdot}$) to be

$$
\begin{aligned}
\omega^A = (\omega^1, \omega^2) &\mapsto \hat{\omega}^A = (-\overline{\omega^2}, \overline{\omega^1}) \ , \\
\pi_{A'} = (\pi_{1'}, \pi_{2'}) &\mapsto \hat{\pi}_{A'} = (-\overline{\pi_{2'}}, \overline{\pi_{1'}}) \ .
\end{aligned}
\tag{A.5}
$$

Notice that this operation squares to $-1$ so must be applied four times to return to the original value, hence the name.

We also define inner products given by

$$
\begin{aligned}
\|\omega\|^2 = [\omega\hat{\omega}] = \omega^A \hat{\omega}_A = \varepsilon_{AB}\omega^A\hat{\omega}^B = \varepsilon^{AB}\omega_B\hat{\omega}_A \ , \\
\|\pi\|^2 = \langle\pi\hat{\pi}\rangle = \pi^{A'}\hat{\pi}_{A'} = \varepsilon_{A'B'}\pi^{A'}\hat{\pi}^{B'} = \varepsilon^{A'B'}\pi_{B'}\hat{\pi}_{A'} \ .
\end{aligned}
\tag{A.6}
$$

These may be explicitly written in terms of the components as

$$
\begin{aligned}
\|\omega\|^2 = \omega^1\overline{\omega^1} + \omega^2\overline{\omega^2} = \omega_1\overline{\omega_1} + \omega_2\overline{\omega_2} \ , \\
\|\pi\|^2 = \pi_{1'}\overline{\pi_{1'}} + \pi_{2'}\overline{\pi_{2'}} = \pi^{1'}\overline{\pi^{1'}} + \pi^{2'}\overline{\pi^{2'}} \ .
\end{aligned}
\tag{A.7}
$$

Here, we are using the raising and lowering conventions

$$
\begin{aligned}
\omega^A = \varepsilon^{AB}\omega_B \ , \qquad \omega_A = \varepsilon_{AB}\omega^B \ , \qquad \varepsilon^{AC}\varepsilon_{CB} = \delta^A_B \ , \\
\pi^{A'} = \varepsilon^{A'B'}\pi_{B'} \ , \quad \pi_{A'} = \varepsilon_{A'B'}\pi^{B'} \ , \quad \varepsilon^{A'C'}\varepsilon_{C'B'} = \delta^{A'}_{B'} \ .
\end{aligned}
\tag{A.8}
$$

Explicitly, the values of the anti-symmetric tensors are chosen to be

$$
\varepsilon_{12} = +1 \ , \quad \varepsilon_{1'2'} = +1 \ , \quad \varepsilon^{12} = -1 \ , \quad \varepsilon^{1'2'} = -1 \ .
\tag{A.9}
$$

Using these conventions, the action of the conjugation on the inverted indices is

$$
\hat{\omega}_A = \left(-\overline{\omega_2}, \overline{\omega_1}\right) \ , \quad \hat{\pi}^{A'} = \left(-\overline{\pi^{2'}}, \overline{\pi^{1'}}\right) \ .
\tag{A.10}
$$

Now, with this notation in place, we can explicitly construct the diffeomorphism from $\mathbb{PT}_\mathbb{E}$ to $\mathbb{PS}_\mathbb{E}$. The inverse of the projection map is given by

$$
\begin{aligned}
p^{-1} &: \mathbb{PT}_\mathbb{E} \to \mathbb{PS}_\mathbb{E} \ , \\
p^{-1} &: (\omega^A, \pi_{A'}) \mapsto (x^{AA'}, \pi_{A'}) = \left( \frac{\hat{\omega}^A\pi^{A'} - \omega^A\hat{\pi}^{A'}}{\|\pi\|^2}, \pi_{A'} \right) \ .
\end{aligned}
\tag{A.11}
$$

When doing calculations on Euclidean twistor space, we have two natural choices of homogeneous coordinates. We can view $\mathbb{PT}_\mathbb{E}$ as a subspace of $\mathbb{CP}^3$ and work with the coordinates $Z^\alpha = (\omega^A, \pi_{A'})$ defined up to the equivalence relation $Z^\alpha \sim rZ^\alpha$ for $r \in \mathbb{C}\backslash\{0\}$. Alternatively, we can view $\mathbb{PT}_\mathbb{E}$ as diffeomorphic to $\mathbb{PS}_\mathbb{E} = \mathbb{E}^4 \times \mathbb{CP}^1$ and use the coordinates $(x^{AA'}, \pi_{A'})$ defined up to the equivalence relation $\pi_{A'} \sim r\pi_{A'}$ for $r \in \mathbb{C}\backslash\{0\}$.

## A.2  Basis of Forms and Vector Fields

Thinking of $\mathbb{PT}_\mathbb{E}$ as diffeomorphic to $\mathbb{PS}_\mathbb{E} = \mathbb{E}^4 \times \mathbb{CP}^1$ and using the coordinates $(x^{AA'}, \pi_{A'})$, we can define a basis of 1-forms by

$$
\begin{aligned}
e^0 = \langle\pi\mathrm{d}\pi\rangle \ , && e^A = \pi_{A'}\mathrm{d}x^{AA'} \ , \\
\bar{e}^0 = \frac{\langle\hat{\pi}\mathrm{d}\hat{\pi}\rangle}{\langle\pi\hat{\pi}\rangle^2} \ , && \bar{e}^A = \frac{\hat{\pi}_{A'}\mathrm{d}x^{AA'}}{\langle\pi\hat{\pi}\rangle} \ .
\end{aligned}
\tag{A.12}
$$

This basis of 1-forms is split into the $(1,0)$-forms and $(0,1)$-forms with respect to the complex structure inherited from $\mathbb{CP}^3$. They should also be understood as being valued in line bundles, since although they are normalised to have weight zero under the rescaling $\hat\pi_{A'} \sim \bar r \hat\pi_{A'}$, they carry weight under $\pi_{A'} \sim r\pi_{A'}$.

The dual basis of vector fields is given by

$$\partial_0 = \frac{\hat\pi_{A'}}{\langle\pi\hat\pi\rangle}\frac{\partial}{\partial\pi_{A'}} \;, \qquad\qquad \partial_A = -\frac{\hat\pi^{A'}\partial_{AA'}}{\langle\pi\hat\pi\rangle} \;,$$
$$\bar\partial_0 = -\langle\pi\hat\pi\rangle\,\pi_{A'}\frac{\partial}{\partial\hat\pi_{A'}} \;, \qquad\qquad \bar\partial_A = \pi^{A'}\partial_{AA'} \;. \tag{A.13}$$

This basis of 1-forms, and their duals, enjoy the structure equations,

$$\bar\partial e^A = e^0 \wedge \bar e^A \;, \quad \partial\bar e^A = e^A \wedge \bar e^0 \;,$$
$$[\bar\partial_0\,,\partial_A] = \bar\partial_A \;, \quad [\bar\partial_A\,,\partial_0] = \partial_A \;. \tag{A.14}$$

## A.3   Hyper-Kähler Structure

Recall that $\mathbb{E}^4$ can be given a hyper-Kähler structure consisting of a triplet $\mathcal{J}_i$, $i = 1,2,3$ of complex structures obeying $\mathcal{J}_i^2 = -\mathrm{id}$ and $\mathcal{J}_1\mathcal{J}_2\mathcal{J}_3 = \mathrm{id}$. For each $\mathcal{J}_i$ the corresponding Kähler forms $\varpi_i$ are self-dual with respect to the metric $ds^2 = \sum_{\mu=0}^3 dx^\mu \otimes dx^\mu$ and may be given explicitly as

$$\varpi_i = \frac{1}{2}\,\epsilon_{ijk}dx^j \wedge dx^k + dx^0 \wedge dx^i \;. \tag{A.15}$$

For completeness, in the coordinate basis $\{dx^\mu\}$, $\mu = 0\ldots3$, i.e. $\mathcal{J}_idx^\nu = (\mathcal{J}_i)_\mu{}^\nu dx^\mu$, the components of the complex structures are given as

$$(\mathcal{J}_1)_\mu{}^\nu = \begin{pmatrix} 0 & 1 & 0 & 0 \\ -1 & 0 & 0 & 0 \\ 0 & 0 & 0 & 1 \\ 0 & 0 & -1 & 0 \end{pmatrix} \;, \quad (\mathcal{J}_2)_\mu{}^\nu = \begin{pmatrix} 0 & 0 & 1 & 0 \\ 0 & 0 & 0 & -1 \\ -1 & 0 & 0 & 0 \\ 0 & 1 & 0 & 0 \end{pmatrix} \;, \quad (\mathcal{J}_3)_\mu{}^\nu = \begin{pmatrix} 0 & 0 & 0 & 1 \\ 0 & 0 & 1 & 0 \\ 0 & -1 & 0 & 0 \\ -1 & 0 & 0 & 0 \end{pmatrix} \;.$$

Taking combinations of these $\mathcal{J}_i$, we find a space of complex structures parameterised by $\pi_{A'} \sim (1,\zeta) \in \mathbb{CP}^1$,

$$\mathcal{J}_\pi = \frac{\zeta + \bar\zeta}{1 + \zeta\bar\zeta}\,\mathcal{J}_3 - \frac{\mathrm{i}(\bar\zeta - \zeta)}{1 + \zeta\bar\zeta}\,\mathcal{J}_2 + \frac{1 - \zeta\bar\zeta}{1 + \zeta\bar\zeta}\,\mathcal{J}_1 \;, \qquad \mathcal{J}_\pi^2 = -\mathrm{id}\,. \tag{A.16}$$

We can also define this $\mathbb{CP}^1$-dependent complex structure in spinor notation with $\pi_{A'} \sim (1,\zeta)$ as

$$\mathcal{J}_\pi = -\frac{\mathrm{i}}{\|\pi\|^2}\,(\pi^{A'}\hat\pi_{B'} + \hat\pi^{A'}\pi_{B'})\,\partial_{AA'} \otimes \mathrm{d}x^{AB'} = \mathrm{i}\,\partial_A \otimes e^A - \mathrm{i}\,\bar\partial_A \otimes \bar e^A \;. \tag{A.17}$$

Thinking of $\mathbb{PT}_\mathbb{E}$ as the bundle of complex structures over $\mathbb{E}^4$, the coordinates $v^A = \pi_{A'}x^{AA'}$ are holomorphic coordinates with respect to the complex structure $\mathcal{J}_\pi$. We may denote these coordinates by $v^A = (z_\pi, w_\pi)$. The Kähler form corresponding to $\mathcal{J}_\pi$ can equally be expressed as

$$\varpi_\pi = \frac{1}{2}\,(\mathcal{J}_\pi)_{AA'BB'}\,\mathrm{d}x^{AA'} \wedge \mathrm{d}x^{BB'} = -\frac{\mathrm{i}}{\|\pi\|^2}\,\varepsilon_{AB}\,\pi_{A'}\hat\pi_{B'}\,\mathrm{d}x^{AA'} \wedge \mathrm{d}x^{BB'}$$
$$= -\frac{\mathrm{i}}{1 + \zeta\bar\zeta}\,(\mathrm{d}z_\pi \wedge \mathrm{d}\bar z_\pi + \mathrm{d}w_\pi \wedge \mathrm{d}\bar w_\pi) \;. \tag{A.18}$$

Holomorphic self-dual $(2,0)$- and $(0,2)$-forms are given by

$$\mu_\pi = \varepsilon_{AB}\,\pi_{A'}\pi_{B'}\,\mathrm{d}x^{AA'} \wedge \mathrm{d}x^{BB'} = 2\,\mathrm{d}z_\pi \wedge \mathrm{d}w_\pi \;,$$
$$\bar\mu_\pi = \varepsilon_{AB}\,\hat\pi_{A'}\hat\pi_{B'}\,\mathrm{d}x^{AA'} \wedge \mathrm{d}x^{BB'} = 2\,\mathrm{d}\bar z_\pi \wedge \mathrm{d}\bar w_\pi \;. \tag{A.19}$$

# B  Alternative Localisation of CS$_4$ to IFT$_2$

In §6 the defect algebra approach to 4d CS (for a complete discussion see [29, 30]) was utilised in the passage to the 2d IFT. In this appendix we will perform the localisation of 4d CS to the 2d IFT via a more standard route, mirroring the localisation approach from 6d hCS to the 4d IFT, showing that we also land on (5.3).

We recall the action (4.9)

$$S_{\mathrm{CS}_4} = \frac{1}{2\pi i} \int_{\Sigma \times \mathbb{CP}^1} \omega \wedge \mathrm{Tr}\left( \hat{\mathcal{A}} \wedge \mathrm{d}\hat{\mathcal{A}} + \frac{2}{3}\hat{\mathcal{A}} \wedge \hat{\mathcal{A}} \wedge \hat{\mathcal{A}} \right) , \tag{B.1}$$

with the meromorphic one form of eq. (4.10) and boundary conditions as per eq. (4.7).

We work with the parametrisation of $\hat{\mathcal{A}}$ of eq.(6.1), recalled here for convenience

$$\hat{\mathcal{A}}_{\bar{\zeta}} = \hat{h}^{-1}\partial_{\bar{\zeta}}\hat{h} , \quad \hat{\mathcal{A}}_I = \hat{h}^{-1}\mathcal{L}_I\hat{h} + \hat{h}^{-1}(\partial_I\hat{h}) , \quad I = w, \bar{w} . \tag{B.2}$$

Viewing $\hat{\mathcal{A}} = \mathcal{L}^{\hat{h}}$ as the formal gauge transform of $\mathcal{L}$ by $\hat{h}$, we use the following identity satisfied by the Chern-Simons density

$$\mathrm{CS}(\hat{\mathcal{A}}) = \mathrm{CS}(\mathcal{L}^{\hat{h}}) = \mathrm{CS}(\mathcal{L}) - \mathrm{d}\,\mathrm{Tr}\left(\hat{J} \wedge \hat{h}^{-1}\mathcal{L}\,\hat{h}\right) - \frac{1}{6}\mathrm{Tr}\left(\hat{J} \wedge [\hat{J}, \hat{J}]\right) , \tag{B.3}$$

in which $\hat{J} = \hat{h}^{-1}\mathrm{d}\hat{h}$. Noting that on shell $\mathrm{CS}(\mathcal{L}) = \mathcal{L} \wedge \mathrm{d}\mathcal{L}$ and $\omega \wedge (\partial_{\zeta}\mathcal{L}) \wedge \mathcal{L} = 0$ we then arrive at the following action

$$S_{\mathrm{CS}_4} = -\frac{1}{2\pi i} \int \mathrm{d}\omega \wedge \mathrm{Tr}\left(\hat{J} \wedge \hat{h}^{-1}\mathcal{L}\,\hat{h}\right) - \frac{1}{12\pi i} \int \omega \wedge \left(\hat{J} \wedge [\hat{J}, \hat{J}]\right) . \tag{B.4}$$

In this form we see how our action will localise at the poles of $\omega$ giving a 2d theory

$$S = r_+ \int_{\Sigma} \mathrm{Tr}\left(\hat{J} \wedge \hat{h}^{-1}\mathcal{L}\,\hat{h}\right)|_{\alpha} + r_- \int_{\Sigma} \mathrm{Tr}\left(\hat{J} \wedge \hat{h}^{-1}\mathcal{L}\,\hat{h}\right)|_{\tilde{\alpha}} + \mathrm{WZ\ terms} , \tag{B.5}$$

where we recall

$$r_+ = K \frac{\langle\alpha\gamma\rangle\langle\alpha\hat{\gamma}\rangle}{\langle\alpha\tilde{\alpha}\rangle\langle\alpha\beta\rangle^2} \quad \text{and} \quad r_- = -K \frac{\langle\tilde{\alpha}\gamma\rangle\langle\tilde{\alpha}\hat{\gamma}\rangle}{\langle\alpha\tilde{\alpha}\rangle\langle\tilde{\alpha}\beta\rangle^2} .$$

To complete the construction we need to specify the meromorphic structure of $\mathcal{L}$ that ensures the theory is well defined given the form of $\omega$ and is compatible with the boundary conditions. This requires that

$$\mathcal{L}_w = \frac{\langle\pi\beta\rangle}{\langle\pi\gamma\rangle} \mathcal{M}_w + \mathcal{N}_w , \qquad \mathcal{L}_{\bar{w}} = \frac{\langle\pi\beta\rangle}{\langle\pi\hat{\gamma}\rangle} \mathcal{M}_{\bar{w}} + \mathcal{N}_{\bar{w}} , \tag{B.6}$$

where $\mathcal{M}_I, \mathcal{N}_I \in C^{\infty}(\Sigma, \mathfrak{g})$. The boundary conditions in this parametrisation read,

$$\begin{aligned}
\hat{h}|_{\beta} &= \mathrm{id} , \quad \mathcal{L}|_{\beta} = 0 , \\
\mathrm{Ad}_h^{-1}\mathcal{L}_w|_{\alpha} + J_w &= ts\left(\mathrm{Ad}_{\tilde{h}}^{-1}\mathcal{L}_w|_{\tilde{\alpha}} + \tilde{J}_w\right) , \\
\mathrm{Ad}_h^{-1}\mathcal{L}_{\bar{w}}|_{\alpha} + J_{\bar{w}} &= t^{-1}s\left(\mathrm{Ad}_{\tilde{h}}^{-1}\mathcal{L}_{\bar{w}}|_{\tilde{\alpha}} + \tilde{J}_{\bar{w}}\right) ,
\end{aligned} \tag{B.7}$$

in which we use the definitions,

$$\hat{h}|_{\alpha} = h , \qquad \hat{h}|_{\tilde{\alpha}} = \tilde{h} , \qquad \hat{J}|_{\alpha} = J , \qquad \hat{J}|_{\tilde{\alpha}} = \tilde{J} . \tag{B.8}$$

Solving these conditions uniquely determines the Lax connection

$$\begin{aligned}
\mathcal{M}_w &= \frac{\langle\alpha\gamma\rangle}{\langle\alpha\beta\rangle} \mathrm{Ad}_h \left[1 - \sigma\,\mathrm{Ad}_{\tilde{h}}^{-1}\mathrm{Ad}_h\right]^{-1}\left(ts\tilde{J}_w - J_w\right) , & \mathcal{N}_w &= 0 , \\
\mathcal{M}_{\bar{w}} &= \frac{\langle\alpha\hat{\gamma}\rangle}{\langle\alpha\beta\rangle} \mathrm{Ad}_h \left[1 - \sigma^{-1}\mathrm{Ad}_{\tilde{h}}^{-1}\mathrm{Ad}_h\right]^{-1}\left(t^{-1}s\tilde{J}_{\bar{w}} - J_{\bar{w}}\right) , & \mathcal{N}_{\bar{w}} &= 0 .
\end{aligned} \tag{B.9}$$

where we have introduced the parameter $\sigma = t\sqrt{\frac{\langle\alpha\gamma\rangle\langle\tilde{\alpha}\hat{\gamma}\rangle}{\langle\tilde{\alpha}\gamma\rangle\langle\alpha\hat{\gamma}\rangle}}$. It will also be useful to state the alternative forms

$$
\mathcal{M}_w = \frac{\langle\tilde{\alpha}\gamma\rangle}{\langle\tilde{\alpha}\beta\rangle}\,\mathrm{Ad}_{\tilde{h}}\left[1 - \sigma^{-1}\mathrm{Ad}_h^{-1}\mathrm{Ad}_{\tilde{h}}\right]^{-1}\left(t^{-1}s^{-1}J_w - \tilde{J}_w\right)\ ,
$$
$$
\mathcal{M}_{\bar{w}} = \frac{\langle\tilde{\alpha}\hat{\gamma}\rangle}{\langle\tilde{\alpha}\beta\rangle}\,\mathrm{Ad}_{\tilde{h}}\left[1 - \sigma\mathrm{Ad}_h^{-1}\mathrm{Ad}_{\tilde{h}}\right]^{-1}\left(ts^{-1}J_{\bar{w}} - \tilde{J}_{\bar{w}}\right)\ .
$$
(B.10)

Inserting (B.9) and (B.10) into (B.5) we obtain

$$
\begin{aligned}
S = {} & r_+ \int_\Sigma \mathrm{vol}_2\,\mathrm{Tr}\Big(J_w\left[1 - \sigma^{-1}\mathrm{Ad}_{\tilde{h}}^{-1}\mathrm{Ad}_h\right]^{-1}\left(t^{-1}s\tilde{J}_{\bar{w}} - J_{\bar{w}}\right)\Big) \\
& - r_+ \int_\Sigma \mathrm{vol}_2\,\mathrm{Tr}\Big(J_{\bar{w}}\left[1 - \sigma\mathrm{Ad}_{\tilde{h}}^{-1}\mathrm{Ad}_h\right]^{-1}\left(ts\tilde{J}_w - J_w\right)\Big) \\
& - r_- \int_\Sigma \mathrm{vol}_2\,\mathrm{Tr}\Big(\tilde{J}_w\left[1 - \sigma\mathrm{Ad}_h^{-1}\mathrm{Ad}_{\tilde{h}}\right]^{-1}\left(ts^{-1}J_{\bar{w}} - \tilde{J}_{\bar{w}}\right)\Big) \\
& + r_- \int_\Sigma \mathrm{vol}_2\,\mathrm{Tr}\Big(\tilde{J}_{\bar{w}}\left[1 - \sigma^{-1}\mathrm{Ad}_h^{-1}\mathrm{Ad}_{\tilde{h}}\right]^{-1}\left(t^{-1}s^{-1}J_w - \tilde{J}_w\right)\Big) \\
& + \text{WZ terms}\ ,
\end{aligned}
$$
(B.11)

where $\mathrm{vol}_2 = d\bar{w} \wedge dw$. Expanding out this action, collecting together terms, and Wick rotating to Minkowski space, we arrive at the action (5.3).

# C  Alternative CS$_4$ Setup for the $\lambda$-Model

In this section, we will consider an alternative symmetry reduction of our hCS$_6$ setup which also recovers the $\lambda$-deformed IFT$_2$. In order to recover a 1-field IFT$_2$, we need one of the semi-local residual symmetries of the IFT$_4$ to become a gauge symmetry under symmetry reduction. Let us denote the symmetry reduction vector fields by $V_1$ and $V_2$. Taking the example of the residual left-action parameterised by $\ell$, this must obey the constraint $\beta^{A'}\partial_{AA'}\ell = 0$. In order for this to become a gauge symmetry of the IFT$_2$, the symmetry reduction constraints $L_{V_1}\ell$ and $L_{V_2}\ell = 0$ must coincide with the pre-existing constraints on $\ell$. This means that we must choose to symmetry reduce along the vector fields

$$
V_1 = \mu^A\beta^{A'}\partial_{AA'}\ ,\qquad V_2 = \hat{\mu}^A\beta^{A'}\partial_{AA'}\ .
$$
(C.1)

Following the recipe described elsewhere in this paper, we deduce that the CS$_4$ 1-form is given by[31]

$$
\omega = K\,\frac{1}{(\zeta - \alpha_+)(\zeta - \alpha_-)}\,d\zeta\ .
$$
(C.2)

In the 4d CS description, we can already see that we have eliminated one degree of freedom relative to other symmetry reductions. The symmetry reduction zeroes have eliminated the double pole at $\beta$, effectively removing one field from the IFT$_2$.

Furthermore, if we denote the surviving coordinates on $\Sigma$ by $y^1 = \hat{\mu}^A\hat{\beta}^{A'}\partial_{AA'}$ and $y^2 = -\mu^A\hat{\beta}^{A'}\partial_{AA'}$, the boundary conditions reduce to

$$
\hat{\mathcal{A}}_1|_\alpha = \sigma\hat{\mathcal{A}}_1|_{\tilde{\alpha}}\ ,\qquad \hat{\mathcal{A}}_2|_\alpha = \sigma^{-1}\hat{\mathcal{A}}_2|_{\tilde{\alpha}}\ .
$$
(C.3)

Since the localisation from CS$_4$ to the IFT$_2$ has been described in detail elsewhere, we will be brief in this section. In the parametrisation

$$
\hat{\mathcal{A}} = \hat{h}^{-1}\mathcal{L}\hat{h} + \hat{h}^{-1}d\hat{h}\ ,
$$
(C.4)

---

[31] Since $\beta^{A'}$ appears in both of our symmetry reduction vector fields, the two zeroes from symmetry reduction have cancelled the double pole. Similarly, the boundary condition $\mathcal{A}_A|_\beta = 0$ can be interpreted as a simple zero in each component of the gauge field. These simple zeroes cancel the simple poles introduced in symmetry reduction, leaving a gauge field with no singularities.

we fix the constraints $\mathcal{L}_{\bar{\zeta}} = 0$ and denote the values of $\hat{h}$ at the poles by $\hat{h}|_\alpha = h$ and $\hat{h}|_{\tilde{\alpha}} = \mathrm{id}$. We can then use the bulk equations of motion and the boundary conditions to solve for $\mathcal{L}_1$ and $\mathcal{L}_2$ in terms of $h$. We find the solutions

$$\mathcal{L}_1 = (\sigma - \mathrm{Ad}_h^{-1})^{-1} h^{-1} \partial_1 h \,, \qquad \mathcal{L}_2 = (\sigma^{-1} - \mathrm{Ad}_h^{-1})^{-1} h^{-1} \partial_2 h \,. \tag{C.5}$$

Finally, the action localises to 2d and is given, up to an overall factor of $K/(\alpha_+ - \alpha_-)$, by

$$-\int_\Sigma \mathrm{d}y^1 \wedge \mathrm{d}y^2 \, \mathrm{Tr}\left( h^{-1}\partial_1 h \cdot \frac{1 + \sigma \mathrm{Ad}_h^{-1}}{1 - \sigma \mathrm{Ad}_h^{-1}} \, h^{-1}\partial_2 h \right) - \frac{1}{6}\int_{\Sigma \times [0,1]} \mathrm{Tr}\left( h^{-1}\mathrm{d}h \wedge h^{-1}\mathrm{d}h \wedge h^{-1}\mathrm{d}h \right) \,. \tag{C.6}$$

This can be recognised as the $\lambda$-deformed IFT$_2$.

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
