# Peer review of "Integrable Deformations from Twistor Space"

_SciPost Physics_

## Round 1 · Referee Report · Anonymous (Referee 1) · 2024-3-11

Report

This article studies the higher-dimensional origins of a familly of 2-dimensional integrable field theories (IFT$_2$), belonging to the class of $\lambda$-deformed sigma-models. In particular, it shows that these IFT$_2$ are part of a diamond of theories related by localisations and symmetry reductions, including a 6d holomorphic Chern-Simons theory on twistor space (hCS$_6$), a 4d semi-holomorphic Chern-Simons theory (CS$_4$) and a 4d integrable theory (IFT$_4$) whose equation of motion takes the form of an anti-self-dual Yang-Mills equation. This fits into a general scheme aiming at understanding IFT$_2$ from higher-dimensional gauge theories and in the context of the Penrose-Ward conjecture, which received a lot of attention, including in the recent years. In particular, the inclusion of $\lambda$-deformed sigma-models in this framework was a quite natural open question following various recent works (for instance the references [8,9,11,15] in the article).

The paper presents a quite thorough and systematic analysis of this diamond of theories and its underlying geometric structures. Doing so, the authors reveal new aspects which have been quite less explored so far in the literature, including the possibility of boundary conditions in CS$_4$ which are not based on isotropic subalgebras of the defect algebra. This opens up new avenues and raises interesting questions, for instance towards a complete understanding of the Lax connections and conserved charges of these models. In view of these points, it is my opinion that the results of this submission fit the criteria of originality and significance for publication in SciPost Physics.

Overall, I think the paper is quite clear and well-written. I have a few more specific questions and comments that I would like the authors to address. I explain the two main ones below. The remaining ones are listed in the section "Requested changes" and are essentially typos or small suggestions (that the authors should feel free to implement or not). I am happy to recommend this submission for publication once these minor revisions are addressed.

1) My first comment concerns the passage from CS$_4$ to IFT$_2$ in section 6. Indeed, it seems to me that there is a small inconsistency in the treatment of $\mathbf{h}$ (in this report, $\mathbf{h}$ denotes the field valued in the defect group $\mathbb{D}$). Relating equations (4.19), (6.1) and (6.2), one finds $\mathbf{h} = (\hat{h}|\alpha, \hat{h}|\tilde{\alpha})$, which according to (3.3) or (B.8) should then be $\mathbf{h} = (h, \tilde{h})$, rather than $\mathbf{h} = (h^{-1}, \tilde{h}^{-1})$ used in section 6. However, this should not affect the final results of the section, as I think $\mathbf{h}$ should also be replaced by $\mathbf{h}^{-1}$ in the rest of the E-model formulation. For instance, from (6.2), one gets $\mathbf{h}\partial_w \mathbf{h}^{-1} = \mathbb{L}w - \text{Ad}_w$}} \mathbb{A, with $\mathbb{A}_w$ in $\mathfrak{l}_t$. One then wants to construct the projector $W^+_{\mathbf{h}}$ such that $\mathbb{L}w \in \text{Im}(W^+)$} and $\text{Ad}{\mathbf{h}} \mathbb{A}_w \in \text{Ker}(W^+)$}. This requires $\text{Ker}(W^+{\mathbf{h}}) = \text{Ad}_t$}} \mathfrak{l, instead of $\text{Ad}_{\mathbf{h}}^{-1} \mathfrak{l}_t $ as considered in equation (6.4). Let me also note that this seems consistent with the reference [29] as well. Indeed, comparing the constraint (6.3) of this paper with the constraint on $B_\pm$ given above (3.28) in [29], one sees that the field $\mathbf{h}$ considered here corresponds to $h^{-1}$ in [29], so that one should shift $\mathbf{h}$ to $\mathbf{h}^{-1}$ throughout the E-model formulation.

2) My second point concerns the appendix B and the following two related questions : i) how does the term $\mathcal{M}_w / \langle \pi \gamma \rangle$ in the expression (B.6) of $\mathcal{L}_w$ descend from $\mathbb{C}^2$ to $\mathbb{CP}^1$, as it is of degree -1 in $\pi$ ? ii) why does this term not contribute at $\pi = \beta$ in $\mathcal{L}w|\beta=0$? It seems to me that a consistent form for $\mathcal{L}_w$ should instead be

$$ \mathcal{L}_w = \frac{\langle\pi\beta\rangle}{\langle\pi\gamma\rangle}\mathcal{M}_w + \mathcal{N}_w, $$
which would then ensure that $\pi$ can be seen in $\mathbb{CP}^1$ and that $\mathcal{L}w |\beta = \mathcal{N}_w=0$. Note that in that case, one would have an additional term $\langle \tilde{\alpha}\beta\rangle/\langle \alpha\beta \rangle$ in the coefficient of $\text{Ad}_{\tilde{h}}^{-1} \text{Ad}_h$ in (B.9): in fact , it seems this is necessary to match this coefficient with $\sigma$, using (4.8). Similar comments apply to $\mathcal{L}_{\bar{w}}$. Here also, this should not change the final results of the appendix.

Requested changes

1- First, I would like to ask the authors to address the comments 1) and 2) in the report above and, if relevant, implement the corresponding revisions in the manuscript.

The following points are small typos.

2- In (2.5), I think the $\wedge$ should be removed between $\mathcal{A}_A$ and $\delta\mathcal{A}_B$ as these are components of forms rather than forms. Also, there should be a vol$_4$.

3- After (3.27), there is a "be" missing in "to holomorphic"

4- After (3.41), there is a repetion of "that".

5- In the second line of (3.50), should the $\langle\tilde{\alpha}\beta\rangle$ be $\langle\alpha\beta\rangle$?

6- Before (4.6), hC6$_6$ instead of hCS$_6$

7- Should there be an overall minus sign in (4.14)?

8- Before (5.10), should it be "equations of motion of (5.3)"?

9- After (5.11), "contains implies"

10- In (5.15), should there be derivatives $\partial_\pm$ in addition to $\mathcal{L}^i_\pm$?

11- First paragraph of section 6: should "choice of isotropic subspace" be "choice of two mutually orthogonal susbpaces"?

12- In (6.4), $\xi$ should be $\alpha$

13- $c_G$ in (6.16) is never defined (although I imagine it is the dual Coxeter number)

14- Should there be $\pm$ signs in (6.23)?

15- Before (A.15), should $\mathcal{J}_1\mathcal{J}_2\mathcal{J}_3$ be $-\text{id}$?

Finally, the following are small suggestions which in my opinion could help the reader. The authors should feel free to implement them or not.

16- After (2.2), it could be useful to mention that $\alpha$, $\tilde{\alpha}$ and $\beta$ are constant spinors entering the definition of the model

17- Following footnote 9, the double pole of $\Omega$ at $\pi=\beta$ in principle creates terms in equations (3.7) and (3.16) which involve $\partial_z \hat{h}|_\beta$ or $\partial_z \mathcal{A}'|_\beta$, in addition to $\hat{h}|_\beta=\text{id}$ or $\mathcal{A}'|_\beta=0$ . It could be worth explaining why these do not contribute.

18- Two remarks concerning the passage from (3.34) to (3.35). As far as I understand, this is not completely direct as it uses the constraint (2.13) on $\ell$: it could be helpful to mention that. Also, it might be useful to recall that $A_{AA'}=\beta_{A'}\,B_A$ is not a new object introduced here and came from the analysis of $\mathcal{A}'$ before (3.9).

19- Around (3.47) and (3.49), it can be worth mentioning that $\gamma$ is the spinor that will appear later in the symmetry reduction.

20- Since there are many different parameterisations introduced throughout the paper to fit the various formulations / components of the diamond, it can be helpful to summarise around the equation (5.3) which ones can be taken as a set of independent physical parameters for the IFT$_2$, as for instance $(r_+,r_-,t,\sigma)$.

21- Around (6.2), it might be worth stating explicitly that $\mathbf{h}=(h|\alpha,h|)$} and $\mathbb{L}=(\mathcal{L}|\alpha,\mathcal{L}|)$}.

---

## Round 1 · Referee Report · Anonymous (Referee 2) · 2024-3-19

# Report on *Integrable Deformations from Twistor Space*

In his 1985 paper titled 'Integrable and Solvable Systems, and Relations Among Them' Ward conjectured that many integrable differential equations arise as symmetry reductions of the anti-self-dual Yang-Mills (ASDYM) equations. Since then, an alternative organising principle for 2d integrable models has emerged: 4d Chern-Simons theory (CS4). Building on work of Costello, Bittleston and Skinner proposed that these two approaches might be related via a holomorphic Chern-Simons (HCS) theory on twistor space. The authors investigate this proposal in the context of $\lambda$-models: integrable deformations of the 2d Wess-Zumino-Witten (WZW) model.

They begin by identifying alternative boundary conditions for HCS on twistor space. Implementing the descent to space-time via the Penrose-Ward transformation, they obtain a novel 4d integrable field theory (IFT) depending on two group valued fields. Much like the 4d WZW model, this has two semi-local symmetries and its equations of motion follow from the ASDYM equations for the Lax. Reducing HCS by translations along a non-degenerate 2-plane gives CS4 with an unconventional set of boundary conditions and poles: the boundary values of the connection do not lie in an isotropic subalgebra of the defect algebra. Descending in the familiar way to a 2d IFT gives a multi-parameter family of $\lambda$-deformations for coupled WZW models. This model has been obtained from CS4 previously, although the description in the paper is novel. Reducing directly on space-time lands on the same 2d IFT. By adapting the translation group to the boundary conditions, it specialises to the $\lambda$-model. Finally, the authors consider RG flow of these 2d integrable models, verifying conjectures of Costello regarding the evolution of the meromorphic (1,0)-form $\omega$.

This work represents an important contribution to a burgeoning field of study concerning integrable models in 4 space-time dimensions. Furthermore, it raises the possibility that a wider class of boundary conditions in CS4 are viable, doubtless worth further investigation. I'm happy to say that in my view the paper is clear and effectively written, and it certainly meets the SciPost criterion for originality and significance. I wholeheartedly recommend publication.

I have a few comments which the authors may wish to address:

- Although the equations of motion (EoM) of the two field 4d IFT follow from the ASDYM equations for a suitable Lax, it's not clear to me whether they arise as a partial gauge fixing. If so, this would seem to confirm that the (EoM of the) $\lambda$-deformation is an instance of the Ward conjecture.

- Above equation (3.6) the field $\hat{h}$ is extended to $\mathbb{PT} \times [0,1]$, presumably so that its restriction to $\mathbb{PT} \times \{0\}$ coincides with $\hat{h}$. Certainly further assumptions on this extension are needed. From context it seems natural to choose a smooth homotopy to a constant map. Alternatively, fixing an archipelago type gauge on $\hat{h}$ (if attainable) will supply independent homotopies to the constant map for $h, \tilde{h}$.

- I believe section 3.3 on reality conditions and parameters might be improved by including a twistor interpretation. For example: it's immediately clear that Euclidean reality conditions are incompatible with the twistor description, since $\Omega$ has a single double pole which cannot be paired with a double pole at an antipodal point. To obtain natural reality conditions on the fields one can work in split signature. Whilst $\beta$ needs to be real, there's freedom in the reality conditions on $(\alpha, \tilde{\alpha})$ and $(\mu, \tilde{\mu})$. $\tilde{\alpha} = \bar{\alpha}$ seems like a particularly interesting case, as twisting by the anti-involution of $G^{\mathbb{C}}$ corresponding to the real form $G$ implies that $\bar{h} = \tilde{h}$. I expect this reduces the complex two-field model to a real one-field model for a single $G^{\mathbb{C}}$ valued field.

I also have a couple of minor points:

- In footnote 7 reference is made to a 5d Chern-Simons theory, which could refer to a few different models. It might be less ambiguous if referred to as 5d Kähler Chern-Simons.

- I think footnote 8 would be clearer if it read 'More generally, a manifold whose boundary is a disjoint union of copies of $\mathbb{PT}$'.

---

## Editorial Decision

resubmitted